# Ultra-sensitive metaproteomics redefines the dark metaproteome, uncovering host-microbiome interactions and drug targets in intestinal diseases

Feng Xian [1], Malena Brenek[1], Christoph Krisp[2], Elisabeth Urbauer[3],
Ranjith Kumar Ravi Kumar [1], Doriane Aguanno[3], Tharan Srikumar [2], Qixin Liu[4],
Allison M. Barry [1], Bin Ma [4], Jonathan Krieger[2], Dirk Haller [3,5],
Manuela Schmidt [1,6] & David Gómez-Varela [1,6] ✉

The functional characterization of host-gut microbiome interactions remains limited by the sensitivity of current metaproteomic approaches. Here, we present uMetaP, an ultra-sensitive workflow combining advanced LC-MS technologies with an FDR-validated de novo sequencing strategy, novoMP. uMetaP markedly expands functional coverage and improves the taxonomic detection limit of the gut dark metaproteome by 5000-fold, enabling precise detection and quantification of low-abundance microbial and host proteins. Applied to a mouse model of intestinal injury, uMetaP revealed host-microbiome functional networks underlying tissue damage, beyond genomic findings. Orthogonal validation using transcriptomic data from Crohn's disease patients confirmed key host protein alterations. Furthermore, we introduce the concept of a druggable metaproteome, mapping functional targets within the host and microbiota. By redefining the sensitivity limits of metaproteomics, uMetaP provides a highly valuable framework for advancing microbiome research and developing therapeutic strategies for microbiome-related diseases.

The gut microbiome, a densely populated and diverse ecosystem, plays a central role in host physiology, contributing to immune development, nutrient metabolism, and protection against pathogens[1]. Although a core set of bacterial species is commonly shared among individuals, a large proportion of microbiome variability is driven by medium- and low-abundance taxa, which together shape personalized microbiomes[2–4]. This variability contributes to personalized microbiomes and challenges the concept of a unique healthy microbiome[5]. Genomic approaches, particularly metagenomics and 16S rRNA sequencing, have greatly expanded our knowledge of taxonomic diversity. However, due to functional redundancy across taxa, taxonomy alone offers limited insights into the actual biological functions carried out within a microbial community.

Metaproteomics has emerged as a promising tool to directly quantify microbial and host protein functions in complex environments[6]. Yet, the field remains constrained by several technical limitations: insufficient sensitivity to detect low-abundance species,

[1]Center of Excellence for Metaproteomics and Systems Biology of Pain Laboratory, Division of Pharmacology & Toxicology, Department of Pharmaceutical Sciences, Faculty of Life Sciences, University of Vienna, Vienna, Austria. [2]Bruker Daltonics GmbH & Co. KG, Bremen, Germany. [3]Chair of Nutrition and Immunology, Technical University of Munich, Freising, Germany. [4]Rapid Novor, Kitchener, ON, Canada. [5]ZIEL - Institute for Food & Health, Technical University of Munich, Freising, Germany. [6]These authors contributed equally: Manuela Schmidt, David Gómez-Varela. ✉e-mail: david.gomez.varela@univie.ac.at

incomplete reference databases (DBs), and the underutilization of mass spectrometric data. More than 80% of microbial species detected by genomic methods remain undetected at the protein level, constituting the so-called dark metaproteome[4]. As a result, many ecologically or clinically relevant microbial players, particularly those present at low abundance, are effectively invisible to current metaproteomic methods. Significant improvements in the sensitivity of metaproteomic approaches are needed to explore the highly complex and largely uncharted functional landscape of the gut microbiome.

Overcoming these barriers requires innovations in both hardware and computational strategies. Recent advances in mass spectrometry, such as trapped ion mobility spectrometry coupled with parallel accumulation–serial fragmentation (PASEF), provide higher sensitivity and throughput[7–9]. However, conventional DB search approaches often fail to leverage the full depth of acquired data, particularly for peptides lacking corresponding sequences in reference DBs. De novo peptide sequencing offers a powerful alternative, but its application in metaproteomics has been limited by poor control of false discoveries, especially in the highly complex spectral space of environmental samples[10,11]. There is thus a pressing need for metaproteomic workflows that combine next-generation instrumentation with well-controlled, high-confidence peptide identification strategies to enable deeper, more accurate functional profiling of microbiomes.

Here, we show that uMetaP, an ultra-sensitive metaproteomic workflow integrating advanced LC-MS technologies with an FDR-validated de novo sequencing pipeline (novoMP), redefines the detection limits of the gut dark metaproteome. uMetaP markedly improves the identification and quantification of low-abundance microbial and host proteins and expands taxonomic and functional coverage in mouse gut samples. In a transgenic mouse model of intestinal injury, uMetaP reveals microbial community shifts and metabolic adaptations that extend beyond genomic findings[12]. Importantly, we validate key host protein changes using previously published transcriptomic data from Crohn's disease patients[13]. We further introduce the concept of a druggable metaproteome, identifying functional targets across host and microbial networks with potential therapeutic relevance. Our findings establish uMetaP as a powerful platform for uncovering host-microbiome interactions and advancing precision approaches to microbiome-related diseases.

## Results

### uMetap enables novoMP: a de novo sequencing strategy improving metaproteomic database construction

Our previous work introduced the benefits of Parallel Accumulation Serial Fragmentation (PASEF) in metaproteomics, including during the construction of a metaproteomic DB[14]. Remarkably, analysis of the same eight peptide fractions using the technological solutions integrated into uMetaP (including the timsTOF Ultra mass spectrometer) enabled the fragmentation of 4 times more precursor ions than when using our previous workflow based on a timsTOF Pro mass spectrometer (Fig. 1a), resulting in 4 times more identified peptides (129,425 vs. 30,460; Supplementary Fig. 1a) and a significant shift towards higher peptide intensities (Supplementary Fig. 1b). Despite this considerable improvement, the classical DB search identified fewer than 30% of the precursors fragmented (Fig. 1a), leaving most biological data uncharacterized. We hypothesized that a de novo search strategy, which does not rely on a target sequence DB, could rescue part of this valuable information. However, to our knowledge, no published de novo algorithms have been trained in the 4-dimensional data structure of PASEF. Moreover, previous studies applying de novo for metaproteomic DB construction lacked methodologies to test the confidence of peptide assignments[10]. This is especially critical in metaproteomics due to the immense peptide landscape of these complex samples[11]. We constructed novoMP, a strategy integrating the first algorithm, to the best of our knowledge, trained in PASEF data structure, together with a

multi-layered quality control filtering strategy to rigorously select high-confidence de novo peptide-spectrum matches (PSMs), and an FDR-validation step (see "Methods" for details).

We implemented a custom version of Novor[15] (BPS-Novor) that was generated using over 1.750,000 PSMs from PASEF data acquired on various timsTOF platforms (see "Methods" for details). The evaluation in a human-E. coli-yeast dataset not used during model training shows how the post-trained model maintains higher precision as recall increases compared to the pre-training model (Fig. 1b; Supplementary Fig. 1c). These improvements result in an average of 5–7% gains concerning correct amino acid and peptide assignments in human, E. coli, and yeast peptides (Supplementary Fig. 1d). Similar improvements were found when samples were prepared with various enzymes (Supplementary Fig. 1e). Next, we applied a multi-layered filtering strategy to the BPS-Novor peptide callouts derived from the analysis of pH-fractionated mouse fecal peptides acquired using Data-Dependent Acquisition (DDA)-PASEF. As a result, solely novoMP-derived peptides and annotated species counts decreased as the filtering steps progressed (Fig. 1c and Supplementary Fig. 1f–k). In comparison to taxonomy annotation using only peptides from classic database searches (DB-search), the integration of de novo peptides (Combined) improved taxonomic coverage, particularly for archaea, fungi, and viruses (Fig. 1d). Of a total of 774 annotated species (Fig. 1e and Supplementary Data 1) from all peptides (DB-search + novoMP), only 223 species could have been identified by using solely DB-search peptides (aka. DB-search alone would have discovered a minimum of three species-specific peptides). Detailed analysis revealed the gains in taxonomic coverage reached by novoMP. For example, there is a marked increase in the number of peptides representing the above-mentioned 223 species when including de novo data (Fig. 1f), compared to using DB-search peptides alone (Fig. 1g). Moreover, the combination of peptides from DB-search + novoMP (Combined strategy) enabled the annotation of 551 additional species, suggesting a potential increase in taxonomic coverage of up to 247% (Fig. 1e). Applying novoMP to archived DDA-PASEF data from our previous study[14], increased the taxonomic coverage by 139 % (from 89 to 213 species; Supplementary Fig. 1l). The bigger gains enabled by novoMP in our new dataset, together with the taxonomic overlap among these independent sets of samples (Supplementary Fig. 1m; uMetaP discovers 90% of species from our previous study using a timsTOF Pro), demonstrated the benefit of novoMP to access valuable but otherwise hidden precursor information produced by the latest mass spectrometry technology.

Unlike DB-search, de novo sequencing does not inherently assign proteins to detected peptides. Thus, we conducted BLAST+ homology searches against the NCBI RefSeq DB, applying the same pipeline to both novoMP-derived and DB-search-derived peptides. We set an 80% sequence identity threshold between query sequences and references to exclude low-confidence matches (Fig. 1h). This approach retrieved 73,168 and 98,189 protein sequences for DB-search and novoMP-derived peptides, respectively, with 13,153 shared between the two (Fig. 1i), totaling 158,204 unique protein sequences. Finally, we add 53,502 proteins identified through the classic DB-search against the mouse gut MGnify catalog. As a result, we created a carefully curated mouse fecal metaproteomic DB comprising 208,254 microbial protein sequences, which is available to the metaproteomic community via PRIDE.

In summary, the uMetaP workflow presented here demonstrated the potential of the latest mass spectrometry instrumentation paired with a purpose-built de novo strategy for DB construction, a critical step in metaproteomics.

### uMetaP powered by DIA-PASEF enhances taxonomic and functional coverage, sensitivity and quantitative precision

Our previous study introduced the benefits of combining Data-Independent Acquisition (DIA)-PASEF with deep neural

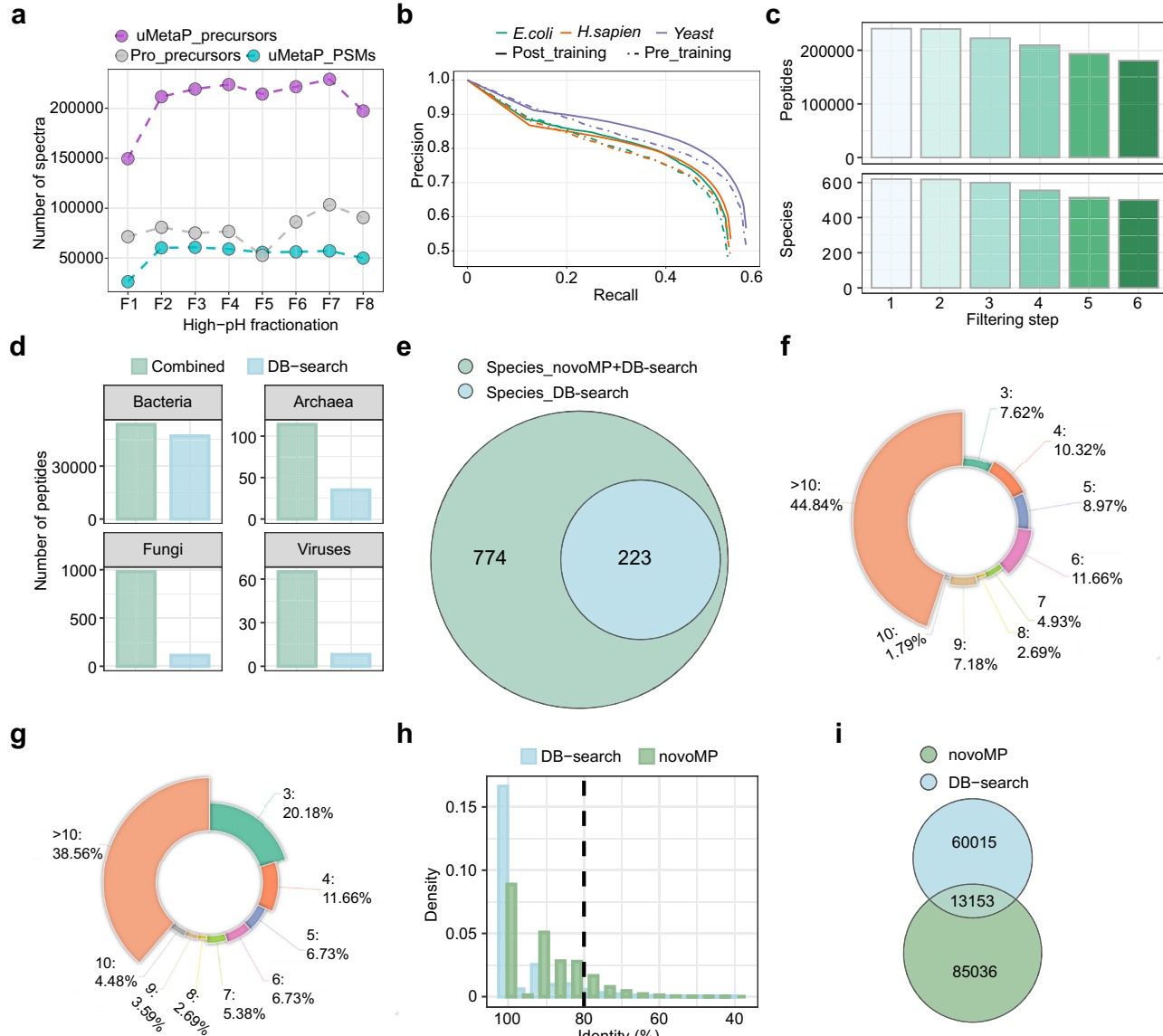

**Fig. 1 | Development and impact of novoMP on metaproteomic database construction. a** Comparison of the number of precursor ions fragmented and the resulting identified peptide-spectrum matches between the uMetaP workflow and our previous workflow across eight high-pH fractions of a pooled mouse fecal sample. **b** Precision-recall curves illustrating the performance of the BPS-Novor algorithm on *E. coli*, *H. sapiens*, and yeast datasets before (pre-) and after (post-) training with PASEF datasets. **c** Reduction in the number of peptides and species identified as filtering steps progress in the novoMP workflow. **d** Taxonomic coverage comparison of peptides annotated via the combined peptides (DB-search + novoMP) and DB-search peptides alone across bacteria, archaea, fungi, and viruses. **e** Venn diagram showing unique and shared species identified using the novoMP integrated strategy versus DB-search alone. Distribution of species-specific peptide counts for 223 species shared between Combined strategies (**f**) and DB-search (**g**). **h** Density plot of sequence identity percentages for BLAST+ homology searches against the NCBI RefSeq database using DB-search peptides and novoMP-derived peptides. A threshold of 80% (black-dotted line) sequence identity was used to filter high-confidence protein matches. **i** Venn diagram comparing protein sequences identified by DB-search and novoMP-derived peptides for the database construction. Source data for (**a**, **b**, **f**, **g**, **h**) are provided in the Source Data file.

network-based data analysis for complex metaproteomic samples[14]. uMetaP powered by DIA-PASEF increased 3–4 times the identifications of microbial and host peptides and proteins compared to our previous workflow[14] (Fig. 2a) when comparing similar conditions. Peptide identifications increased linearly and gradually plateaued at a total of 96,513 peptides (89,128 microbial and 7385 mouse peptides; averaged across three replicates) when 25 ng of peptides were injected over a 30-min gradient. Extending the LC gradient to 66 min further boosted the number of identified peptides and protein groups to 141,811 and 79,693, respectively (averaged across three replicates with 100 ng peptide). Reflecting improved sensitivity, uMetaP detected an average of 200 microbial and 76 host protein groups at an ultra-low sample

amount of 10 pg (Fig. 2a and Supplementary Data 2). uMetaP identified peptides spanning over four orders of magnitude using 25 ng of injected peptides with a 30-min gradient (Supplementary Fig. 2a) and showed a remarkable quantitative precision with more than 84% of peptides exhibiting a coefficient of variation (CV) lower than 0.2 (Supplementary Fig. 2a). In total, 210,051 microbial peptides were identified, with 32,400 of these added by novoMP to the mouse fecal metaproteomic database (novoMP-DB; Fig. 2b).

Our approach represents an orthogonal strategy for FDR control of novoMP-detected peptides, validating their confidence. The evaluation of CScore and posterior error probability (PEP) showed that precursors from the two sources exhibited similar distributions

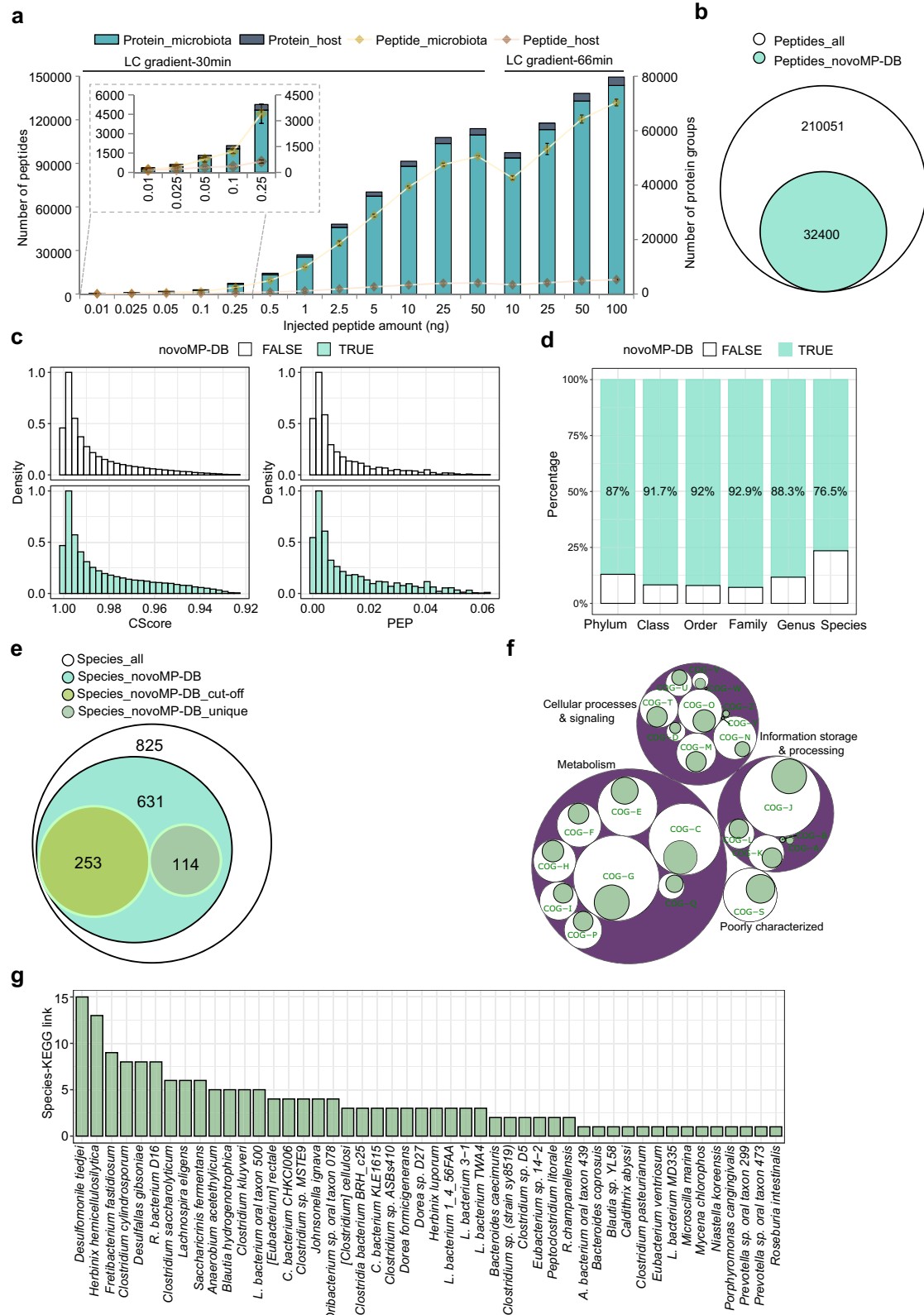

(Fig. 2c). Furthermore, novoMP-DB peptides demonstrated equal or slightly better quantitative precision compared to DB-searched peptides across various sample loadings and gradient lengths (Supplementary Fig. 2b).

At the taxonomic level, novoMP-DB peptides contributed up to 92.9% of annotated taxa at different ranks (Fig. 2d). Across the dataset, 825 species were annotated (using 3 species-specific peptides as cut-off),

with novoMP-DB peptides enabling the detection of 631 (Fig. 2e). This suggests a potential 6-fold improvement in detected species compared to our previous DIA-PASEF workflow[14]. Notably, 253 species would not have met the minimum cutoff of three species-specific peptides without the addition of novoMP-DB peptides, and 114 species were exclusively identified through unique novoMP-DB peptides (Fig. 2e). From the 118,937 identified protein groups (PGs; ProteinIDs_all), 47,739 groups

**Fig. 2 | uMetaP powered by DIA-PASEF enhances metaproteome taxonomic and functional coverage, sensitivity, and quantitative precision. a** Number of identified microbial and host peptides and protein groups using uMetaP powered by DIA-PASEF across varying sample amounts and LC gradient lengths of a pooled mouse fecal sample. Error bars represent the standard deviations across technical triplicates. **b** Venn diagram showing the total microbial peptides identified, including 32,400 additional peptides added by novoMP to the metaproteomic database (novoMP-DB). **c** Distribution of CScore and posterior error probability (PEP) values for peptides originated from novoMP-DB ("TRUE") and from DB-search ("FALSE"). **d** Taxonomic contributions of novoMP-DB peptides across ranks (phylum to species). **e** Venn diagram of species annotated using any type of peptide source (825, Species_all). The 631, Species_novoMP-DB, represents species with novoMP-DB peptides assigned, of which 253 need novoMP-DB peptides to reach

the 3-species-specific peptide cut-off (Species_novoMP-DB_cut-off), and 114 correspond to species uniquely annotated with novoMP-DB peptides (Species_novoMP-DB_unique). The rest of 264 species represent those that were co-annotated with peptides from both DB-search and novoMP-DB, but that could be identified (reaching the 3-species-specific peptide cut-off) through DB-search alone. **f** Functional representation of protein groups mapped to Clusters of Orthologous Genes (COGs). The percentages of proteins solely discovered by novoMP-DB were shown in green. The white bubble area represents the percentages of functional clusters discovered by all detected proteins. **g** Species-KEGG associations uniquely revealed by the addition of novoMP-DB, enabling the identification of 175 unique pathways for 48 species. Source data for (**a**, **d**, **f**, **g**) are provided in the Source Data file.

include proteins originating from novoMP-DB (ProteinIDs_novoMP-DB; i.e., at least one protein in the protein group was derived from novoMP-DB), among which 26,149 include proteins solely identified by novoMP-DB (ProteinIDs_novoMP-DB_unique; i.e., all proteins in the group were from novoMP-DB; Supplementary Fig. 2c). These PGs spanned all 24 functional Clusters of Orthologous Genes (COG) categories, with minor differences in KEGG pathway counts (Supplementary Fig. 2d). Interestingly, proteins unique to novoMP-DB were largely represented in COG categories like RNA processing (COG-A), chromatin dynamics (COG-B), extracellular structures (COG-W), nuclear structure (COG-Y), and cytoskeleton (COG-Z) (Fig. 2f). Overall, uMetaP enabled the study of 199 KEEG additional pathways compared to our previous work[14]. Examining species-to-function links, we demonstrated how novoMP-DB proteins uniquely revealed 175 species-KEGG associations from 48 species (Fig. 2g). Interestingly, uMetaP uncovered previously hidden functions by identifying 1043 proteins (196 proteins originated from novoMP-DB) of unknown function[16] (PUFs). Further, we identified 2342 small proteins[17,18] (sProt; 321 proteins originated from novoMP-DB) and 581 proteins (86 proteins originated from novoMP-DB) with predicted antimicrobial peptide sequences[19] (AMPs; Supplementary Fig. 2e).

Altogether, uMetaP powered by DIA-PASEF, considerably improves taxonomic and functional coverage, as well as quantification quality in complex metaproteomics samples, advancing the study of previously hidden host-microbial interactions.

## Improving the detection limit of dark metaproteome

The human gut microbiome harbors an average of 200 bacterial species[20,21] comprising a core of abundant species present in most individuals[2,3] and a second pool formed by low-abundant species (more than 50% of the total), underlining the increasingly important inter-individual variability of microbiome profiles in health and disease. Current metaproteomic approaches do not achieve sufficient sensitivity to study these low-abundant species[4]. We hypothesized that the benefits of uMetaP to significantly improve the study of this uncharacterized dark metaproteome.

We set out to develop an approach to calculate the real lower limits of detection (LoD) and quantification (LoQ) by calculating the number of bacterial cells that can be accurately identified and quantified in a complex microbial sample. To minimize identification uncertainty, we used stable isotope labeling by amino acids in cell culture (SILAC) for *Ligilactobacillus murinus* (*L. murinus*), a bacterium native to the mouse gut microbiome. DDA-PASEF analysis of the SILAC culture confirmed an average incorporation efficiency of 97.42% (Supplementary Fig. 3a; Supplementary Data 3). Additionally, we employed *Salinibacter ruber* (*S. ruber*) as an exogenous spiked bacterium (Fig. 3a). We observed that the number of detected peptides and protein groups declined as we decreased the number of SILAC-labeled *L. murinus* and unlabeled *S. ruber* cells spiked into 10 mg of mouse feces (Fig. 3a, b; Supplementary Fig. 3b). After applying strict filtering criteria for taxonomic identifications (including a non-spiked

control; see "Methods"), we identified 6 and 20 peptides for *L. murinus* and *S. ruber*, respectively, when spiked 10,000 cells (Fig. 3b; Supplementary Data 4). Visual inspection of selected spectra confirmed these identifications (examples in Supplementary Fig. 3c, d). By extracting precursor ions and fragments from DIA-PASEF spectra, we determined a reliable LoQ of 1 million *L. murinus* cells and 5 million *S. ruber* cells (examples in Fig. 3c, d). The differences in LoQ for each bacterium possibly reflect differences in bacterial size (Supplementary Fig. 3e) and protein content.

Species abundance within a microbial community is an important parameter for microbiome studies. By summing the intensities of species-specific peptides, we showed that uMetaP abundance assessments are driven by a limited number of species, with just eight species (excluding spiked *L. murinus* and *S. ruber*) accounting for 53.5% of the microbiota biomass (Fig. 3e; Supplementary Data 5). Among 115,127 peptides identified in this dataset, 21,457 can be traced back to the novoMP-DB (Supplementary Fig. 3f), which contributed to the detection of species down to 0.006% relative abundance. Peptide intensity analysis indicated that the 10,000 spiked *L. murinus* and *S. ruber* cells detected by uMetaP constituted 0.0003% and 0.0159% of the total biomass, respectively (Fig. 3f; Supplementary Data 5). Considering the spectral quality of precursor and fragment ions, we confidently quantified these spiked bacteria, representing 0.0044% for *L. murinus* and 0.0297% for *S. ruber* (Fig. 3f). Based on genomic estimates (which assume $1 \times 10^{12}$ bacterial cells per gram of mouse feces[22]), we achieved a LoD of 0.0001% (1 cell detected among 1 million) for both species, and an LoQ of 0.01% and 0.05% for *L. murinus* and *S. ruber*, respectively. These results improved the previously reported limits in metaproteomics[4] by up to 5000-fold. Additionally, functional annotation of identified protein groups to KEGG pathways diminished below 100 million bacteria and plateaued at 1 million bacteria (Fig. 3g). We annotated 85 and 18 functional pathways with as few as 10,000 *L. murinus* and *S. ruber* cells, encompassing a variety of metabolic and biosynthetic pathways.

Our data lower detection and quantification limits in complex metaproteomic samples, enabling a more precise definition of individual functional microbiomes.

## Shedding light on microbial-metabolic circuits underlining tissue injury during intestinal inflammation in vivo

The mutual relationship between the microbiome and the host is essential for maintaining intestinal homeostasis, and its disruption plays a role in the onset and progression of diseases, including inflammatory bowel diseases (IBD)[23,24]. It has been recently demonstrated how mitochondrial (MT) perturbation in intestinal epithelial cells (IECs) causes metabolic injury, a self-autonomous mechanism of tissue wounding associated with microbial dysbiosis[12], and triggers the recurrence of chronic intestinal inflammation[25]. Beyond taxonomic associations using shallow shotgun metagenomics, how metabolic changes in the intestinal epithelium select the growth of certain bacteria and how specific bacteria interfere with epithelial regeneration

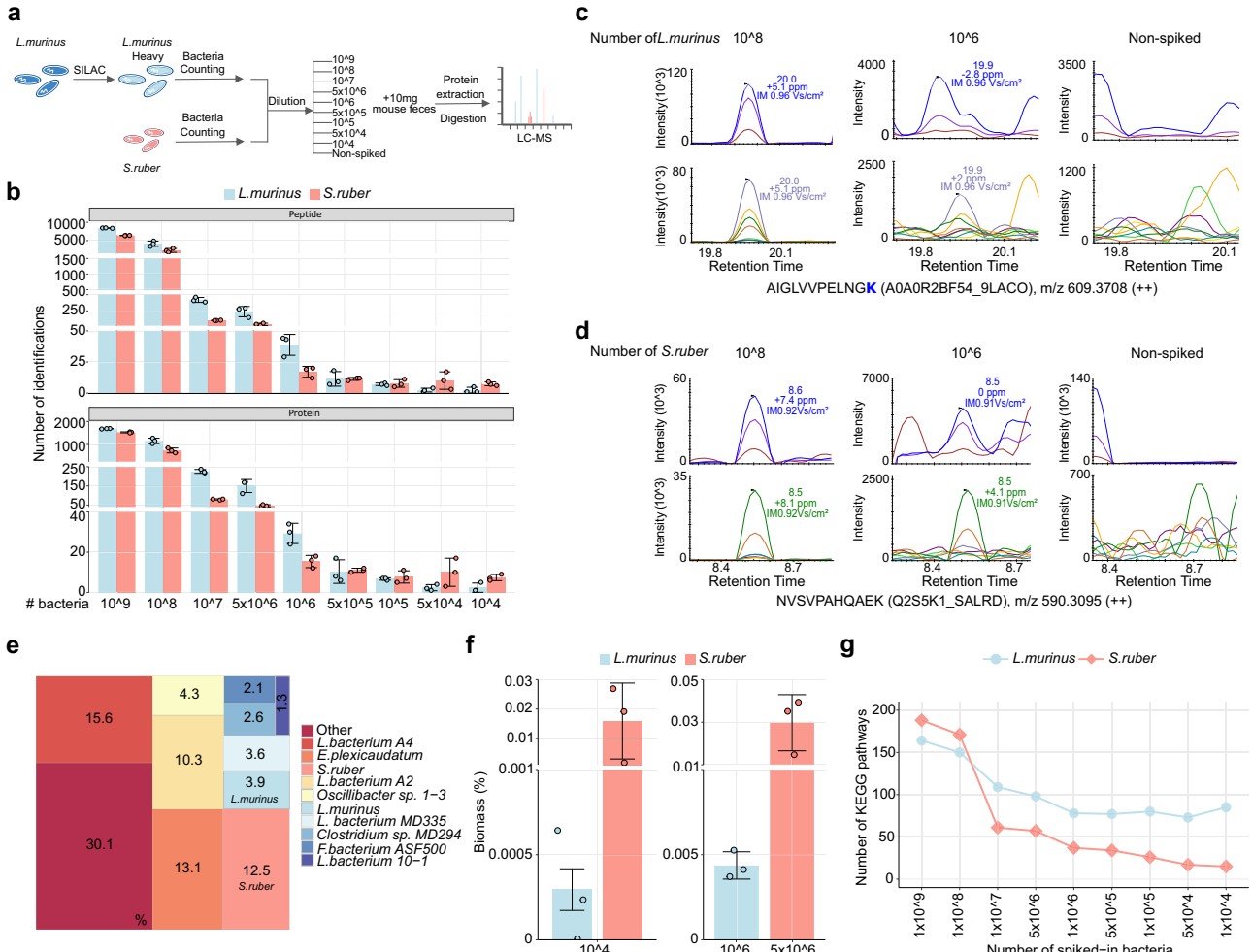

**Fig. 3 | Improving the detection limit of the dark metaproteome using uMetaP workflow. a** Workflow combining SILAC labeling and spiking of *Ligilactobacillus murinus* (*L. murinus*) and spiking of unlabeled exogenous *Salinibacter ruber* (*S. ruber*) into a mouse fecal sample. Each spike-in includes three replicates (*n* = 3). **b** Number of peptides (top) and protein groups (bottom) identified for *L. murinus* and *S. ruber* across bacterial cell inputs. Bar plots represent the mean values, and error bars show the standard deviations across spike-in triplicates (*n* = 3). The mean and standard deviations values are provided in the Source Data file. Representative extracted MS/MS spectra from complex DIA-PASEF data of peptides from *L. murinus* (**c**) and *S. ruber* (**d**) to illustrate reliable limit of quantification (LoQ), in

comparison with the $10^8$ spike-in and the non-spiked control. **e** Microbial community biomass composition ($10^8$ spike-in samples) showing that eight species (excluding spiked *L. murinus* and *S. ruber*) account for 53.5% of the total microbiota biomass. **f** Biomass percentages (presented as mean ± standard deviations across spike-in triplicates) for spiked *L. murinus* and *S. ruber* cells at $10^4$ (LoD; 0.000295 ± 0.000122 and 0.015900 ± 0.012931, respectively), $10^6$ (LoQ for *L. murinus*; 0.004368 ± 0.000803), and $5 × 10^6$ (LoQ for *S. ruber*; 0.029741 ± 0.013172) input levels. **g** Number of KEGG pathways annotated from identified protein groups as bacterial inputs decreased. Source data for (**b**, **e**, **f**) are provided in the Source Data file.

are unknown, which precludes understanding of host-microbiome interactions and defining potential therapeutic targets. We set uMetaP to the test by investigating the dynamics of microbial-host circuits in recurrent intestinal inflammation in vivo.

To test the role of MT function in epithelial stem cell homeostasis, we took advantage of a published model of MT dysfunction in the intestinal epithelium, in which the MT chaperone heat shock protein 60 (Hsp60) is transiently deleted, specifically in mouse IECs (Fig. 4a). This deletion triggered temporary MT dysfunction, leading to metabolic stress and transient structural changes in the colonic epithelium similar to the ones observed in patients suffering from intestinal inflammatory diseases[12]. We explored the microbiome shifts and the host functional changes during tissue injury by analyzing the colonic contents from control (Hsp60$^{fl/fl}$) and metabolic injured (Hsp60$^{\Delta/\Delta IEC}$) mice at two-time points after tamoxifen cessation: day 0 (D0) which corresponds to Hsp60 complete loss but there are no apparent histological aberrations, and D8 corresponding to the peak of metabolic injury[12].

Overall, peptide and protein identifications revealed distinct proteomic profiles between the two genotypes, with both host and microbial IDs decreasing at D8 in the metabolic injured model (Supplementary Fig. 4a). The metabolic injury phenotype was associated with significant changes in microbiota richness (Fig. 4b) and community structure (β-diversity; Fig. 4c). Differential abundance analysis identified 16 significantly altered genera across the four conditions, with 10 genera, including *Bacteroides*, *Anaerotruncus*, and *Parabacteroides*, showing increased abundance during injury, while *Lactobacillus*, for instance, decreased at D8 compared to the other groups (Fig. 4d; Supplementary Fig. 4b), reflecting microbiome taxonomic adaptation to IEC dysfunction. At the species level, 25 and 18 differentially abundant species were observed at D0 and D8, respectively, in response to metabolic injury (fl/fl compared to Δ/ΔIEC; Supplementary Fig. 4c, d). Four species were commonly regulated at both time points, with *Bacteroides caecimuris* uniquely showing increased abundance from D0 on already and more accentuated at D8 (Fig. 4e), suggesting their potential role in modulating colon metabolic injury. Remarkably,

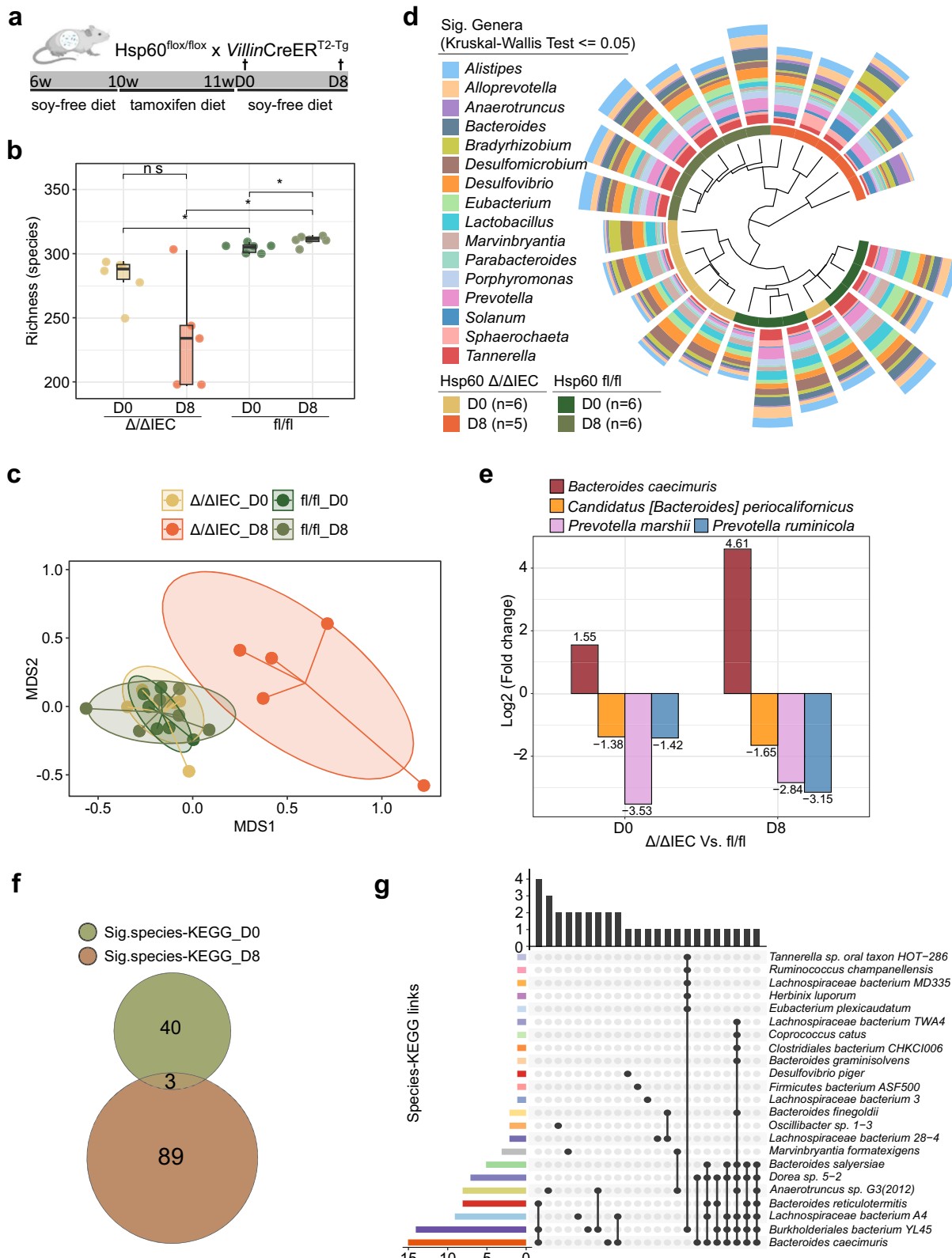

the above-reported taxonomic changes discovered by uMetaP mirrored (e.g., β-diversity, an increase of *Bacteroides caecimuris*) and extended (previously unreported changes in several genera and species) the metagenomic findings on these same samples[12].

Functional analysis indicated 43 and 92 significantly regulated species-KEGG pathways at D0 and D8, respectively, with only three pathways in common (Fig. 4f and Supplementary Data 6),

underscoring distinct functional shifts within the microbial community in response to injury. Notably, while *Bacteroides caecimuris* increased in abundance at both time points, it only showed significant KEGG pathway alterations at D8 (15 pathways; Fig. 4g and Supplementary Fig. 4e). Detailed analysis revealed that *Bacteroides caecimuris* was the only species showing a significant alteration of two pathways at D8 - Carbon fixation by Calvin cycle and Biosynthesis of Ansamycins

**Fig. 4 | uMetaP sheds light on microbial-metabolic circuits underlining tissue injury during intestinal inflammation in vivo. a** Schematic of the experimental design showing the strategy for the transient deletion of mitochondrial heat shock protein 60 (Hsp60) in mouse intestinal epithelial cells (IECs) using tamoxifen induction. Colonic contents from control (Hsp60$^{fl/fl}$) and injured (Hsp60$^{\Delta/\Delta IEC}$) mice were analyzed at day 0 (D0) and day 8 (D8) after the tamoxifen diet. Adapted from ref. 12, with permission from Elsevier. **b** Microbial richness (alpha-diversity) in the four experimental groups. Box plots show the distribution of species richness per sample group. Each box represents the inter-quartile range, spanning from the 25th percentile (lower bound of the box) to the 75th percentile (upper bound). The center line indicates the median (50th percentile). Asterisks indicate statistical significance from two-sided unpaired *t*-tests for comparisons: ns = not significant,

* representing $p < 0.05$. **c** β-diversity showing distinct microbial community structures between control and injured mice across D0 and D8. **d** Dendrogram shows the relative abundance of 16 genera significantly changed in response to metabolic injury discovered by uMetaP. The significance level was set with an adjusted *p* value ≤ 0.05 (Benjamini-Hochberg) after Kruskal–Wallis test. **e** Differential abundance of four commonly regulated species at D0 and D8. **f** Overlap of significantly regulated (adjusted *p* value ≤ 0.05; moderated two-sided *t*-tests followed by Benjamini-Hochberg for adjusted *p* value) species-KEGG pathways at D0 and D8. **g** UpSet plot showing uniqueness and shared presence of significantly regulated KEGG pathways among 23 species at D8. Source data for (**b**, **c**, **d**, **g**) are provided in the Source Data file.

(Fig. 4g and Supplementary Data 6), offering mechanistic hints on how this bacterium might modulate colonic microenvironments to dominate over other taxa during metabolic injury[12].

These data showcase the benefits of uMetaP, beyond genomic findings, to shed light on taxonomic and functional alterations underlying intestinal tissue injury in vivo.

## Defining the druggable gut metaproteome: Mining the microbiota and host proteome offers potential therapeutic targets in human intestinal inflammatory diseases

Similar to the druggable genome in cancer research[26], where genes are prioritized for therapeutic targeting, mining the gut metaproteome could allow researchers to identify proteins influencing host-microbiota interactions during disease states as prime candidates for therapeutic intervention, particularly in IBD. However, the concept of the druggable gut metaproteome remains unexplored. We applied uMetaP to characterize proteome changes in both microbiota and host during intestinal tissue injury.

We first focused on microbiota-specific activities during inflammation. At day 8, we identified 593 significantly regulated microbial COGs, with 216 linked to 59 distinct functions (Supplementary Data 7). Vitamin biosynthesis emerged prominently, including B2 (riboflavin), B7, B9, and B12 pathways (Fig. 5a). These vitamins, essential for host metabolism and immune regulation, largely depend on microbial synthesis. Specifically, riboflavin biosynthesis showed down-regulation of COG0108 (EC 4.1.99.12) and up-regulation of COG0196 (EC 2.7.7.2), alongside decreased expression of pentose phosphate pathway enzymes (Fig. 5b), suggesting a potential deficiency in microbial riboflavin production in injured mice ($\Delta/\Delta IEC\_D8$)—a finding supported by emerging evidence of vitamin B2's role in gut immune homeostasis[27].

Host proteome analysis revealed a stronger response at D8 compared to D0, with 990 versus 144 significantly regulated proteins (Supplementary Fig. 5a, c). Consistent with the observed functional shifts in the microbial community (Fig. 4f), there was minimal overlap in enriched host functions between D0 (Supplementary Fig. 5b) and D8 (Supplementary Fig. 5d, Supplementary Data 8). Reflecting the injury-induced inflammatory environment, host functional shifts at D8 highlighted tissue homeostasis, epithelial development, cytokine regulation, and ERK1/2 signaling pathways (Supplementary Fig. 5d, Supplementary Data 8). We obtained orthogonal validation of the metaproteomic findings by comparison with transcriptome data obtained from mucosal biopsies of Crohn's disease patients[13] (CD; 343 samples from 204 patients originating from inflamed ileum or M0I, and the non-inflamed ileal margin or M0M). We found 490 proteins significantly regulated at D8 that were also presented in the list of regulated human transcripts (Supplementary Data 9), where several proteins (e.g., RELA[28], NOS2[29], and ITGAM[30]) are reported to be linked to intestinal inflammatory diseases in humans.

To prioritize therapeutic targets, we narrowed this list using mouse colonic transcriptomics[12], retaining 97 proteins expressed across datasets (Supplementary Data 9), and further refined to 33

proteins consistently regulated (with concordant directionality) across mouse fecal metaproteomics, human, and mouse transcriptomics (Fig. 5c). Functional annotation linked these to metabolic processes, oxidoreductase activity, and MT components (Supplementary Fig. 5e, Supplementary Data 10). Protein interaction network analysis highlighted hub proteins like Rela[28], Nos2[29], and Itgam[30] with high centrality scores (Fig. 5d), reaffirming their key roles in intestinal inflammation. Drug-gene interaction mapping resulted in 204 interactions corresponding to 187 drugs (Supplementary Data 11) for 20 out of the 33 genes/proteins. Among these, 77 are approved for several indications associated with human inflammatory diseases, including a current treatment for Crohn's disease (e.g., Natalizumab®, targeting ITGB1; Fig. 5e), an anti-inflammatory drug for IBD treatment (e.g., prednisone, targeting ITGB2; Fig. 5e), and other target genes/proteins with high centrality in the interaction network (e.g., Hydrocortisone targeting ITGAM; Fig. 5e). We also revealed 57 drugs approved for other indications (Fig. 5e; e.g., rhil-11 and clarithromycin) and 110 investigational compounds, expanding potential therapeutic avenues (Supplementary Data 11).

Finally, integrating recently reported drug-microbiota interaction data[31], we found several approved drugs known to modulate microbial taxa altered in our model (Supplementary Fig. 5f). For instance, *Bacteroides*, enriched in Δ/ΔIEC_D8 mice (Supplementary Fig. 4c), was reduced upon exposure to nimodipine, indomethacin, and fluoxetine[31], suggesting potential synergy between host-directed therapies and microbiota modulation. Similarly, *Desulfovibrio*, reduced in our model (Δ/ΔIEC_D8), showed increased abundance with tacrolimus. However, antagonistic effects can also be observed for the same drug treatments, i.e., decreased expression of *Eubacterium* with nimodipine and tacrolimus. Thus, the therapeutic potential of dual-target strategies should be considered carefully.

As a result, our analysis establishes the concept of the druggable metaproteome as an ecological network of microbial functions and host protein hubs that can be pharmacologically targeted. Through systematic mapping of protein functions and drug interactions, uMetaP enables the identification of candidate therapies with potential synergy across host and microbiota, supporting hypothesis generation and prioritization of targets for clinical decision-making in intestinal inflammatory diseases.

## Discussion

Metaproteomics holds significant potential to advance microbiome research. However, current methods struggle to achieve high sensitivity, deep protein profiling, and quantitative accuracy and precision. As a result, many medium- and low-abundance taxa identified by widely used genomic methods, along with their functional repertoires, often remain uncharacterized. In uMetaP, we combined latest LC-MS technology with a de novo strategy to address these limitations. We demonstrated the benefits of uMetaP in meaningful biological scenarios using an in vivo mouse model of metabolic injury and by benchmarking our findings against human transcriptomic data from Crohn's patients. The translational potential of our data was

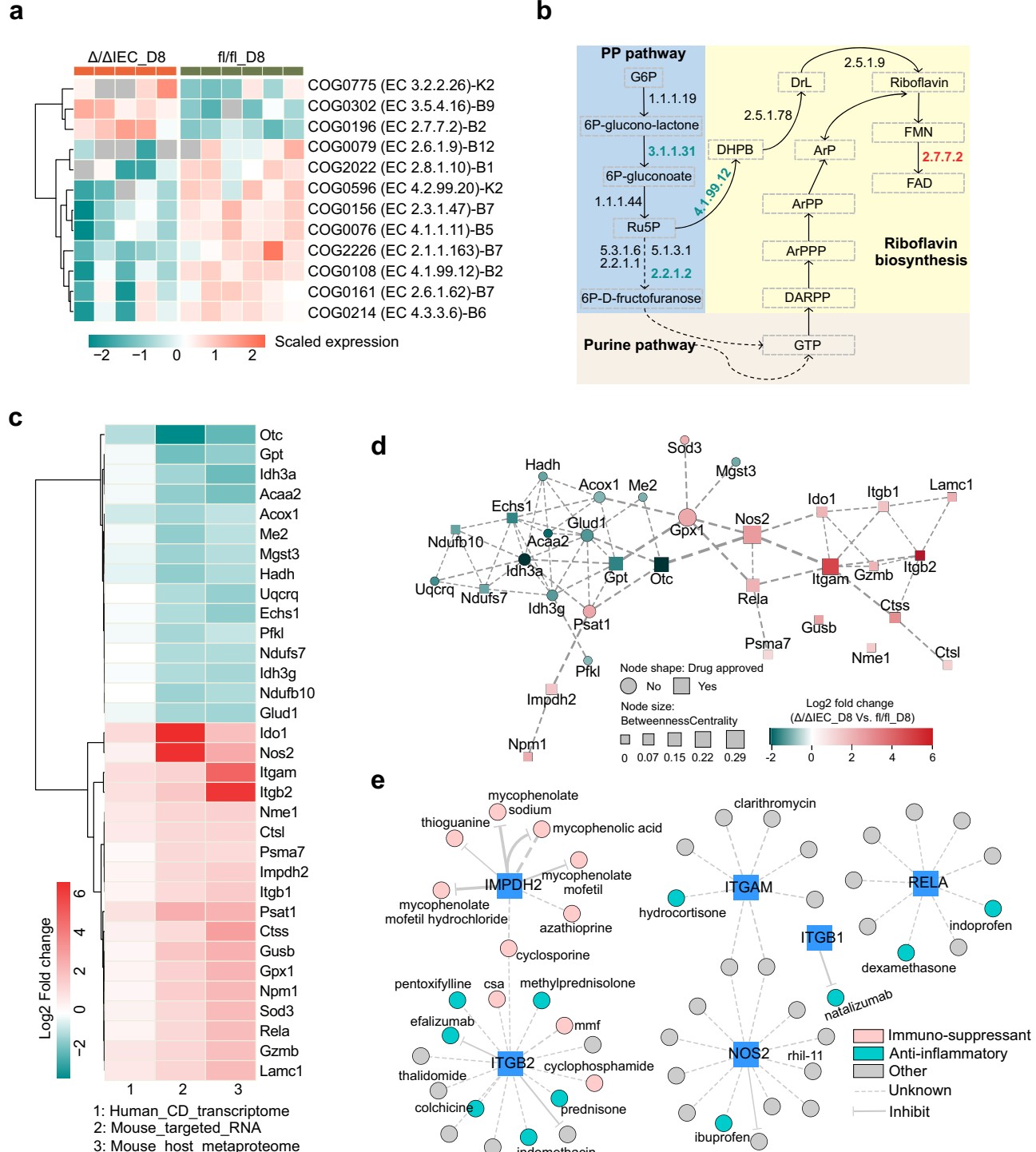

**Fig. 5 | Mining microbial and host proteome changes during intestinal tissue injury defines the druggable metaproteome and reveals potential therapeutic targets for inflammatory diseases. a** Scaled expression of COG IDs involved in Vitamin biosynthesis in Δ/ΔIEC_D8 and fl/fl_D8 mice. **b** Overview of the riboflavin biosynthesis pathway and related metabolic routes (adapted from Yang, B. et al.[64] and MetaCyc[65]). Enzymes detected in our dataset that catalyze the respective reactions are indicated with EC numbers. Enzyme labels are color-coded: black denotes no significant change between Δ/ΔIEC_D8 and fl/fl_D8 mice, cyan indicates down-regulation, and red indicates up-regulation in Δ/ΔIEC_D8 mice. **c** Heatmap showing the directionality of log2 fold changes for 33 proteins consistently regulated in mouse metaproteomics (colonic contents), mouse targeted RNA analysis (colon tissue), and human transcriptomics datasets from Crohn's patients (ileum biopsy). **d** Protein-protein interaction network analysis of 33 proteins. Node shapes indicate whether the target protein has approved drugs based on drug-gene interaction analysis. **e** Examples of drug-gene interaction network of ITGAM, ITGB2, RELA, IMPDH2, NOS2, and ITGB1. Node colors represent therapeutic categories (e.g., anti-inflammatory, immunosuppressant), and the edge types indicate the drug effects (inhibitory or unknown).

demonstrated by detailed drug-gene analysis, enabling hypothesis-driven drug repurposing efforts.

In recent years, metaproteomics has advanced by integrating sophisticated mass spectrometry platforms[14,32], superior data acquisition methods (e.g., DIA-PASEF[14]), and machine-learning-based data analysis[14,33]. Nonetheless, metaproteomic studies are still limited by DB construction, with classic approaches relying on reference catalogs and DB-search workflows. These capture only a small fraction of proteomic diversity within complex samples. Our data show that around 70% of the spectral information acquired by uMetaP has not been utilized for result generation. Thus, metaproteomics would greatly benefit from de novo sequencing solutions. However, the spectral complexity of these samples and the lack of methods for controlling de novo peptide confidence limit its application[10].

We addressed this challenge by developing novoMP, a de novo strategy tailored for metaproteomics DB construction. Compared to previous studies[10], novoMP encompasses three key aspects. First, it represents a de novo algorithm trained on the PASEF data structure, obtained from various timsTOF platforms, different species, and using different cleavage enzymes. Second, it implements a multi-layered quality control pipeline to rigorously select high-confidence de novo PSMs. Third, it offers orthogonal FDR validation using DIA-PASEF, demonstrating confidence in novoMP-DB peptides on a par with peptides obtained via classical DB search workflows. Due to the fundamental algorithmic differences between de novo sequencing (i.e., BPS-Novor) and classical DB-search methods (i.e., FragPipe), the two approaches shared 87 annotated species in common (Supplementary Fig. 6a). Further assessment of de novo peptide confidence mapped to the 87 shared species versus those uniquely assigned to the 414 species detected by novoMP revealed similar de novo score distributions (Supplementary Fig. 6b). These results suggest comparable confidence levels of de novo peptides, regardless whether the species were shared with the DB-search or unique to novoMP. Our de novo-based approach enhanced taxonomy-level resolution of the gut microbiome by enabling detection of underrepresented taxa—particularly Eukaryota such as fungi (Supplementary Fig. 6c)—underscoring novoMP's value for capturing microbial diversity beyond the bacterial domain.

We significantly expanded the taxonomic and functional representation in metaproteomic DBs by combining the depth, sensitivity, and spectral quality of uMetaP with novoMP. However, it is important to note that the estimated improvements in taxonomic profiling presented in this study may represent an upper bound. These estimates should be interpreted with caution due to current limitations in metaproteomics, including incomplete reference DBs, uncertainty in species-level assignment thresholds[34], and the absence of robust FDR estimation methods for taxonomic annotation in complex metaproteomic samples.

Interestingly, the DIA-PASEF analysis with solely novoMP-DB covers more than 99% of the COG and KEGG pathways (Supplementary Fig. 2d), strongly suggesting the possibility of reaching maximum functional coverage without the need for metaproteomic DB construction in future studies. Moreover, the benefits of novoMP for analyzing our previously published DDA-PASEF data could be extended to PASEF datasets acquired in prior metaproteomic studies[33–35]. For the metaproteomic community, we provide a roadmap for increasing confidence in de novo solutions and offer an extensive mouse metaproteomic DB composed of 208,254 proteins, representing 774 microbial species and 447 KEGG pathways.

Combined with DIA-PASEF, uMetaP surpasses current proteotyping standards[4], the most optimistic performance forecast in the field[36], and a preliminary evaluation of the Orbitrap Astral mass spectrometer[32]. The fast analysis times and the low variability reached make uMetaP a promising tool for large-scale metaproteomic studies. Integrating novoMP into DB construction enabled the identification of an additional 28% of proteins, and an estimated 80% increase in

taxonomy detection (Supplementary Fig. 2c; Fig. 2e). While our study demonstrates the capability of uMetaP to detect peptides at ultra-low inputs (e.g., 10 pg; Fig. 2a), we acknowledge the general challenges of peptide identification confidence at these input levels, a known limitation in proteomics. A closer look into peptides detected at 10 pg but absent from 100 ng samples (Supplementary Fig. 7a) revealed that none were uniquely identified at 10 pg (Supplementary Fig. 7b), and most showed strong ion signals in higher-input (i.e., 100 ng) datasets when manually inspected (example in Supplementary Fig. 7c). This suggests their absence in quantitative outputs is largely due to false negatives rather than false positives. Such missingness likely reflects stochastic effects in MS/MS acquisition and downstream FDR recalculations during data processing, especially in very complex samples. These findings highlight the importance of cautious interpretation of low-input data while reinforcing the analytical sensitivity achieved with uMetaP. Beyond identification, uMetaP demonstrates precision, reliability in quantification, and ultra-high sensitivity. These benefits were demonstrated by establishing a reliable LoD and LoQ in a complex metaproteome. Unlike previous approaches[4], we accounted for sample preparation losses by using a SILAC-labeled bacterium (*L. murinus*) and an exogenous bacterium (*S. ruber*). As a result, uMetaP can potentially detect a single bacterium in a theoretical background of 1 million, representing a 5000-fold improvement[4]. Importantly, MS2 spectra showed a shift in the reliable quantification limit for *L. murinus* and *S. ruber*, which is likely applicable across metaproteomes and highlights the importance of rigorous spectral quality control for accurate peptide quantification. Establishing LoD and LoQ for the gut dark metaproteome has important implications. By lowering the thresholds for reliably identifying and quantifying bacterial species and their protein products, researchers can better capture the functional contributions of often-overlooked low-abundance species. This is critical for fields that require ultra-sensitivity—from marine metaproteomics[37] to clinical metaproteomics, where subtle but clinically important changes in pathogenic microorganisms demand early detection. Moreover, reliable quantification of medium- and low-abundance species will help answer key questions about individualized and healthy microbiome profiles[2,3].

Our results in a transgenic mouse model of colonic tissue injury showcase the potential of uMetaP for discovering host-microbiome interactions in a relevant in vivo context. In addition to mirroring the taxonomic findings reported using genomic methods[12], uMetaP offers greater sensitivity, allowing earlier detection of taxonomic and functional alterations underlying causal disease mechanisms. Urbauer et al.[12] demonstrated that *Bacteroides caecimuris* increases in abundance during metabolic injury at 8 days after tissue injury and that mono-colonization of germ-free Hsp60 knock-out mice with *B. caecimuris* is sufficient to recapitulate the disease phenotype. Similarly, our data showed an increase in *B. caecimuris* at day 8. Notably, uMetaP also detected this increase during the first 24 h after tamoxifen cessation. This is a significant improvement in the temporal sensitivity for detecting taxonomic changes compared to genomic and immunohistochemistry methods. Moreover, we extended the previously known dysbiosis signature[12] by detecting significant abundance changes in additional bacterial species.

*Bacteroides* species are known to adapt well to inflammatory and stressed conditions[38], potentially explaining the observed selective expansion in response to colonic injury. However, the mechanisms leading to this selective advantage remain unknown. Our functional data provide plausible mechanisms by detailing metabolic reprogramming during disease progression in vivo. We identified *B. caecimuris* as the bacterial species with the majority of altered KEGG pathways at day 8. Notably, two altered KEGG pathways were unique to this species: carbon fixation by the Calvin cycle and biosynthesis of ansamycins. The simultaneous upregulation of these pathways may provide a competitive advantage for *B. caecimuris* in the gut

microbiome, especially in the context of MT dysfunction in the intestinal epithelium. On one hand, the impaired MT function caused by the Hsp60 mutation[39] could lead to reduced $CO_2$ production due to decreased TCA cycle activity. Upregulating carbon fixation via the Calvin cycle equips bacteria with greater metabolic flexibility, allowing them to utilize even small amounts of $CO_2$, which could provide an advantage over competitors. On the other hand, ansamycins, such as rifamycins, are antibiotics produced by certain bacteria[40]. In the disrupted gut environment caused by Hsp60 deletion, the selective elimination of sensitive competitors could allow the ansamycin-producing bacteria to dominate. Interestingly, *Bacteroides* species are not primary producers of ansamycins[41]. Our data discovered specific metabolic adaptation by *B. caecimuris*, potentially contributing to its expansion during tissue injury as detected by metagenomics[12] and uMetaP.

Beyond classical functional analysis of host proteins, we explored the translational potential of our findings. The orthogonal inter-species validation with transcriptomic data from Crohn's patient biopsies[13] validated changes in over 400 mouse proteins. While several of the identified host proteins have previously been linked to IBD, our study advances current understanding by integrating host proteomic changes with microbial taxonomic and functional shifts in a multi-omic framework. This approach not only reveals ecosystem-level interactions and early-stage molecular alterations but also uncovers therapeutic opportunities through network-based target prioritization and drug repurposing. Therefore, we introduced and explored the concept of a druggable metaproteome: the collection of host and microbiota proteins or functions within a given ecological environment that may be prioritized for therapeutical targeting. This concept supports drug discovery and repurposing efforts. We identified more than 200 potential drug-protein interactions, including immune-suppressants used in Crohn's disease (e.g., natalizumab), anti-inflammatory drugs for IBD treatment (e.g., prednisone), and approved drugs for diverse indications. By characterizing microbial druggable functions and integrating drug-microbiota interaction data, our data can be harnessed for refined therapeutic strategies within the broader concept of a druggable metaproteome. Follow-up studies using pre-clinical mouse models or human volunteers are needed to test data-driven hypotheses assessing specific drug repurposing regimes or combinatorial treatments.

By integrating cutting-edge LC-MS technology, developing a de novo strategy, testing these advancements in an in vivo disease model, and introducing the concept of the druggable metaproteome, our study advances metaproteomics. Taken together, these data provide great potential for microbiome research in general and, particularly, for characterizing host-microbiome interactions and their roles in health and disease.

## Methods

### Ethics
Mouse work (i.e., collection of fecal samples) carried out at the University of Vienna was in strict accordance with institutional IACUC guidelines, international ARRIVE guidelines, and the principles of the 3Rs of animal research. All animal experiments carried out at the Technical University of Munich, as well as maintenance and breeding of mouse lines, were approved by the Committee on Animal Health Care and Use of the state of Upper Bavaria (Regierung von Oberbayern; AZ ROB-55.2-2532.Vet_02-14-217, AZ ROB-55.2-2532.Vet_02-20-58, AZ ROB-55.2-2532.Vet_02-18-37) and performed in strict compliance with the EEC recommendations for the care and use of laboratory animals (European Communities Council Directive of November 24, 1986 (86/609/EEC)).

### Animals and housing conditions
In-house (at the University of Vienna) bred C57BL/6 J mice were used for data presented in Figs. 1–3. At the University of Vienna mice were group-housed in individually-ventilated cages in a 12-h light/dark cycle with water and food *ad libitum*.

Mice for in vivo experiments (Figs. 4 and 5) were male and housed under specific pathogen-free (SPF) conditions according to the criteria of the Federation for Laboratory Animal Science Associations (FELASA) (12-h light/dark cycles at 24–26 °C) in the mouse facility at the Technical University of Munich (School of Life Sciences Weihenstephan). All mice received a standard diet (autoclaved V1124-300, Ssniff) *ad libitum*, autoclaved water and were sacrificed by $CO_2$ or isoflurane.

Details of the animal models can be found in our previous study[12]. Briefly, Hsp60[flox/flox] mice and Hsp60[flox/flox] x VillinCreER[T2-Tg] mice were generated as described previously[42] to create IEC-specific Hsp60 knockout mice via tamoxifen induction (Hsp60[Δ/ΔIEC]). For conditional Hsp60 deletion, Hsp60[flox/flox] × VillinCreER[T2-Tg] mice and appropriate control mice were kept on phytoestrogen-reduced diet 1005 (V1154-300, Ssniff) for four weeks under SPF conditions. Afterwards, mice received 400 mg tamoxifen citrate per kg chow feed (CreActive T400 (10 mm, Rad), Genobios) *ad libitum* for 7 days. After the induction phase, tamoxifen diet was replaced with the phytoestrogen-reduced diet. During and after the induction phase, mice were monitored daily and aborted when a combined score considering weight loss, changes in stool consistency, general behavior, and general state of health was reached. Animals were sacrificed at the indicated time points. All mice and their respective genotypes were generated and maintained on an in-house crossing of C57Bl/6 N and C57Bl/6 J background.

### Protein extraction and SP3-assisted protein digestion for metaproteomics analysis
The procedures from protein extraction of gut microbiome material to final peptide preparation were performed as previously described[14].

### High pH reversed-phase fractionation of pooled peptides
The peptide fractionation kit was purchased from Fisher Scientific (Cat. 84868). A total of 40 µg pooled fecal peptides were processed according manufactures instruction. Eight peptide factions were dried using vacuum centrifugation and then re-suspended in 30 µL of MS-grade water. The peptide concentration was measured in duplicate using NanoPhotometer N60 (Implen, Munich, Germany) at 205 nm. Peptide samples were acidified with formic acid to a final concentration of 0.1% and were stored at −20 °C until LC-MS/MS analysis.

### Liquid chromatography-mass spectrometry configurations
Nanoflow reversed-phase liquid chromatography (Nano-RPLC) was performed on NanoElute1 and NanoElute2 systems (Bruker Daltonik, Bremen, Germany) coupled with timsTOF Pro and timsTOF Ultra (Bruker Daltonik, Bremen, Germany) via CaptiveSpray ion source, respectively. Mobile solvent A consisted of 100% water containing 0.1% FA and mobile phase B of 100% acetonitrile containing 0.1% FA.

### Data dependent acquisition (DDA-PASEF) of fractionated peptides on timsTOF Ultra and timsTOF Pro
Twenty-nanograms of each peptide fraction (a total of 8 fractions) were loaded on an Aurora™ ULTIMATE column (25 cm × 75 µm) packed with 1.6 µm C18 particles (IonOpticks, Fitzroy, Australia) with a total gradient time of 66 min. The mobile phase B was linearly increased from 5 to 23% in 56 mins with a flowrate of 0.25 µL/min, followed by another linear increase to 35% within 4 min and a steep increase to 90% in 1 min. The mobile phase B was maintained at 90% for the last 5 min with a flowrate increase from 0.25 to 0.35 µL/min. On both timsTOF Ultra and timsTOF Pro, the TIMS analyzer was operated in a 100% duty cycle with equal accumulation and ramp times of 166 ms each. Specifically, 5 PASEF scans were set per acquisition cycle (cycle time of 1.03 s) with ion mobility range from 0.7 to 1.3 (1/k0). The target intensity and intensity threshold were set to 14,000 and 500, respectively. Dynamic exclusion was applied for 0.4 min. Ions with m/z

between 100 and 1700 were recorded in the mass spectrum. Collision energies were dependent on ion mobility values with a linear increase in collision energy from $1/K0 = 0.6$ Vs/cm$^2$ at 20 eV to $1/K0 = 1.6$ Vs/cm$^2$ at 59 eV.

## Data independent acquisition (DIA-PASEF) on timsTOF Ultra

Peptides were loaded onto an Aurora$^{TM}$ ULTIMATE column (25 cm × 75 μm) packed with 1.6 μm C18 particles (IonOpticks, Fitzroy, Australia) with a total gradient time of either 30 min or 66 min on a NanoElute2 system in triplicates. In the 30-min separation, the mobile phase B was linearly increased from 5 to 23% in 18 min with a flowrate of 0.25 μL/min, followed by another linear increase to 35% within 4 min and a steep increase to 90% in 2 min. The mobile phase B was maintained at 90% for the last 4 min with a flowrate increase from 0.25 to 0.35 μL/min. The composition of mobile phase B over the 66-min separation was the same as described above for the fractionated peptide samples. For the results presented in Fig. 2, precursors with m/z between 400 and 1000 were defined in 8 scans (3 quadrupole switches per scan) containing 24 ion mobility steps in an ion mobility range of 0.64–1.45 (1/k0) with fixed isolation window of 25 Th in each step. The acquisition time of each DIA-PASEF scan was set to 100 ms, which led to a total cycle time of around 0.95 s. For results presented in Fig. 3, 25 ng peptides were separated on the NanoElute 2 system with a 30-min gradient. Precursors with m/z between 350 and 800 were defined in 6 scans (3 quadrupole switches per scan) containing 18 ion mobility steps in an ion mobility range of 0.64–1.2 (1/k0) with fixed isolation window of 25 Th in each step. The acquisition time of each DIA-PASEF scan was set to 100 ms, which led to a total cycle time of around 0.74 s. For data presented in Figs. 4 and 5, 50 ng peptides were separated on the NanoElute 2 system with a 66-min gradient. Precursors with m/z between 350 and 1150 were defined in 13 scans containing 32 ion mobility steps in an ion mobility range of 0.65–1.35 (1/k0) with fixed isolation window of 25 Th in each step. The acquisition time of each DIA-PASEF scan was set to 100 ms, which led to a total cycle time of around 1.48 s.

## DDA-PASEF data processing

Fractionated data generated using timsTOF Ultra and timsTOF Pro were separately submitted to MSfragger[43] (version 4.0) integrated in FragPipe computational platform (version 21.1), searching against the MGnify mouse gut protein catalog v1.0 (https://www.ebi.ac.uk/metagenomics/genome-catalogues/mouse-gut-v1-0, referred as PD1). The decoy DB was generated with reversed sequences. Trypsin was specified with a maximum of two missed cleavages allowed. The search included variable modifications of methionine oxidation and N-terminal acetylation and a fixed modification of carbamidomethyl on cysteine. The mass tolerances of 10 and 20 ppm were set for precursor and fragment, respectively. Peptide length was set to 7–50 amino acids with a mass range from 500 to 5000 Da. The remaining parameters were kept as default settings. During the validation, MSBooster (version 1.1.28) was used for rescoring and Percolator[44] (version 3.6.4, default parameters) was used for PSM validation. FDR level was set to 1% for PSM, peptide, and protein. The identified proteins from the search formed a sample-specific protein database (PD2) containing 53,502 protein sequences. For assessing the labeling efficiency of *L. murinus*, the data was searched against the standard proteome of *L. murinus* downloaded from Uniprot (PD3, https://www.uniprot.org/proteomes/UP000051612, accessed on 2023-07-19), containing 1971 protein sequences. The rest parameters were kept the same in MSfragger as aforementioned.

## Bacterial culture of *L. murinus* and *S. ruber*

*Ligillactobacillus murinus* (DSM 20452, *L. murinus*) and *Salinibacter ruber* (DSM 13855, *S. ruber*) were purchased from DSMZ (Braunschweig, Germany). All culture media were autoclaved right after the

preparation. *L. murinus* was activated in 5 mL MRS medium (CARL-ROTH, Karlsruhe, Germany; prepared according to the manufacturer's instructions) and incubated for 24 h at 37 °C with 220 rpm agitation. At the end of this incubation period, 1 mL of the *L. murinus* culture was taken and centrifuged at 3200 × g for 5 min at 4 °C. The supernatant was carefully removed, and the bacterial pellet was gently resuspended in 5 mL SILAC-heavy medium (Glucose 10 g/L, KH$_2$PO$_4$ 3 g/L, K$_2$HPO$_4$ 3 g/L, sodium acetate 5 g/L, ammonium citrate dibasic 1 g/L, MgSO$_4$·7H$_2$O 0.2 g/L, MnSO$_4$·4H$_2$O 0.05 g/L, Tween-80 1 g/L, L-alanine 0.05 g/L, L-arginine-HCl (13C$_6$, 15N$_4$; Fischer Scientific) 0.05 g/L, L-asparagine 0.1 g/L, L-aspartic acid 0.1 g/L, L-cysteine 0.2 g/L, L-glutamine 0.1 g/L, L-glutamic acid 0.1 g/L, glycine 0.05 g/L, L-histidine 0.05 g/L, L-isoleucine 0.05 g/L, L-leucine 0.05 g/L, L-lysine-2HCl (13C$_6$, 15N$_2$; Fischer Scientific) 0.05 g/L, L-methionine 0.05 g/L, L-phenylalanine 0.05 g/L, L-proline 0.05 g/L, L-serine 0.05 g/L, L-threonine 0.05 g/L, L-tryptophan 0.05 g/L, L-tyrosine 0.05 g/L, L-valine 0.05 g/L, uracil 0.01 g/L, guanine 0.01 g/L, adenine 0.01 g/L, xanthine 0.01 g/L, biotin 0.01 g/, Vitamin Solution 2% (v/v)). The heavy-medium culture was incubated at 37 °C with 220 rpm agitation for 24 h. Bacterial growth was monitored with spectrophotometric measurements (Eppendorf, Hamburg, Germany) at an optical density of 600 nm (OD600). An OD600 above 0.8 was aimed to ensure suitable growth conditions. For daily passage, 500 μL of *L. murinus* culture were taken and transferred to another 5 mL SILAC-heavy medium. The labeling efficiency was evaluated on timsTOF Pro after 10 passages in heavy-medium culture. *S. ruber* was cultured in 5 mL DMSZ-936 medium according to the recommendation (https://mediadive.dsmz.de/medium/936) at 37 °C with 220 rpm agitation. The duration between passages for *S. ruber* was around 7 days due to its slow growth. For enlarged culture, 1 mL/each of *L. murinus* and *S. ruber* cultures were transferred to 30 mL medium, respectively. At the end of cultivation, 2 mL bacteria aliquots were made and pelleted at 3200 × g for 5 min at 4 °C, and one of the aliquots was resuspended in 2 mL of either pre-chilled PBS (*L. murinus*) or DSMZ-936 medium (*S. ruber*). The resuspended bacteria were further serial diluted (2–50 times dilution) with either PBS (*L. murinus*) or DSMZ-936 medium (*S. ruber*) for bacteria counting using QUANTOM Tx Microbial Cell Counter (BioCat, Heidelberg, Germany) according to the procedures supplied with the device. The rest of the aliquots were snap-frozen in liquid nitrogen and stored at −80 °C until further use.

## *L. murinus* and *S. ruber* Spike-in experiment

Counted *L. murinus* and *S. ruber* stocks were resuspended and diluted in pre-chilled PBS to reach various numbers (ranging from $1 \times 10^4$ to $1 \times 10^9$) in triplicates. The same number of *L. murinus* and *S. ruber* were mixed with 10 mg of mouse feces and subjected to protein extraction together (as previously described[14]). To ensure a consistent spike-in background, the fecal sample used here was collected and pooled from the same mouse in 2 consecutive days at the same hour. The resulting peptide samples were analyzed on the timsTOF Ultra in a 30-min gradient as described above with 25 ng of peptide per sample. The workflow is illustrated in Fig. 3a.

## Labeling efficiency check for *L. murinus*

The labeling efficiency was checked by analyzing the heavy-labeled culture of *L. murinus* in DDA-PASEF mode, and the data were searched against its reference proteome (PD3, https://www.uniprot.org/proteomes/UP000051612, accessed on 2023-07-19) in Fragpipe with arginine (+10) and lysine (+8) as additional variable modifications. As a result, a total of 60,485 PSMs were identified (1% FDR), corresponding to 12,852 uniquely stripped peptide sequences were identified. In cases where multiple PSMs were assigned to the same peptide, only the most intense PSMs of one peptide was kept for both labeled and non-labeled forms if the latter was co-identified. If the peptide was identified only in either the heavy- or light-labeled form, the missing intensities were

assigned a value of 1 to apply the following formula[45] for each peptide to calculate the labeling efficiency: Peptide labeling efficiency = (Intensity_Heavy / (Intensity_Heavy + Intensity_Light)) x 100. The average of calculated efficiency (97.42%) for all peptides was presented in the study.

## Training of Novor algorithm with PASEF datasets and performance evaluation

In order to obtain a robust tool for de novo sequencing using 4-dimension PASEF data, a custom version of Novor[15] (BPS-Novor) was generated by training Novor's decision tree-based scoring functions on over 1,750,000 PSMs acquired in PASEF mode from a variety of timsTOF instruments. This training dataset included experiments with fixed collision energy measurements of deeply fractionated (a total of 60 high-pH offline fractions) peptide samples from Vero cells and human K562 cells digested with GluC, Pepsin, Elastase, Chymotrypsin, and Trypsin (training data available in PRIDE through identifier PXD051792). The ground truth data was taken from ProLuCID-GPU[46] DB search results filtered with 1% FDR with DTASelect[47] at PSM level.

To evaluate the performance of the newly trained BPS-Novor, a publicly available mixed species (*H. sapiens, Yeast, E. coli*) dataset[48] (ProteomeXchange ID: PXD014777) excluded in the training phase was used to determine the accuracy of the model. In addition, the performance of BPS-Novor was validated against K562 cell lysates digested with non-tryptic enzymes, specifically Elastase, Pepsin, GluC, and Chymotrypsin to ensure accuracy with mimicked non-proteotypic peptides. These samples were analyzed on a 35-min gradient using an EASY-nLC (Thermo Fisher) and a timsTOF Pro instrument. The precision and recall values were calculated as previously described[15].

## Components of novoMP workflow

While the term "uMetaP" refers to the overall ultra-sensitive metaproteomics workflow combining advanced LC-MS acquisition and analysis strategies, novoMP is a dedicated de novo sequencing module that can be integrated within uMetaP or applied independently to suitable datasets. The name "novoMP" represents the complete de novo sequencing-based workflow that incorporates: Step 1: de novo sequencing with BPS-Novor with DDA-PASEF data; Step 2: multi-tier filtering to enhance de novo peptide confidence; Step 3: Creation of microbial protein DB based on de novo peptides; and Step 4: an FDR validation framework using DIA-PASEF.

## de novo sequencing of DDA-PASEF data

Fractionated data were submitted to BPS-Novor integrated in ProteoScape (Bruker Daltonik, Bremen, Germany) for de novo sequencing. The mass tolerances for precursors and fragments were set to 20 ppm and 0.02 Da, respectively. Tryptic peptides with a maximum of two missed cleavages were allowed. Carbamidomethyl was set as a fixed modification on cysteine, and methionine oxidation and N-terminal acetylation were set as variable modifications. A maximum of two variable modifications per peptide was allowed. In addition, only the top candidate sequence per spectrum was exported in the output.

## Multi-tier filtering of de novo sequenced PSMs

The de novo sequencing outputs were imported into R and subjected to the following six filters sequentially. (1) De novo score: The first filter was based on the BPS-Novor software, applying a score threshold of 65. (2) Charge state: We excluded PSMs with a charge state of 1 due to their less reliable fragmentation patterns. (3) Peptide length: we removed peptides shorter than seven amino acids to reduce the risk of ambiguous matches. (4) Mass error: We evaluated the mass error of sequenced precursors and retained only 95% of the sequenced PSMs that fell within the upper and lower cut-offs calculated using qnorm function in R based on the mass error distribution. (5) Retention time

shift: Retention time predictions were performed using DeepLC[49] (version 2.2.27). We retained 95% of the remaining PSMs, which showed a strong correlation between observed and predicted retention times, based on the upper and lower cutoffs calculated using the qnorm function in R. 6) Collisional cross-section (CCS) shift: CCS predictions were performed using IM2Deep[50] (version 0.1.7). We retained 95% of the remaining PSMs that showed a strong correlation between measured and predicted CCS values, using cutoffs calculated as described above.

## Blast homology search of de novo sequenced peptides for the construction of microbial protein database

Unique peptides remaining after multi-tier filtering were subjected to a BLAST+ homology search[51] to retrieve potential protein sequences for microbial protein DB construction. The blastp function embedded in Diamond[52] (version 2.1.9; command line) was used to search against the non-redundant protein sequence DB "nr.gz" (ftp://ftp.ncbi.nlm.nih.gov/blast, updated 2024-02-27). The search of de novo sequenced peptides in ultra-sensitive mode was restricted to the following taxa due to the nature of our samples: bacteria (taxaID: 2), fungi (taxaID: 4751), archaea (taxaID: 2157), and viruses (taxaID: 10239). All BLAST searches used the PAM30 scoring matrix. The top 5 protein assignments per query sequence were listed in the output file (output format: 6). In addition, another search with same parameters but different output format (output format: 102) was performed to generate taxonomic classifications of sequenced peptides based on the lowest common ancestor (LCA) algorithm. To select the only one protein assignment per query sequence among the top 5 candidates, we used LCA-guided procedure. Specifically, if the taxonomic annotation of one protein candidate matches exactly the taxonomy assignment in the LCA output, then this candidate is kept. In the case that the taxonomic annotations of the protein candidates do not match exactly to the LCA output but belong to taxon rank in the LCA output, these candidates were kept. Finally, if the above two steps did not generate one protein per query sequence, the blast parameters (Bitscore, pident and e-value) will be applied to keep the most confident candidates. To further increase the quality of the blast search result, we applied a minimum of 80% cut-off for sequence identity as previously reported[53,54], then further retrieved the protein sequences from NCBI using the protein sequence IDs in the blast output to form a microbial DB based on novoMP (novoMP-DB; PD4). As a comparison, peptides identified using the aforementioned MSFragger search were subjected to the same blast homology search referred as DB-search (PD5) in the manuscript.

## DIA-PASEF data processing

DIA-NN[55] (version 1.9) was used to process DIA-PASEF data in library-free mode to generate the predicted spectrum library. A deep learning-based method was used to predict theoretical peptide spectra along with their retention time and ion mobility. Trypsin/P was used for in silico digestion with an allowance of a maximum of 2 missed cleavages. Variable modifications on peptides were set to N-term methionine excision, methionine oxidation and N-terminal acetylation, while carbamidomethylation on cysteine was a fixed modification. The maximum number of variable modifications on a peptide was set to 2. Peptide length for the search ranged from 7 to 30 amino acids. The m/z ranges were specified accordingly depending on the experiment which aligned with the DIA-PASEF acquisition method, and fragment ions were set to a range from 100 to 1700. Mass accuracy for both MS1 and MS2 was set to automatic determination. Protein inference was set to "Protein names (from FASTA)" and the option of "Heuristic protein inference" was unchecked. Match-between-run (MBR) was enabled for cross-run analysis. RT-dependent cross-run normalization and QuantUMS[56] (high precision) options were selected for quantification. Generally, all searches in DIA-NN included a *Mus musculus* reference proteome (https://www.uniprot.org/proteomes/UP000000589,

accessed on 2023.04.07) together with different microbial DBs. Specifically, results presented in Figs. 2 and 4 were searched against PD2, PD4, and PD5 (de-duplicated). Data shown in Fig. 3 were searched against PD2, PD4, PD5, as well as the standard proteome of *L. murinus* (PD3) and *S. ruber* (PD6; https://www.uniprot.org/proteomes/UP000008674, accessed on 2023-07-19*)*. In addition to the searching parameters mentioned above, heavy isotopic labeling of arginine (+ 10.0082699 Da) and lysine (+8.014199 Da) were set as variable modifications.

The DIA-NN search outputs were further processed with the R package, DiaNN (https://github.com/vdemichev/diann-rpackage), to calculate the MaxLFQ[57] quantitative intensities for all identified peptides and protein groups with *q*-value ≤ 0.01 as criteria at precursor and protein group levels.

### DIA-PASEF spectrum visualization
Skyline[58] (version 23.1.0.380) was used to visualize the spectra of peptides identified by DIA-NN. Briefly, the spectral library generated by DIA-NN after DB searching was imported into Skyline to construct a library containing precursor information for the detected peptides. Precursors listed in the library and their associated fragment ions were then extracted from the raw DIA-PASEF data. During extraction, mass accuracy was set to 10 ppm for both precursors and fragments. To minimize false matches, only scans within 5 min of the retention times listed in the library were extracted.

### Taxonomic and functional annotation and quantification
MetaLab[59] (version 2.3.0) was used for taxonomic annotation. Peptide sequences and their corresponding intensity data were imported into MetaLab, and the built-in taxonomy DB was used for mapping, with blanks ignored below the rank of Superkingdom and a minimum unique peptide count of three required. For the quantification of specific taxonomic ranks[14], the annotation output was processed in R to extract peptides commonly detected across samples for taxonomic rank of interest (e.g., genus and species). The intensity of each taxon was calculated by summing up the intensities of common peptides in each sample. The resulting summed intensities were log2-transformed for statistical analysis.

The microbial protein DBs used in this manuscript were annotated using EggNOG-mapper[60] (http://eggnog-mapper.embl.de/) with default settings to retrieve potential functions and pathways. In addition, all quantified microbial protein sequences were aligned against the COG DB (ftp://ftp.ncbi.nih.gov/pub/COG/COG2024/data) using Diamond[52]. For each query, the best hit was selected for annotation. The corresponding COG ID, name, EC number, and associated functional information were extracted from the COG DB annotation file. To estimate the abundance of each COG ID in a given sample, the MaxLFQ intensities of all proteins annotated with the same COG ID were summed.

### Taxon-specific functions analysis
Meta4P[61] was used to analyze taxon-specific functions. The peptide quantification data from DIA-NN, taxonomic annotation output from MetaLab, and functional annotation files from EggNOG-mapper were used as inputs for Meta4P. Quantification of taxon-specific functions was performed by summing the peptide intensities associated with specific functions. The resulting summed intensities were log2-transformed for statistical analysis.

### Identification filters of *L. murinus* and *S. ruber* for spike-in experiment
Peptide and protein identifications generated from DIA-NN search of the spike-in experiment were further filtered to ensure species-specific identifications: (1) Only heavy-labeled peptides were considered for *L. murinus* to exclude the interference from endogenous species. Heavy-labeled peptides assigned to *S. ruber* were removed

as they represent false-matches. (2) Co-assigned peptides and protein groups shared between *L. murinus* and *S. ruber* were excluded. (3) Peptides assigned to *L. murinus* or *S. ruber* that were also identified in any of the non-spike controls (three replicates) were removed.

### Functional enrichment analysis of differentially expressed host proteins
Quantified host proteins were statistically compared in R using the ProTIGY package (https://github.com/broadinstitute/protigy) with a two-sample moderated t-test. Functional enrichment of differentially expressed host proteins was performed using the clusterProfiler[62] R package, with all identified proteins in the study as background genes for enrichment analysis against the Gene-Ontology Biological Process DB. The Benjamini-Hochberg method was used to adjust *p* values, with an adjusted *p*-value cutoff of 0.05 used to identify significantly enriched pathways.

Protein-protein interaction networks were analyzed using STRING within Cytoscape (version 3.10.2) under default parameters. Drug-gene interactions were retrieved using DGIdb[63] (version 5.0.7) with default settings.

### Statistical analysis
The Kruskal-Wallis test was performed in R to identify significant differences in genera among conditions. Differentially expressed species and taxon-specific functions were analyzed using the limma package (moderated two-sided *t*-test) in R for the respective comparisons. The Benjamini–Hochberg method was applied for multiple comparisons in all statistical analyses.

### Reporting summary
Further information on research design is available in the Nature Portfolio Reporting Summary linked to this article.

## Data availability
The mass spectrometry proteomics data have been deposited to the ProteomeXchange Consortium via the PRIDE partner repository with the dataset identifier PXD051792. Source data are provided with this paper.

## Code availability
The custom codes generated in the manuscript have been deposited to GitHub (https://github.com/CoEMetaproteomics/DeNovo/tree/main/NovoMP).

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

## Acknowledgements

We thank Elisabeth Clifford (Division of Pharmacology & Toxicology, University of Vienna, Austria) for assistance during sample preparation. We thank Astrid Horn, Ph.D. (Centre for Microbiology and Environmental Systems Science, Division of Microbial Ecology, Austria) for suggestions regarding bacterial counting. We thank Vadim Demichev for the discussions related to DIA-NN analysis. We thank Biognosys AG for the 1-month free license of Spectronaut. This research was funded in whole or in part by the University of Vienna and by the Austrian Science Fund (FWF; 10.55776/P35856 and 10.55776/P36554 to M.S.). E.U., D.A., and D.H. were funded by Deutsche Forschungsgemeinschaft (DFG; 395357507 and 469152594). For open access purposes, the author has applied a CC BY public copyright license to any author accepted manuscript version arising from this submission. The computational results presented have been achieved in part using the Vienna Scientific Cluster (VSC).

## Author contributions

Conceptualization: D.G.V.; Experimental design: D.G.V., F.X., D.A., E.U., and D.H.; Biochemistry and mass spectrometry: F.X. and C.K.; Sample collection and preparation: F.X., M.B., R.K.R.K., E.U., and D.A.; Data analysis: F.X., D.G.V., J.K., T.S., Q.L., A.M.B., and B.M.; Writing: D.G.V., F.X., D.A. All authors edited and approved the final manuscript; study supervision: D.G.V.; project administration: M.S. and D.G.V.; Funding acquisition: M.S. and D.G.V.

## Competing interests

M.S. received research awards and travel support from the German Pain Society (DGSS) both of which were sponsored by Astellas Pharma GmbH (Germany). M.S. received research awards from the Austrian Pain Society. M.S. received a one-time consulting honorarium from Grunenthal GmbH (Germany). None of these sources influenced the content of this study, and M.S. declares no conflict of interest. D.G.V. and M.S. have an ongoing scientific collaboration with Bruker (Center of Excellence for Metaproteomics, University of Vienna—Bruker). F.X., M.B., E.U., R.K.R.K., A.M.B., D.A., D.H., M.S., and D.G.V. declare no competing interests. C.K., J.K., and T.S. are employees of Bruker Daltonics GmbH & Co., and Q.L. and B.M. are employees of Rapid Novor.
