## [Transparent Peer Review file · Nature Communications]

Ultra-sensitive metaproteomics redefines the “dark metaproteome”, uncovering host-microbiome interactions and drug targets in intestinal diseases

Corresponding Author: Dr David Gomez-Varela

A version of this paper was originally rejected for publication by Nature Communications, however that decision was reconsidered after appeal by the authors.

Version 0:

Reviewer comments:

Reviewer #1

(Remarks to the Author)

Xian et al have submitted a manuscript on a metaproteomics workflow based on timsTOF Ultra mass spectrometer and data-independent acquisition analysis. The authors claim to identify ca. 48,000 protein groups from ca. 200 bacterial species in the mouse gut microbiome, reaching the taxonomic coverage similar to 16S sequencing. They spike SILAC-labeled bacterial proteomes to assess the limits of detection and quantification of bacterial proteins in the gut microbiome and demonstrate detection of proteins from 500 femtogram of sample (corresponding to protein mass of a single bacterium). The manuscript is quite descriptive and will be of interest to the metaproteomics community; however, it lacks the novelty and application to a “real-life” biological question that would merit its publication in Nature Communications.

Specific comments:

1. The authors use a combination of state-of-the-art workflows/technologies, such as HpH fractionation in combination with UHPLC, timsTOF Ultra and DIA analysis. They recently published almost identical workflow in *Frontiers in Microbiology*, without HpH and on a timsTOF Pro MS. Although they (and the field) will profit from the new generation of MS instruments, I do not see enough novelty to call their study a “quantum leap”. The increased sensitivity (<500 fmol) can also be attributed to the new MS, rather than to any specific development in their workflow.

2. While I will not challenge the increased sensitivity and number of detected protein groups by DDA, I have a general problem with DIA analysis, which is the cornerstone of their workflow. DIA-NN is known to have quite “loose” treatment of FDRs and protein inference compared to other DIA-platforms. In combination with noisy spectra and huge databases, this can lead to dramatic accumulation of false positives. Of note, it would be interesting to know how many identification were filtered out falsely assigned SILAC label.

3. Also, the protein inference problem will lead to attribution of one peptide sequence to several protein entries in the database; if not properly grouped, this will lead to an artificial increase in the number of detected proteins (and taxonomic phyla, of course). It is visible in the Methods section that the authors did not use the default inference settings of DIA-NN, but it is not clear what was the reason and what influence this had on the dataset. Limited sequence coverage in metaproteomics measurements is indeed a problem, but increasing false positive rates or tempering with protein inference is certainly not a solution. For any future submission to a more specialized journal, I would advise the authors to benchmark their DIA approach against other commonly used DIA platforms (especially Spectronaut) and include positive and negative controls in their samples and databases to calculate the real FDRs of their measurements.

Reviewer #2

(Remarks to the Author)

Reviewer #3

(Remarks to the Author)

Xian and colleagues have described an ultra-sensitive metaproteomic approach that enables single-organism proteomics and captures taxonomic profiling at a greater depth than previously achieved. While this approach represents a significant advancement over prior strategies, several claims appear overstated or not directly supported by the data presented. For example, the authors compare the depth of their proteomic approach to 16S strategies but did not characterize the microbiome of their samples using the 16S method for a direct comparison. Instead, they compared their proteomic results to published 16S studies conducted on different samples questioning the validity of this comparison.

Other comments:

Lines 49 to 51: The process of pooling peptide samples from previously prepared samples is described.

L. Murinus should be L. murinus and S. Ruber should be S. ruber.

Line 302: The authors claim that uMetaP achieved notable improvements across all taxonomic levels (a three-fold increase at the species level). However, it is unclear what these improvements are and which data support this statement. Additionally, the authors state that this taxonomic coverage sets new standards in metaproteomics by matching the average performance offered by full-length 16S rRNA of the human gut microbiome. What do they mean by this? It would seem necessary for the authors to characterize the gut microbiome of their own samples using full-length 16S rRNA to substantiate this claim.

Lines 307 to 309: The authors state that they identified three species with an abundance of 10 pg. How was this measured?

Does this approach enable absolute quantification of the bacterial species present in the microbiome? Is there a direct correlation between the peptide intensities and the abundances of S. ruber and L. murinus?

Line 413: The authors claim that, on the taxonomic front, uMetaP achieves significantly deeper analysis, surpassing the coverage of commonly used 16S rRNA. Which data support this statement? This evidence is not presented in the paper. Again, a direct comparison would likely be necessary.

Reviewer #4

(Remarks to the Author)

Xian et al. present a metaproteomic analysis of a mouse microbiome sample utilizing the latest timsTOF Ultra mass spectrometer, referred to as uMetaP. The new instrument enabled them to measure over 100K peptides in a 30-minute MS run and over 120K peptides and approximately 50K protein groups in a 66-minute run, marking one of the most extensive single-shot metaproteomic studies to date. uMetaP demonstrated ultra-high sensitivity, with a coefficient of variation (CV) of less than 20% for a 10pg peptide injection and less than 10% for a 5ng peptide injection (Figure 2). Additionally, through a heavy-labeled spike-in experiment and a two-species mix sample analysis, they showcased a remarkable lower limit of detection and quantitation (LLOQ of 1e-3%) and achieved potential single-bacterium resolution with peptide injections as low as 500fg.

Overall, this study is technically robust, comprehensive, and significantly advances the field of metaproteomics, illustrating the instrument's applicability in metaproteomic analyses. However, my main concern with the current manuscript is the overly strong and somewhat bold claims and interpretations of their results.

The authors have termed their solution as uMetaP, which might lead readers to believe that this study introduces a new strategy for metaproteomics. However, I did not observe a novel workflow for metaproteomics in the manuscript beyond the utilization of a new timsTOF Ultra mass spectrometer, when compared to their recently published work with an earlier version timsTOF Pro, as well as other previous metaproteomic studies. The depth of metaproteomic analysis achieved in this study is comparable to a recently published study using Orbitrap Astral mass spectrometer.

Furthermore, the authors frequently use terms such as "unprecedented," "quantum leap," and "unparalleled" to describe the identification and sensitivity achieved. While these descriptions may be accurate in comparison to their previous study with timsTOF Pro, they do not hold true when considering more recent advancements in the field (such as the Orbitrap Astral work mentioned above).

The idea of achieving single bacterium proteomics (SBP) is ambitious. In this study, the authors used a two-bacteria mix and tested the peptide injection amount (500fg) for MS analysis to demonstrate single bacterium resolution. However, this approach falls short of true SBP, which would require starting from an individual bacterium for sample processing, rather than using a peptide injection amount equivalent to that of a single bacterium. Additionally, based on the results presented, SBP is unlikely to be applied to microbiomes at this moment.

Other comments:

1. Line 73: why timsTOF Pro, not Ultra, was used to evaluate the labeling efficiency? Details are needed regarding the database search and the method used to calculate labeling efficiency. Additionally, some bacteria can synthesize amino

- acids from others, such as Arginine; did the authors examine heavy labeling in amino acids other than K/R?
2. Line 100: twenty-microgram or twenty nanogram?
 3. Line 261: this is incorrect; there are several DDA-PASEF metaproteomic data published recently with more identified proteins (doi.org/10.1101/2023.11.29.569331);
 4. Line 262-264: Why did the authors use in-silico digestion of the identified proteins, rather than the actually identified peptide sequences, to infer taxonomic coverage?
 5. Line 306-307: I think this comparison is not fair; the identified 220 species in this study are from all four kingdoms of microorganisms, whereas the 16S rRNA sequencing in the cited study focuses only on bacteria. How many bacterial species were identified in this study, and how does this compare to the 16S sequencing results? Ideally, a matched sample sequencing would be needed to directly compare metaproteomics and 16S sequencing.
 6. Line 316-319: The authors highlight the hidden functions (PUFs, sProt and AMPs) throughout the manuscript from the title to introduction and discussion, readers would expect a more detailed analysis of these portions of information, rather than just the numbers of identifications;
 7. Line 328: is this really the first time in metaproteomics? In ref#6 of the manuscript, Duan et al used a heavy labeled E. coli to assess the limit of detection in metaproteomics as well.
 8. Line 361: Is it really 5000 times greater compared to reference #6? Duan reported a limit of 0.5% with their method, while here it is reported as 1e-3 %; shouldn't this be 500 times instead of 5000 times? Maybe I missed something here. More details on how the comparison was made are needed to provide an accurate and fair representation of the reported increase.
 9. Line 402: unassigned spectral data; to my understanding, the information in these data is truly hidden function or dark matter (instead of those identified proteins without known functions).
 10. Line 406-407: Does the identification in this study surpass the previous study with Astral? In the Astral study, ~122K peptides and 38K protein groups were identified with a 30min MS run, while this study reports about 102K peptides with a 30min MS run (Line 278).
 11. Line 408: to be applicable for large scale metaproteomic study, the instrument needs to be robust with stable performance. Could the authors comment on this as well when comparing to other instruments?

Version 1:

Reviewer comments:

Reviewer #3

(Remarks to the Author)

The authors have adequately addressed my previous concerns. However, the newly added section on defining the druggable gut metaproteome raises additional questions. While the concept of a “druggable metaproteome” is intriguing, this section appears to fall outside the microbiome-focused scope of the study, as it primarily characterizes the host proteome (“druggable proteome”) rather than the metaproteome.

Several aspects require clarification: (1) How novel is this approach in the context of studying the druggable proteome? (2) What advantages does the authors' methodology offer over existing approaches for analyzing the host proteome? (3) Many of the identified proteins have already been reported as being affected in IBD. What new insights does this study provide beyond existing knowledge? Addressing these points would strengthen the impact and relevance of this section.

Reviewer #4

(Remarks to the Author)

The authors have made substantial revisions to their manuscript, including incorporating a de novo peptide identification strategy for database construction and providing a concrete example to illustrate the applicability of their workflow. Overall, they have addressed most of the reviewers' comments. However, I do have some comments regarding the newly added de novo sequencing section, along with a few additional comments for the authors' consideration.

The reviewer appreciates the use of a de novo sequencing strategy, enhancing the comprehensiveness of the database used for metaproteomic identification. This authors stated that 1.75 million PSMs from PASEF data were used to train the Novor model. However, there is no detailed information on these 1.75 million PSMs, nor on how they are specific to metaproteomics. Previous Bruker public releases and application notes (e.g., for immunopeptidomics) have indicated that BPS-Novor was trained with 1.75 million PSMs. Were the same datasets used in this study? If so, were there any specific adaptations or optimizations for metaproteomics? If such optimization were made, more details are needed for this manuscript, particularly since the authors have renamed the approach as NovoMP. If no specific optimizations were performed, I suggest keeping the name BPS-Novor to avoid potential misunderstandings.

Line 96: the authors indicated the identification of 551 additional species using NovoMP-DB. Could the authors include a figure panel to illustrate these species and discuss their previous evidence of presence in mouse/human gut? A Venn diagram comparing species/taxa identified by NovoMP and DB-search would also be informative.

An 80% sequence identity for blastp has been used in this study. Could the authors justify this choice? Given that peptide sequences identified using bottom-up proteomics are relatively short (7–30 amino acids or 21–90 nucleotides), whereas classical BLAST searches typically use sequences >100 nucleotides, was any evaluation performed to determine the suitability of the 80% identity threshold in this context?

In general, I would be very careful to make claims that metaproteomics achieves higher coverage than 16S or metagenomic sequencing. Specifically, in Figure R9/S4B, the large differences in the number of identified taxa and the low number for shotgun sequencing is surprising (even given the fact of shallow shotgun sequencing). For metaproteomics, this study uses a threshold of three unique peptides for taxon identification, but was this criterion validated for this particular dataset? A statistical strategy is needed to justify the use of 3 unique peptides for confidently identifying a species, in particular when the authors make claims for taxonomic coverage in this study.

Figure 1A: why there is no Pro-PSM?

Figure 2E: could the authors explain what are the remaining 264 species (631- 253- 114 =264)? The figure is somewhat unclear, and additional explanation would help improve its interpretation.

Line 170: is this based on enrichment analysis? What are the statistical criteria used for this 'detailed functional analysis'?

Reviewer #5

(Remarks to the Author)

Many of the issues highlighted by the previous reviewers remain unresolved. Here is a few examples:

Supplementary Table 1: The data shows that 774 species are identified with 3 or more peptides. However, this number drops significantly to 345 species with at least 5 peptides and only 131 species with 10 or more peptides. Relying on 3 peptides to identify a bacterial species is not appropriate, as the false discovery rate (FDR) would likely be high, given the total number of peptides identified. This raises concerns about the accuracy of species identification, particularly when comparing metaproteomics to 16S rRNA sequencing and metagenomics. The latter two methods are inherently more accurate for species identification.

Furthermore, Supplementary Figure 4B is problematic. The expected trend should show metagenomics detecting more species than 16S rRNA sequencing, which in turn should detect more species than metaproteomics. The analysis should be done using deep sequencing metagenomics to accurately determine the species present and validate the findings.

Table 2 also presents significant issues, particularly regarding the claim that the method can detect species down to 10 pg. Of the peptides identified in the microbiome at the 10 pg level, 56 are not detected in all of the 100 ng experiments. Among these, 29 are entirely absent in the 100 ng experiments, and 11 are detected in only one experiment. These peptides are very likely to be false positives. As well, the peptide LSAEFGSLR is reported at one-third the intensity in the 10 pg sample compared to the 100 ng sample, which is not possible.

Additionally, I performed a BLAST search on a few de novo sequenced peptides claimed to be of bacterial origin. These peptides show strong homology to mouse and human proteins. This raises the question of whether these peptides are truly of bacterial origin or if they are contaminants from human/mouse samples or misassigned to bacteria. This issue further undermines the reliability of the species identification.

Version 2:

Reviewer comments:

Reviewer #3

(Remarks to the Author)

The authors have addressed my previous concerns. I have no additional comments.

Reviewer #4

(Remarks to the Author)

I would like to thank the authors for taking another significant effort to address the comments raised by myself and other reviewers. It was encouraging to see that the authors removed the comparison of taxonomic coverage between metaproteomics and metagenomics/16S data. This was one of the most concerning aspects of the previous version of the manuscript, and its removal addresses several of the concerns raised by both myself and Reviewer #5. For the justification of using 3 unique peptides for species identification, I am not convinced by the authors. The way for the calculation of the FDR and F-values seems to be very questionable. Both MetaLab or Unipept assign taxonomic annotations based on exact match of identified peptide sequences to the sequences in their databases; it would be very surprising that the decoy peptide sequences generated by the authors could have a lot of matches in the databases. As a result, the 'false positives (FP)' would be very low, leading to very low FDR values across all levels. In my opinion, the unique peptide number threshold selection might be highly relevant to the depth of identification in metaproteomics. In particular in this ultra-deep dataset using DIA, the accuracy of species identification could be likely impacted/compounded by the potential lower

confidence of peptide identification as well. As said this is a known challenge of the field, not limited to this work itself. However it would be necessary to discuss this limitation particularly when the authors make conclusions/statements on the significantly increased coverage of taxonomic identification.

A similar point applies to the analysis based on 10 pg input. The authors are correct in noting that the proteomic analysis of extremely low input samples is a known challenge in the field. This limitation should be stated in the manuscript along with the discussion of improved sensitivity. The analysis exploring missing peptide identifications in high-input datasets is insightful and helps address some of these concerns. I would recommend that this part of the analysis be included in the supplementary information and discussed in the main text when addressing the limitations and interpreting the results.

Line 44: uMetaP enables novoMP? Shouldn't this be novoMP enables uMetaP? There should be a clear definition of uMetaP, novoMP as well as an explanation of their relationship in the manuscript.

Supplementary Figure 1b-e: please make sure these data has not been previously published; otherwise these need to be removed or clearly indicate their prior publication or public release with links/references.

General response

We deeply appreciate the time and effort you have invested in reviewing our manuscript. Your insightful feedback has been instrumental in improving our study and clarifying key findings. In response to your comments, we have revised our manuscript and summarized the key changes and improvements below:

- To address the “*lack of novelty*”:
 - We have developed a *de novo* solution (termed novoMP) for the high-confidence construction of metaproteomic databases. It is composed of several new tools:
 - the first *de novo* algorithm trained in the 4D PASEF data structure (using more than 1,7 million PSMs; in collaboration with Bruker and RapidNovor teams).
 - a machine-learning guided multi-factorial filtering step to increase the confidence of *de novo* peptides.
 - an orthogonal FDR control method using DIA-PASEF, which demonstrates equal confidence for *de novo* and classical database-discovered peptides.

We characterised the benefits of novoMP for increasing the taxonomic and functional landscape reached by uMetaP.

- To address the “*lack of application to a real-life biological question*” :
 - We have performed *in vivo* experiments using a conditional transgenic mouse of colonic injury (in collaboration with Professor Dirk Haller’s laboratory). Our data demonstrated how uMetaP matched and even extended the taxonomic changes previously reported using genomics (doi.org/10.1016/j.chom.2024.06.013). Remarkably, uMetaP detected dysbiotic changes 8 days earlier than genomic methods, demonstrating the increased sensitivity, and found extended changes in the bacterial species altered during tissue injury. Moreover, we added a previously unknown functional angle to this injury model, showing injury-specific changes in both microbiota and the host.
 - We orthogonally validated our data using transcriptomic data from biopsies of 204 Crohn's patients.
We explore the translational potential of our findings by defining, for the first time, the concept of “druggable metaproteome”. Among the 204 drug-protein interactions discovered are treatments for intestinal inflammatory diseases (e.g., Natalizumab), showcasing uMetaP's potential for disease signature discovery and data-driven drug repurposing strategies.
- In addition, we have addressed all technical/methodological questions with a set of new experiments (see Figure R1-12 in this document) presenting new insights regarding DIA-based metaproteomic data analysis. Below is a summary of the key points raised by the reviewers:
 - We have conducted several FDR entrapment experiments that demonstrate the similar confidence of DIA-NN compared to Spectronaut v19 (in

collaboration with both, Prof. Vadim Demichev, DIA-NN developer, and the Spectronaut team).

- We have characterized the effect of inference settings in DIA-NN (in collaboration with Prof. Vadim Demichev).
- We have performed new 16S RNA and metaproteomic sequencing experiments and data analysis demonstrating the bigger taxonomic depth reached by uMetaP compared to both, 16S rRNA in mouse feces and shallow shotgun metagenomic sequencing in mouse colonic content.

The details of the above-mentioned experiments, along with other efforts to address all the questions raised by the reviewers and the resulting changes to the manuscript's content, are provided in the following "point-to-point response".

Point-to-point response

Reviewer #1:

Comment 1: Xian *et al.* have submitted a manuscript on a metaproteomics workflow based on timsTOF Ultra mass spectrometer and data-independent acquisition analysis. The authors claim to identify ca. 48,000 protein groups from ca. 200 bacterial species in the mouse gut microbiome, reaching the taxonomic coverage similar to 16S sequencing. They spike SILAC-labeled bacterial proteomes to assess the limits of detection and quantification of bacterial proteins in the gut microbiome and demonstrate detection of proteins from 500 femtogram of sample (corresponding to protein mass of a single bacterium). The manuscript is quite descriptive and will be of interest to the metaproteomics community; however, it lacks the novelty and application to a "real-life" biological question that would merit its publication in Nature Communications.

We acknowledge the Reviewer's concerns and have performed new experiments to address the "lack of novelty" and the lack of "application to a real-life biological question".

- In regards to the "lack of novelty":

As shown in our new Figure 1A, more than 70% of recorded precursors did not result in peptide-spectrum-matches (PSMs). Therefore, classical database search-based workflows do not extract the vast majority of biological information enabled by the sequencing power offered by uMetaP. For this reason, we have developed, tested, and validated a novel *de novo* strategy, termed novoMP, tailored for metaproteomics DB construction.

Compared to previous studies (doi.org/10.26434/chemrxiv-2024-4v6q0), novoMP is unique in three key aspects:

- It is the first *de novo* algorithm trained on the PASEF data structure (in collaboration with RapidNovor and Bruker). We trained Novor with more than

1,7 million PSMs obtained from various timsTOF platforms, different species, and using different cleavage enzymes (Figure 1B and Supp. Figure 1C-1E).

- It implements a new multi-layered quality control pipeline to select high-confidence *de novo* PSMs (Figure 1 and Supp. Figure 1F-1K).
- It implements a novel orthogonal FDR validation method using DIA-PASEF. We demonstrate equivalent confidence in novoMP-DB peptides compared to peptides obtained through classical database search workflows (Figure 2C and Supp. Figure 2B).

Our new results show the clear advantages of novoMP for:

- Improving the construction of metaproteomic databases by increasing microbial database coverage specifically for archaea, fungi and viruses (Figure 1D-1E and Supp. Figure 1L-1M).
- Increased taxa annotation confidence by adding more taxa-specific peptides compared to classic database search (Figure 1F-1G).
- Increased metaproteome profiling depth without sacrificing the quality using uMetaP powered by DIA-PASEF analysis (Figure 2C and Supp. Figure 2B).
- The increased functional coverage (Figure 2F) and the number of species-specific KEGG pathways (Figure 2G and Supp. Figure 2D).

We provide the metaproteomic community with a roadmap for increasing confidence in *de novo* solutions. On the one side, our multi-tier filtering system can be leveraged to any *de novo* algorithm of choice. Furthermore, we show the applicability of novoMP to previously published DDA-PASEF datasets.

- In regards to the “lack of application to a real-life biological question”:

Our recent collaboration with Prof. Dirk Haller’s laboratory (doi.org/10.1016/j.chom.2024.06.013) defined the concept of metabolic injury by investigating the role of mitochondrial (MT) function in epithelial stem cell homeostasis and gut health. Transient and conditional deletion of the MT chaperone heat shock protein 60 (Hsp60) in intestinal epithelial cells (IECs) in mice, triggered temporary mitochondrial dysfunction, leading to metabolic stress and transient structural changes in the colonic epithelium. Shotgun metagenomic detected taxonomic changes led by a significant increase of *Bacteroides caecimuris*. RNA sequencing from mouse colons and Chron’s patient’s biopsies determine transcriptional signatures of disease relapse/remission. Mono-colonization of germ-free mice with *Bacteroides caecimuris* replicate the disease phenotype, validating the importance of this bacteria species for tissue injury upon MT dysfunction. However, how metabolic changes in the intestinal epithelium select the growth of certain bacteria and how specific bacteria interfere with epithelial regeneration are unknown, which precludes a functional understanding of the disease and the exploration of potential therapeutical targets.

We tested uMetaP by analyzing the colonic contents from control (Hsp60^{fl/fl}) and injured (Hsp60^{Δ/ΔIEC}) mice at two time points, the beginning (D0) and the peak of the

injury (D8). The result has been implemented as new Figure 4-5 and Supp. Figure 4-5 in the revised manuscript, and here we provided a summary of the most important results:

- uMetaP mirrored (e.g., β -diversity, an increase of *Bacteroides caecimuris*) and extended (previously unreported changes in several genera and species) the metagenomic taxonomic findings on these same samples.
- uMetaP offered greater depth and temporal sensitivity than metagenomics and immunohistochemistry methods, allowing earlier detection of taxonomic alterations underlying causal disease mechanisms. On the one side, we extended the previously known dysbiosis signature by detecting significant abundance changes in additional bacterial species. On the other side and similarly to metagenomics, our data showed an increase for *Bacteroides caecimuris* at day 8. Notably, uMetaP also detected an increase in *Bacteroides caecimuris* during the first 24 hours after tamoxifen cessation.
- From the microbiome functional perspective, our data provide plausible mechanisms by detailing metabolic reprogramming during disease progression *in vivo*. We identified *Bacteroides caecimuris* as the bacterial species with the most KEGG pathways altered at day 8. Notably, two altered KEGG pathways were unique to this species: carbon fixation by the Calvin cycle and biosynthesis of ansamycins. In the Discussion section, we hypothesize about the benefits of these two metabolic changes for providing a competitive advantage in the gut microbiome for *Bacteroides caecimuris*, especially in the context of Hsp60 deletion in IECs.
- We validated our findings using two ortholog datasets: RNA-targeted sequencing on colon tissue of the injured and control mice and RNA sequencing from more than 340 colon biopsies taken to 204 Chron's patients (Figure 5C).
- We extend the validity of uMetaP in a "real life biological question" by introducing and testing the concept of "druggable metaproteome". An integrative analysis pipeline validated host proteins significantly changed upon tissue injury as targetable nodes using approved drugs. Significantly, this approach rendered drugs currently used for the treatment of IBD disorders (Natalizumab[®], targeting ITGB2 for the treatment of Crohn's disease), as well as other approved drugs used for the treatment of several inflammatory and immune disorders (Figure 5E).

Comment 2: The authors use a combination of state-of-the-art workflows/technologies, such as HpH fractionation in combination with UHPLC, timsTOF Ultra and DIA analysis. They recently published almost identical workflow in *Frontiers in Microbiology*, without HpH and on a timsTOF Pro MS. Although they (and the field) will profit from the new generation of MS instruments, I do not see enough novelty to call their study a "quantum leap". The increased sensitivity (<500 fmol) can also be attributed to the new MS, rather than to any specific development in their workflow.

We truly thank the reviewer for encouraging us to increase the novelty and quality of our workflow.

We now offer the development and application of novoMP, improvements in depth, sensitivity, and accuracy, the benchmark on a meaningful *in vivo* disease model, and

the ability to suggest drug-target interactions by defining the “druggable metaproteome”.

We believe that the new data offers enough support to uMetaP as a significant qualitative (not only quantitative) improvement in the metaproteomic field. Thus, we have eliminated expressions like “quantum leap” from the new version.

Finally, while we did not dissect the benefits brought by each of the LC-MS components, there is literature supporting a compound benefit due to increased ion transmission brought by the CaptiveSpray, the sharper peaks resulting from a zero-dead volume IonOptics columns (doi.org/10.1038/s41467-024-46380-y), and the substantial sensitivity brought by the timsTOF Ultra mass spectrometer (doi.org/10.1038/s41467-024-49651-w) to reach the sensitivity improvements described in our study.

Comment 3.1: “While I will not challenge the increased sensitivity and number of detected protein groups by DDA, I have a general problem with DIA analysis, which is the cornerstone of their workflow. DIA-NN is known to have quite “loose” treatment of FDRs and protein inference compared to other DIA-platforms.”

Thank you very much for the comment.

Likely due to the delayed adoption of DIA methods in metaproteomics, we acknowledge the hesitations regarding the confidence of DIA analysis and DIA-NN, despite being increasingly used by leaders in the field (e.g. doi.org/10.1186/s40168-024-01766-4 and doi.org/10.20517/mrr.2024.21).

To address these concerns, in our previous work (doi.org/10.3389/fmicb.2023.1258703), we presented the first (to our knowledge) entrapment experiment demonstrating DIA-NN's superior FDR control compared to MaxQuant in complex metaproteomic samples. Additionally, recent studies by Jeroen Krijgsveld's and Uri Keich's laboratories demonstrating DIA-NN's effectiveness in controlling FDR with low sample amounts (similar to ours; doi.org/10.1101/2024.01.10.575010) and its superiority to Spectronaut and Encyclopedia-DIA (doi.org/10.1101/2024.06.01.596967).

On the contrary, we are not aware of any published data supporting the reviewer's statement “DIA-NN is known to have quite “loose” treatment of FDRs and protein inference compared to other DIA-platforms”. We would be thankful if such data could be pointed out to us, for incorporation into the current state-of-the-art set by the studies mentioned above.

Nevertheless, we have addressed the reviewer's opinion by performing new experiments:

- Figure R1 shows a new entrapment analysis comparing DIA-NN's FDR control to the latest version of Spectronaut (v19 or SN19; kindly provided by Biognosys AG as a one-month free version). Our results demonstrate that the FDR control by DIA-NN (v1.9) is superior or similar to the one performed by the latest version of SN19. As shown in Figure R1, the amount of SILAC-labelled *L. murinus* and of non-labelled *S. ruber* precursors in the non-spiked sample (used in our previous Figure 4 as control samples) are lower when using DIA-NN (cyan-dots in the plots) compared to SN19 (red dots). Both tools nearly saturated the target identifications (in this case, microbial precursors) when the estimated FDR is around 0.05.

Figure R1: Entrapment analysis to evaluate the FDR performance of DIA-NN (v1.9) and SN19. Triplicate of mouse gut microbiome samples without spike-in of SILAC-*L. murinus* and non-labelled *S. ruber*, were searched against a spectral library that includes 94,443 microbial mouse gut precursors, 9,231 heavy-labeled *L. murinus* precursors, and 8,422 *S. ruber* precursors. FDR filters were turned off in both software. Thus, any *S. ruber* precursors and heavy-labeled *L. murinus* precursors identified were considered false identifications and used for the calculation of estimated FDR (doi.org/10.1038/s41592-019-0638-x) for identified microbial precursors (targets).

- Figure R2 shows the evaluation of the PEP (posterior error probability) value distribution of identified precursors (FDR \leq 0.01) from DIA-NN (v1.9) and SN19. The results demonstrate similar confidence: DIA-NN identified peptides had a mean PEP score of 0.0099 ± 0.0254 , while Spectronaut v.19 peptides had a mean PEP score of 0.0097 ± 0.0315 .

Figure R2: The posterior error probability (PEP) distributions for identified microbial precursors with FDR \leq 0.01. (A) the overlap of precursors (FDR \leq 0.01) identified by DIA-NN and SN19. (B) the histogram shows the number of precursors distributed along the PEP values in two software. (C) Boxplot shows the PEP distributions of identified precursors from DIA-NN and SN19, where DIA-NN exhibits higher consistency of PEP values for the identified precursors.

- Figure R6 (see Page 10) demonstrate a better protein-grouping-dependent quantification performance of DIA-NN versus SN19.

These new analyses confirmed the latest publications (see above) and our previous published data, supporting the reliability of DIA-NN for analyzing DIA-type data in metaproteomics. We would like to point out that an additional advantage compared to Spectronaut is the cost-free availability of DIA-NN.

Comment 3.2: “In combination with noisy spectra and huge databases, this can lead to dramatic accumulation of false positives. Of note, it would be interesting to know how many identification were filtered out falsely assigned SILAC label.”

We fully agree with the reviewer that this could be problematic if the two requisites (huge databases and noisy spectra) were present in our study. However, this is not the case as shown by the presented data:

- The previous version of our manuscript uses a project-specific database (48,970 protein sequences) and not the huge mouse gut catalogue (~2.6 M protein sequences; doi.org/10.1038/nbt.3353). To bring context, the size of our database is 10-100 fold smaller as the ones used in recent metaproteomic studies (437,578 protein entries in doi.org/10.1186/s40168-024-01766-4, or 1,217,422 entries in doi.org/10.1038/s41522-023-00373-9).
- The visual inspection of the spectra corresponding to ultra-low amounts of spike-in used in our study (ie, SILAC-labelled *L. murinus* and non-labelled *S. ruber* in previous current Figure 3 and Supp. Figure 3), notably a sensitivity never reached in metaproteomics, demonstrates the quality of the spectra generated and supports the optimal control of false positives in our study.

Finally, in response to the reviewer’s interest, we detected a total of 105 SILAC peptides out of 100,529 in the non-spike controls (data presented in the previous version of our manuscript), therefore corresponding to an FDR value of 0.001%, well below the widely-accepted 1%. In addition, as mentioned, identified *L. murinus* and *S. ruber* peptides at the lowest spike-in amounts were checked visually to ensure the reliability of their detection.

Comment 4: Also, the protein inference problem will lead to attribution of one peptide sequence to several protein entries in the database; if not properly grouped, this will lead to an artificial increase in the number of detected proteins (and taxonomic phyla, of course).

We share the reviewer's concern about protein interference being a significant challenge in proteomics in general and especially in complex metaproteomic samples.

However, we would like to clarify what seems to be a misunderstanding: uMetaP's improvements in taxonomic depth do not suffer from the issue mentioned by the reviewer because we employ a peptide-centric approach (and not a protein-centric one), which is a well-accepted strategy in metaproteomics (doi.org/10.1002/pmic.201400361, doi.org/10.1093/Bioinformatics/Bty466).

The peptide identification reliability through proper FDR control (as discussed extensively above) together with increased depth and sensitivity reached by uMetaP (now, using the new novoMP module; see above) become paramount as the basis for the increased taxonomic identification.

Comment 4.1: It is visible in the Methods section that the authors did not use the default inference settings of DIA-NN, but it is not clear what was the reason and what influence this had on the dataset.

Thank you for your observation.

Similarly to the lack of proper FDR-related benchmark in DIA-dependent metaproteomic studies (see above), the influence of the protein inference mode of choice in DIA-NN ("non-heuristic" as used in doi.org/10.3389/fmicb.2023.1258703, or "heuristic" as used in doi.org/10.1186/s40168-024-01766-4) has not been validated in metaproteomic studies. Future studies focused on this important question will shed light on this topic.

After consultation with Prof. Vadim Demichev (DIA-NN developer), none of the two options offers a perfect/reliable choice. The main influences for both options are:

- Using heuristic protein inference will reduce the number of protein IDs reported in the final list (as reported by Dumas et al. in Figure 3C; doi.org/10.1186/s40168-024-01766-4). This is because every shared peptide among two or more proteins will be assigned to only one of the proteins. However, this assignment does not follow any criteria that assure its correctness. As explained in <https://github.com/vdemichev/DiaNN/discussions/830>, the choice is based on the presence (or lack of) of specific proteins in the SwissProt database or the alphabetic order of their codes.
- Furthermore, using heuristic protein inference has limitations for differential abundance calculations. As explained in the above-mentioned forum discussion: "If you have two proteins, A and B, with peptides A:X,Y, B:Y,Z, then heuristic will quantify A with X and Y and B with Z only."

Therefore, our choice to not use heuristic protein inference will increase the number of protein groups identified (but does not affect the number of unique proteins) but will avoid limitations for quantifying differential abundance. The latter was key in our decision to increase the confidence in taxonomical and functional changes detected *in vivo*.

Protein grouping has value for functional analysis. Standard practice involves using the first protein ID within a group for functional analysis (doi.org/10.1002/imt2.25), as proteins within a group typically share similar Gene Ontology (GO), Clusters of Orthologous Groups (COG), or KEGG annotations.

For these reasons, we decided to use all the available peptide information to quantify every protein (therefore, every functional pathway) without relying on non data-supported inference criteria (see above).

As pointed out before, the effects of this option have not been addressed in metaproteomic studies using DIA-NN. To fill this gap, we now offer new analyses. The non-spike control samples presented in Figure 4 of the previous version were used for the following tests:

- First of all, enable/disable the “heuristic protein inference” option did not change the amount of identified peptides (complete overlap in the Venn diagram below).

Figure R3: Overlap of peptide identifications from DIA-NN when heuristic protein inference was enabled and disabled.

- Secondly, protein grouping is different as expected (see the overlap below). Note that 94% protein groups reported with heuristic inference were covered by protein groups reported when it was heuristic inference was off.

Figure R4: Overlap of protein groups identified from DIA-NN when heuristic protein inference was enabled and disabled.

- Though proteins grouped can share the same functions (due to sequence homologue), this is not always the case. Thus, we broke down the protein groups into single proteins and checked their functional annotations. As shown in Figure R5A, all proteins from the heuristic search overlapped with the heuristic-disabled search. Furthermore, 5,162 proteins solely reported in the “heuristic-off” search mapped to 8 unique KEGG pathways and 70 unique PFAM classes, indicating that choosing the “heuristic protein inference” can lead to functional information loss.

Figure R5: Functional comparison of proteins reported by DIA-NN when heuristic protein inference was enabled and disabled. (A) Overlap of broke-down proteins in two grouping methods. (B) Overlap of annotated KEGG pathways annotated with common proteins (18,873) and unique proteins (5,162 with disabled heuristic protein inference). (C) Overlap of annotated PFAM classes annotated with common proteins (18,873) and unique proteins (5,162 with disabled heuristic protein inference).

- Moreover, apart from the number of protein groups reported, inference algorithms do affect the quantification of protein groups. To benchmark this, we compared the aforementioned two searches (heuristic inference on and off) in DIA-NN with two inference algorithms offered in SN19 (namely IDPicker and All matching proteins). Only commonly quantified protein groups from 2 searches in each tool were considered. As shown in the correlation plots below, commonly quantified protein groups from DIA-NN searches (16,845 protein groups, cyan) correlate much better than the ones from SN19 algorithms (15,221 protein groups, pink). This finding, together with the equal FDR control (see answer to Comment 3.1), further supports the use of DIA-NN for the analysis of our DIA-PASEF datasets.

Figure R6: Correlation of commonly quantified protein groups between two protein grouping algorithms offered in DIA-NN (left) and Spectronaut 19 (right).

Based on these new set of data, we have decided to keep the non-heuristic protein inference option.

Comment 4.2: Limited sequence coverage in metaproteomics measurements is indeed a problem, but increasing false positive rates or tempering with protein inference is certainly not a solution.

Thank you for the comment.

As demonstrated with new entrapment experiments (see above), our DIA-NN analysis does not “increase the false positive rates”. Furthermore, we have discussed the advantages and disadvantages of the different protein inference options in DIA-NN and demonstrated the lack of impact on protein identification but the positive impact on functional analyses and protein quantification of the non-heuristic option chosen in our study (see Figures R3-6, above).

Moreover, we now offer new analyses demonstrating the improvements offered by uMetaP in a previously published dataset and at three different coverage levels: peptide-protein coverage, peptide-taxa, and protein-function coverage. We applied uMetaP to 21 mouse fecal samples that were analyzed in our previous publication (doi.org/10.3389/fmicb.2023.1258703). As shown in Figure R7 below, uMetaP increased 21.9%, 24.1%, and 11.5% the coverage at all these three levels compared to

the same samples analyzed by our previous workflow (doi.org/10.3389/fmicb.2023.1258703), which demonstrates the benefits of this new pipeline for metaproteomic studies.

Figure R7: Coverage improvements offered by uMetaP in comparison with our previous workflow. Left: Number of peptides assigned to commonly detected protein groups (41,483). Middle: Number of peptides annotated to commonly detected species (244). Right: Number of proteins mapped to commonly annotated KEGG pathways (264).

Comment 4.3: For any future submission to a more specialized journal, I would advise the authors to benchmark their DIA approach against other commonly used DIA platforms (especially Spectronaut) and include positive and negative controls in their samples and databases to calculate the real FDRs of their measurements.

Thank you very much for the suggestion.

We fully agree that the increased use of DIA-based methods in metaproteomics lacks a careful benchmark of the most extended DIA data analysis solutions (i.e., DIA-NN vs Spectronaut). We believe that the new analyses presented here support the data analysis options chosen during our study (related to FDR control, peptide-taxa mapping and protein inference).

The extent of the new experiments presented in this detailed answer goes beyond the study's main goals. The importance of this topic for the metaproteomic community, together with our experience in developing and validating DIA methods ([doi: 10.1002/pmic.201700021](https://doi.org/10.1002/pmic.201700021), [doi: 10.1074/mcp.RA117.000314](https://doi.org/10.1074/mcp.RA117.000314)), stimulates us to follow the reviewer's suggestion and prepare an additional bioinformatic-based study presenting the benchmark experiments shown above.

Reviewer #2:

I co-reviewed this manuscript with one of the reviewers who provided the listed reports. This is part of the Nature Communications initiative to facilitate training in peer review and to provide appropriate recognition for Early Career Researchers who coreview manuscripts.

Thank you very much for the review.

Reviewer #3:

Comment 1: Xian and colleagues have described an ultra-sensitive metaproteomic approach that enables single-organism proteomics and captures taxonomic profiling at a greater depth than previously achieved. While this approach represents a significant advancement over prior strategies, several claims appear overstated or not directly supported by the data presented. For example, the authors compare the depth of their proteomic approach to 16S strategies but did not characterize the microbiome of their samples using the 16S method for a direct comparison. Instead, they compared their proteomic results to published 16S studies conducted on different samples questioning the validity of this comparison.

Thank you very much for the positive feedback and the constructive criticism, which we have followed to improve our study.

In the follow-up text (see below), we offer answers to all points raised, including new experimental data to support our claims regarding the deeper taxonomical depth reached by uMetaP compared to 16S rRNA. In addition, we have added a similar comparison with the deeper shallow shotgun sequencing performed in a second set of mouse colonic samples.

As demonstrated in Figure R8, uMetaP reaches a deeper taxonomical profile than 16S rRNA in the same 21 mouse fecal samples. Notably, 16S rRNA analysis was independently performed by the Joint Microbiome Facility at the University of Vienna.

Figure R8: Comparison of detected taxa between the previous workflow (performed on a timsTOF Pro instrument), uMetap and the 16S analysis of the same mouse fecal samples (n=21). A minimum of 3 taxa-specific peptides was set as cut-off for the metaproteomic data. Note that: 250ng of peptides were analyzed in a 2-hour gradient on the timsTOF Pro (our previous workflow), while only 50ng peptides were analyzed in 1-hour gradient in uMetaP. All metaproteome data were processed in DIA-NN (v1.9).

Furthermore, uMetaP reaches a better profiling depth than shallow shotgun sequencing in 23 mouse colonic content samples from 4 different conditions (Figure R9). The 16S rRNA analysis on the same samples only reached sequence depth at the genus level. 16S rRNA and shallow shotgun sequencing were independently performed by our collaborators from Professor Dirk Haller's laboratory (TUM) as published in Urbauer et. al (doi.org/10.1016/j.chom.2024.06.013). We have added these new data as Supplementary Figure 4B.

Figure R9: Comparison of detected taxa using different methods in microbiome research on the same sample set. A minimum of 3 taxa-specific peptides was set as cut-off for the metaproteomic data.

Comment 2: Lines 49 to 51: The process of pooling peptide samples from previously prepared samples is described.

The reviewer is correct, we described it in lines 49-51 in our previous version of manuscript. We are not sure to understand the concern raised by the reviewer. We would be grateful if the reviewer could give us a more detailed explanation.

Comment 3: *L. murinus* should be *L. murinus* and *S. ruber* should be *S. ruber*.

Thanks to the reviewer for the correction. We have replaced the species names with "*L. murinus*" and "*S. ruber*" throughout the manuscript as well as in the figures.

Comment 4: Line 302: The authors claim that uMetaP achieved notable improvements across all taxonomic levels (a three-fold increase at the species level). However, it is unclear what these improvements are and which data support this statement.

The improvements were shown in the previous Supplementary Figure 3B (as stated in the previous line 305). As explained in the text (previous lines 303-305), we compared the taxa obtained with uMetaP (previous Figure 3A using 25 ng of injected peptides) with our previous workflow (doi.org/10.3389/fmicb.2023.1258703) where 250 ng of pooled mouse fecal peptides were analyzed on a timsTOF Pro mass spectrometer in a

60-min gradient. The results demonstrated a 3-fold increase in the number of species detected by uMetaP (Figure R10; 66 microbial species vs. 203 microbial species).

Figure R10: Species overlap between our previous workflow and uMetaP workflow in the previous version of the manuscript.

However, in the new version of the manuscript, we have included two important improvements that influence this comparison: the use of the MGnify database and the new novoMP module. To better illustrate this improvement, we now include the species overlapping as Figure R11 (see below): The Venn diagram shows how uMetaP covered 65 out of the 66 species previously detected. The only species non-detected by uMetap was *Lepisosteus oculatus*, which was previously detected only with 3 peptides (our threshold for identification). It is very likely that the taxonomical variability among the two different fecal pools used in these studies accounts for this unique miss.

Figure R11: Species overlap between our previous workflow and new uMetaP powered with novel proposed NovoMP module.

Comment 5: Additionally, the authors state that this taxonomic coverage sets new standards in metaproteomics by matching the average performance offered by full-length 16S rRNA of the human gut microbiome. What do they mean by this? It would seem necessary for the authors to characterize the gut microbiome of their own samples using full-length 16S rRNA to substantiate this claim.

We referred to the comparison of our data to the published coverage of full-length 16S rRNA, which was used as a benchmark of the lower limit of detection level (LLoD) in the, until our study, the paper of reference in metaproteomics on this topic (doi.org/10.1021/acs.analchem.2c02452).

We agree with the reviewer and would like to thank her/him for the suggestion. We now offer two new experiments in which we compared the taxonomical coverage reached by uMetaP compared to:

- 16S rRNA in a set of 21 mouse fecal samples (see Figure R8, above), and
- Shallow shotgun genomics a set of 23 mouse colonic samples (see Figure R9, above).

As shown in the Figure R8, Figure R9 and the new Supp. Figure 4B, the new uMetaP workflow offers significant taxonomic increases compared to the tested genomic methods.

Comment 6: Lines 307 to 309: The authors state that they identified three species with an abundance of 10 pg. How was this measured?

This seems to be a misunderstanding.

We did not measure 10 pg of peptides for each of the three species. Instead, we referred to the ability to identify 3 bacterial species when the dilution of the original sample reached a theoretical amount of 10 pg (previous Figure 2 and previous lines 307 to 309).

Comment 6.1: Does this approach enable absolute quantification of the bacterial species present in the microbiome? Is there a direct correlation between the peptide intensities and the abundances of *S. ruber* and *L. murinus*?

We thank the reviewer for this important question.

In our opinion, absolute quantification of bacterial species in a microbiome is still not possible for the following main reasons/challenges:

1) Complexity of microbiome samples: metaproteomic samples typically contain a vast diversity of organisms, including bacteria, archaea, fungi, and viruses. This diversity results in a highly complex mixture of proteins, making it challenging to identify and accurately quantify unique peptides/proteins representing a specific taxon.

2) The absolute quantification strategy using a mass spectrometer involves the spike-in of stable isotope-labelled proteins and/or peptides as internal standards, which are endogenously presented in the sample. In metaproteomics, incorporating such standards uniformly across all organisms and proteins of these complex sample would be prohibited.

3) Protein databases used for metaproteomics studies are often incomplete, especially for non-model organisms. The absence of comprehensive databases for all possible organisms in a sample limits the accuracy of protein identification and quantification.

Following the logic presented above, this experiment shown in previous Figure 4 (current Figure 3) only allows for relative quantification. However, values of relative quantification produced by an experiment that does not take into account sample preparation losses are of limited value. For this reason, we performed the experiment shown in previous Figure 4 (current Figure 3), which, to the best of our knowledge, is the first one calculating LLoD and LLoQ taking into account sample prep losses (thus termed as real LLoD or LLoQ)

We did see the correlation between the peptide intensity and the number of bacteria spiked in the samples. As shown in the figure below (Figure R12), we showcased the two peptides of *L. murinus* (IGVIEHLLDK) and *S. ruber* (NVSVPAHQAEK) presented in

previous Figure 4C-D in our manuscript. The average intensities (triplicates) of peptides decreased gradually with less bacteria spike-in.

Figure R12: Intensities of two example peptides in different spike-in conditions.

Comment 7: Line 413: The authors claim that, on the taxonomic front, uMetaP achieves significantly deeper analysis, surpassing the coverage of commonly used 16S rRNA. Which data support this statement? This evidence is not presented in the paper. Again, a direct comparison would likely be necessary.

Please, see our previous answer and the new experiments performed to address the same concern raised previously by the reviewer.

Reviewer #4:

Comment 1: Xian *et al.* present a metaproteomic analysis of a mouse microbiome sample utilizing the latest timsTOF Ultra mass spectrometer, referred to as uMetaP. The new instrument enabled them to measure over 100K peptides in a 30-minute MS run and over 120K peptides and approximately 50K protein groups in a 66-minute run, marking one of the most extensive single-shot metaproteomic studies to date. uMetaP demonstrated ultra-high sensitivity, with a coefficient of variation (CV) of less than 20% for a 10pg peptide injection and less than 10% for a 5ng peptide injection (Figure 2). Additionally, through a heavy-labeled spike-in experiment and a two-species mix sample analysis, they showcased a remarkable lower limit of detection and quantitation (LLOQ of 1e-3%) and achieved potential single-bacterium resolution with peptide injections as low as 500fg.

Overall, this study is technically robust, comprehensive, and significantly advances the field of metaproteomics, illustrating the instrument's applicability in metaproteomic analyses. However, my main concern with the current manuscript is the overly strong and somewhat bold claims and interpretations of their results.

Thank you very much for acknowledging the significant advances that uMetaP represents for the metaproteomic field. Also, we truly appreciate the constructive criticism.

In the follow-up text (see below), we have answered all the points raised by the reviewer, including new experimental data to support our claims.

Comment 2: The authors have termed their solution as uMetaP, which might lead readers to believe that this study introduces a new strategy for metaproteomics.

However, I did not observe a novel workflow for metaproteomics in the manuscript beyond the utilization of a new timsTOF Ultra mass spectrometer, when compared to their recently published work with an earlier version timsTOF Pro, as well as other previous metaproteomic studies.

Thank you very much for the criticism.

We humbly acknowledge the reviewer's concern regarding novelty. We have added a new methodological module to uMetaP: a *de novo* sequencing strategy for metaproteomic database construction from DDA-Pasef datasets. We have named this module novoMP.

The details related to its novelty, construction, benchmark, and application have been described in detail in our reply to Reviewer's #1, as well as in the new version of the manuscript (new Figure 1-2, Supp. Figure 1-2, and Supp. Figure 3F).

Comment 3: The depth of metaproteomic analysis achieved in this study is comparable to a recently published study using Orbitrap Astral mass spectrometer.

We thank the reviewer for pointing out this comparison.

We regret to disagree with the reviewer. A detailed analysis of the published data mentioned by the reviewer reinforces the fact that uMetaP is the up-to-date metaproteomic solution in terms of profiling depth, quantification accuracy, and sensitivity:

- As for the depth, we would like to acknowledge critical differences between the two studies. The Astral study (Dumas et. al; doi.org/10.1186/s40168-024-01766-4) was based on human fecal samples, while ours uses mouse feces. Furthermore, Dumas et. al used a database comprised of 437,578 protein entries for DIA analysis, while ours is 2 times smaller (208,254). Unfortunately, we can not comment on the FDR applied for DIA-NN in Dumas et. al as it is not described in the Method section of the paper. Assuming that Dumas and colleagues used 1% at peptide and protein levels (as done in our study), and given their 2-fold bigger database, the protein numbers reported by Dumas et.al. carry a significantly higher amount of false positive identifications. However, and despite these differences which make the direct depth comparison difficult (if not impossible), Dumas et. al. reported the identification of 118,262 peptide sequences and 37,934 protein groups when using 125ng of peptide in column and 60min gradient length versus 141,811 peptides and 79,693 protein groups from our study when using 100ng of peptide in column and 66min gradient length (new Figure 2A): a 2-fold increase in the protein profiling depth.
- Unfortunately, the comparison of the quantification accuracy and sensitivity is not possible as Dumas et. al. did not study them. These two parameters are important in proteomics (and therefore, in metaproteomics) as they dictate the ability to report biological changes accurately and sensitively.

Comment 4: Furthermore, the authors frequently use terms such as "unprecedented," "quantum leap," and "unparalleled" to describe the identification and sensitivity achieved. While these descriptions may be accurate in comparison to their previous study with timsTOF Pro, they do not hold true when considering more recent advancements in the field (such as the Orbitrap Astral work mentioned above).

We acknowledge the point raised by the Reviewer.

We fully agree with the reviewer that the field of metaproteomics will benefit from avoiding superlative language (e.g., “The *astounding* exhaustiveness and speed of the Astral mass analyzer for highly complex samples is a *quantum leap* in the functional analysis of microbiomes” Astral paper cited by the reviewer). We have performed an exhaustive revision of the superlative adjectives used in the new version of the manuscript accordingly.

As for the comparison with the Orbital Astral work performed in Jean Armengaud’s laboratory (doi.org/10.1186/s40168-024-01766-4), the data shown in our manuscript confirm the significantly superior performance of uMetaP (see the answers to Comments 3 above).

Comment 5: The idea of achieving single bacterium proteomics (SBP) is ambitious. In this study, the authors used a two-bacteria mix and tested the peptide injection amount (500fg) for MS analysis to demonstrate single bacterium resolution. However, this approach falls short of true SBP, which would require starting from an individual bacterium for sample processing, rather than using a peptide injection amount equivalent to that of a single bacterium. Additionally, based on the results presented, SBP is unlikely to be applied to microbiomes at this moment.

Thank you for the comment.

As we stated (previous manuscript text: lines 368), we aimed to test if the sensitivity improvements brought by uMetaP could tackle “the challenge of detecting single bacterium peptide amounts (sub-picogram levels)”. Furthermore, we discussed how our results suggest that SBP is unlikely using the timsTOF Ultra (“We believe the first factor to parallel the expected losses during future SBP sample preparation protocols, highlighting the need for further improvements in current methods (doi.org/10.1038/s41592-023-01785-3”); previous manuscript text: lines 438-440). In summary, we have not claimed to aim for achieving SBP, nor that our results demonstrated SBP.

However, we would like to point out that uMetaP is the first study, to the best of our knowledge, to demonstrate that single bacterial resolution is now possible. This is the classical approach that was taken in the early days of single cell proteomic (SCP) field, and it is still used for evaluating the ability to profile single-cell comparable inputs. An example of this strategy is the latest manuscript from the Karl Mechtler’s lab using the Astral mass spectrometer (doi.org/10.1101/2024.02.01.578358; please see lines 80-81 of the preprint).

As we also discuss in the previous version of our manuscript, and to frame the proper context for our results, uMetaP reaches a bigger proteome coverage than the one demonstrated in the above-referred preprint (which sets the current state-of-the-art in the SCP field) for the same input amount and similar chromatographic gradient (previous manuscript text: lines 440-443). As a step further in this evaluation, we demonstrate the challenges of working with such extreme sample amounts (500 fg) for calculating quantitative accuracy (previous Figure 5E). We expect this to be improved with upcoming instruments and new techniques.

The ultra-sensitivity reached by uMetaP is demonstrated in the manuscript using several experimental evidences (e.g., Figure 3). Moreover, as suggested by the Reviewers, we now include a novel bioinformatic tool (novoMP) and *in vivo* results

that have significantly improved the reach of our study. We strongly believe that this SBP experiment deserves a specific and separate study. For this reason, we decided to remove the result section “Pushing the sensitivity limits to quantify biology at single-bacterium resolution” from the current version of our study.

Comment 7: Line 73: why timsTOF Pro, not Ultra, was used to evaluate the labeling efficiency?

At the time of developing the SILAC labelling, only the timsTOF Pro was available. We don't expect the analysis in the Ultra could change the main result of the experiment: we have reached a very high labelling efficiency, especially taking into account the challenges of the SILAC method in bacteria (doi.org/10.1016/j.jprot.2018.12.011).

Comment 8: Details are needed regarding the database search and the method used to calculate labeling efficiency. Additionally, some bacteria can synthesize amino acids from others, such as Arginine; did the authors examine heavy labeling in amino acids other than K/R?

We checked the labeling efficiency by analyzing the heavy-cultured *L. murinus* in DDA-PASEF mode, and the data were searched against its reference proteome in Fragpipe. As a result, a total of 60,485 PSMs were identified (1% FDR), corresponding to 12,852 unique stripped peptide sequences (see previous Supplementary Table 9). In cases where multiple PSMs were assigned to the same peptide, we only kept the most intense PSMs of one peptide for both labeled and non-labeled forms if the latter was co-identified. If the peptide was identified only in either the heavy- or light-labeled form, the missing intensities were imputed as 1 to apply the following formula (doi.org/10.1016/j.jprot.2018.12.025) for each peptide:

$$\text{Peptide labeling efficiency} = \left(\frac{\text{Intensity}_{\text{Heavy}}}{\text{Intensity}_{\text{Heavy}} + \text{Intensity}_{\text{Light}}} \right) * 100$$

The average labelling efficiency of 12,852 peptides is 97%, (see previous Supplementary Table 9, sheet 2). Of note, only 146 peptides were identified as non-labeled out of 12,852 peptides, whereas 10,808 peptides were identified only in heavy-labeled forms.

We now incorporate this detailed explanation in the revised version.

As the reviewer pointed out some bacteria are auxotrophic for amino acids, which is the main challenge when aiming to reach high labelling efficiency (i.e., >95%; doi.org/10.1007/978-1-4939-1142-4).

Apart from arginine and lysine, we did not examine the heavy labelling of other amino acids for the following reasons:

- 1) In *L. murinus*, there are metabolic biosynthesis pathways that can convert certain amino acids into other amino acids. But pathways involving arginine and lysine are not demonstrated according to the BioCyc database (https://biocyc.org/GCF_000364205/NEW-IMAGE?type=ECOCYC-CLASS&object=Super-Pathways). One main challenge in examining other heavy-labeled amino acids derived from heavy-arginine and heavy-lysine is determining the exact number of heavy-labeled carbon and nitrogen atoms that are

transferred during these biosynthesis processes. This complexity makes it difficult to track accurately how these labelled atoms are incorporated into new amino acids.

- 2) Including too many variable modifications in the database search (e.g., potential unknown heavy labelling of other amino acids) significantly increases the search space, subsequently affecting the FDR estimation.

Comment 9: Line 100: twenty-microgram or twenty nanogram?

We appreciate that the reviewer found this typo.

It is corrected in the revised version.

Comment 10: Line 261: this is incorrect; there are several DDA-PASEF metaproteomic data published recently with more identified proteins (doi.org/10.1101/2023.11.29.569331);

Thank you for pointing out this interesting preprint.

We regret to disagree with the reviewer.

On the one side, MetaExpertPro identifies 58,952 protein groups (lines 200-205). The improvements added to our new manuscript's version (use of MGnify and development of novoMP) enabled the identification of 208,254 unique protein sequences.

On the other side, there are two important differences between both studies. First, MetaExpertPro used 30 pH fractions, compared to 8 fractions in our study. Second, and more importantly, the FDR applied in MetaExpertPro is around 5%, compared to 1% in our study.

Therefore, our study identifies 4-fold more proteins using DDA-PASEF using less sample fractions and more stringent FDR control.

Comment 11: Line 262-264: Why did the authors use in-silico digestion of the identified proteins, rather than the actually identified peptide sequences, to infer taxonomic coverage?

Thank you for the question.

To avoid a misunderstanding, we would like to point out that in our study we aimed to infer the "*theoretical taxonomic profiling potential*" of our reduced database (as stated in previous text: line 263), and not the taxonomic coverage.

Due to the natural bias in DDA acquisition, only the TopN abundant precursors/peptides can be fragmented and, therefore, can be identified. Moreover, the taxa annotation only uses taxa-specific peptides, which would not be captured in its totality by DDA. In this case, taxa annotation based on DDA-identified peptides is likely underestimating the landscape of all taxa potentially included. For these reasons, we performed in-silico digestion of the identified proteins and annotated those peptides to check the "*theoretical taxonomic profiling potential*" of our reduced database.

In the revised manuscript, we have excluded this analysis for the constructed metaproteomic database.

Comment 12: Line 306-307: I think this comparison is not fair; the identified 220 species in this study are from all four kingdoms of microorganisms, whereas the 16S

rRNA sequencing in the cited study focuses only on bacteria. How many bacterial species were identified in this study, and how does this compare to the 16S sequencing results? Ideally, a matched sample sequencing would be needed to directly compare metaproteomics and 16S sequencing.

Thank you for the comment.

The number of bacteria out of the 220 microbial species are 218 (previous version of our manuscript), and the two non-bacterial species are eukaryotes. Therefore, 99% of the uMetaP annotated species are bacteria, making the comparison with the mentioned study valid.

We have now added new experiments in which we compare the taxonomic coverage offered by uMetaP, 16S rRNA and the deeper shallow shotgun sequencing in the same samples. The results clearly indicated that uMetaP detects more species than 16S sequencing. For more details, please see the response to Rev#3 (Figure R8-R9 above), as well as the new Supp. Figure 4B.

Comment 13: Line 316-319: The authors highlight the hidden functions (PUFs, sProt and AMPs) throughout the manuscript from the title to introduction and discussion, readers would expect a more detailed analysis of these portions of information, rather than just the numbers of identifications;

We agree with the reviewer that a detailed analysis of these proteins is very interesting as demonstrated by a recent Cell paper from Pedro Coelho's laboratory (doi.org/10.1016/j.cell.2024.05.013).

However, the breath of proteins identified in our study, and the unknown nature of these molecules leave the only option for structural similarity models as recently performed by Tavis and Hettich (doi.org/10.1186/s12864-024-10082-y), which is behind the scope of our study.

We would like to point out that in parallel to this review process, Wang et.al. (Daniel Figeys's laboratory) has published a similar description of the discovery of sProt and AMPs (doi.org/10.20517/mrr.2024.21). In line with the deeper profile of uMetaP compared to classical metaproteomic approaches, in the new version of our manuscript we reached 1.5 to 3.4 times more identifications to these efforts (new Supp. Figure 2E).

We believe, that the resources shared by both studies (Wang et. al., and this manuscript), together with the tools developed by Tavis and Hettich, open the possibility to initiate exciting projects focused on shedding light on the function of these proteins.

Comment 14: Line 328: is this really the first time in metaproteomics? In ref#6 of the manuscript, Duan *et al.* used a heavy labeled E. coli to assess the limit of detection in metaproteomics as well.

Thank you for pointing this out.

We believe our statement (previous text: lines 328-329) to be correct due to the following reasons:

- We explicitly referred to the "real" LLoD and LLoQ. The choice of the word "real" is based on the fact that our study is the first one that acknowledges the losses/variability introduced during the whole sample preparation method. It is well known that this is the main source of variability in

proteomic/metaproteomic studies. Therefore, sensitivity calculations like the LLoD and LLoQ in metaproteomics should take it into account. Duan *et al.* mixed the heavily labelled peptides with human stool peptides. Thus, this pioneering study reports the LLoD at the technical level (aka. sensitivity of their LC-MS method) but not the “real” LLoD in metaproteomics.

- Another important difference between the two studies is the labelling method. While ¹⁵N metabolic labelling of *E.coli* (used by Duan *et al.*) offers a cost-effective approach, it is uneven labelling and the background noise can limit LLoD and LLoQ in metaproteomics. Using heavily labelled arginine and lysine (our study) provides a more targeted and sensitive approach, but comes at a higher cost and might not be universally applicable to all organisms within a complex metaproteome.

Comment 15: Line 361: Is it really 5000 times greater compared to reference #6? Duan reported a limit of 0.5% with their method, while here it is reported as 1e-3 %; shouldn't this be 500 times instead of 5000 times? Maybe I missed something here. More details on how the comparison was made are needed to provide an accurate and fair representation of the reported increase.

We apologize if the unclear calculation is not clear.

As mentioned in the text (previous text: lines 357-358), it is estimated that 1×10^{12} bacteria are presented in 1 gram of mouse feces based on the genomic calculations ([doi.org/ 10.1016/j.dib.2021.107409](https://doi.org/10.1016/j.dib.2021.107409)), which makes 1×10^{10} bacteria in 10 mg of feces (the amount that we used as background). For calculating the limit of detection, we confidently detected peptides from 10,000 spike-in *L. murinus* and *S. ruber* bacteria. Thus, $1 \times 10^4 / 1 \times 10^{10} = 0.000001$, which in relative abundance percentage is 0.0001%. Compared with the 0.5% value reported by Duan *et al.*, it is indeed 5,000 times.

We have clarified this calculation in the new version of the manuscript.

Comment 16: Line 402: unassigned spectral data; to my understanding, the information in these data is truly hidden function or dark matter (instead of those identified proteins without known functions).

We agree with the reviewer.

Unassigned spectra, which can account for > 70% of all spectra produced by the mass spectrometer, are a wealth of hidden functional and taxonomic information. Thus, they are a top priority for the metaproteomic field.

As we mentioned in the text, *de novo* algorithms can potentially help the metaproteome community decipher this dark matter. However, due to the peptidome complexity of microbiome samples, the confidence application of *de novo* to metaproteome data is challenging. Following Reviewer's #1 and #3 suggestions, we took over this challenge in the new version of our manuscript.

We now present a novel novoMP module harboring i) the first algorithm trained in PASEF data structure, ii) a multi-tier filtering system to increase the coverage of metaproteomic databases using very highly confident *de novo* identifications, and iii) a novel orthogonal FDR control strategy using DIA-PASEF.

We kindly refer the reviewer to our reply to Reviewer#1 and the new version of the manuscript for more details.

Comment 17: Line 406-407: Does the identification in this study surpass the previous study with Astral? In the Astral study, ~122K peptides and 38K protein groups were identified with a 30min MS run, while this study reports about 102K peptides with a 30min MS run (Line 278).

Thank you for the question.

As answered to a similar question raised by the reviewer (see Comments #3 and #4, above), a comparison of our data supports the statement.

We would like to kindly point out that the numbers reported by the reviewer correspond to 5-times more peptide amount injected in the Astral (125 ng versus 25 ng in our study). Furthermore, Dumas et. al used a 2-times bigger database (see above). Assuming an equal 1% FDR, the numbers reported by Dumas et. al. will include more false positives.

A fair comparison demands looking at the most similar acquisition conditions. These correspond to 100 ng injected peptide amount and 66-minute gradient in our manuscript versus 125 ng injected peptide and 60-minute gradient from Dumas et. al. Under these conditions, Dumas et al. detected 118,262 peptide sequences and 37,934 protein groups, and uMetaP detected on average 126,819 stripped peptide sequences and 47,562 protein groups (Supplementary Table 5 of our previous manuscript). With the novoMP included in this version, uMetaP detected on average 141,811 stripped peptide sequences and 79,693 protein groups in the same triplicates of 100 ng (new Figure 2A).

We now clarify this comparison in the new revised version.

Comment 18: Line 408: to be applicable for large scale metaproteomic study, the instrument needs to be robust with stable performance. Could the authors comment on this as well when comparing to other instruments?

We appreciate the reviewer's point of view and agree with it.

The robustness and stable performance of instruments are required for large-scale metaproteomic studies. Several published studies have demonstrated the robustness and stable performance of the timsTOF platform (doi.org/10.1101/2024.05.29.596405) as well as other instruments in large-cohort studies ([10.1021/acs.jproteome.3c00646](https://doi.org/10.1021/acs.jproteome.3c00646), [10.1038/s41467-022-34919-w](https://doi.org/10.1038/s41467-022-34919-w), [doi:10.1038/s41467-024-44986-w](https://doi.org/10.1038/s41467-024-44986-w)). To the best of our knowledge, metaproteomic studies analyzing 1000's of samples are still awaiting. This is a goal that is beyond the scope of our study. However, we strongly believe it is an attainable goal given the performance of the latest instrumentations (including new liquid chromatography columns) in other fields (as indicated above).

However, it is worth mentioning that robustness and stable performance alone will not offer the best insights into the physiological role of the metaproteome without gains in profiling depth, quantification accuracy and sensitivity – all factors that our study characterizes in detail. We strongly believe that uMetaP and other future solutions will advance the application of metaproteomics to large-scale cohorts.

REVIEWER COMMENTS

Reviewer #3 (Remarks to the Author):

Comment 1.1: The authors have adequately addressed my previous concerns.

We highly appreciate the positive feedback from the reviewer regarding the revised manuscript. Below, we address the newly raised comments point-by-point.

Comment 1.2: However, the newly added section on defining the druggable gut metaproteome raises additional questions. While the concept of a “druggable metaproteome” is intriguing, this section appears to fall outside the microbiome-focused scope of the study, as it primarily characterizes the host proteome (“druggable proteome”) rather than the metaproteome.

Thank you for your insightful comment. We understand and acknowledge the reviewer’s concern.

Due to the bidirectional relationship between host and microbial functions in the gut environment, changes in the host proteome—particularly during intestinal inflammation—are likely to be strongly shaped by shifts in the gut microbiota. At the same time, the field of targeting microbial alterations using drug compounds is still in its infancy and faces several challenges, including issues of specificity and redundancy, as well as the limited targetable space resulting from incomplete metaproteome coverage. These limitations were the primary reasons we initially focused on targeting host proteome changes associated with the disease model.

Prompted by the reviewer’s comment, we have now expanded our analysis to go beyond host proteome targeting alone. Using a dataset recently released by Daniel Figey’s laboratory (doi.org/10.1101/2025.02.13.637346), we have:

1. Characterized the “druggable microbiota”, therefore completing the “druggable metaproteome” concept.
2. Explored potential synergistic and antagonistic interactions between approved drugs and the microbial changes observed in our study.

“Druggable microbiota” - targeting microbial activities

To explore the therapeutic potential of the changes we detected in the microbial community, we performed a differential regulation analysis of the microbial functions defined by the proteins identified in our dataset. To do so, microbial proteins were annotated using the COG (Clusters of Orthologous Groups) database (<ftp://ftp.ncbi.nih.gov/pub/COG/COG2024/data>), following the approach described by Zhang *et al.* (doi.org/10.1186/s40168-016-0176-z), to retrieve associated COG IDs. For each sample, the intensity values of protein groups annotated with the same COG ID were summed. These aggregated values were then subjected to statistical comparison across groups.

At D8, a total of 593 COG IDs were found to be significantly regulated (adjusted p-value ≤ 0.05), among which 216 COG IDs were associated with 59 distinct microbial functions or pathways (see new Supplementary Data 7). Among the associated microbial pathways, several COG IDs related to vitamin biosynthesis were particularly notable (see Figure R1, below), including vitamin B7 (biotin), vitamin B12, vitamin B9 (folate), and vitamin B2 (riboflavin). These findings are of interest because mammals, including mice and humans, are unable to synthesize these vitamins endogenously, making them promising targets for microbiota-focused interventions. Further, our new results go in line with the importance of microbial vitamin production in gut health and inflammation (doi.org/10.3390/nu14163383).

Figure R1: Scaled expression of COG IDs involved in Vitamin biosynthesis in Δ/Δ IEC_D8 and fl/fl_D8 mice.

Detailed exploration of these new data shows how, in cases where the metabolic synthesis pathways are well-characterised, our uMetaP approach could reach enough coverage to suggest mechanistic insights and, therefore, multi-target interventional strategies. For example, focusing on riboflavin (B2) biosynthesis, we found that two COG IDs—COG0108 and COG0196—were significantly regulated. COG0108 (EC 4.1.99.12) catalyzes a key reaction linking the pentose phosphate pathway to riboflavin biosynthesis, while COG0196 (EC 2.7.7.2) is involved in converting FMN to FAD, the active forms of riboflavin. Interestingly, COG0108 was down-regulated in Δ/Δ IEC_D8 mice, and we also observed down-regulation of two key enzymes in the pentose phosphate pathway (see Figure R2). These findings suggest a potential deficiency in microbial riboflavin production in colon-injured mice and point to vitamin biosynthesis pathways as potentially modifiable targets in gut microbiota-associated interventions.

[editorial note: figure redacted]

Figure R2: Overview of the riboflavin biosynthesis pathway and related metabolic routes, adapted from doi.org/10.3389/fbioe.2021.704650 and MetaCyc. Enzymes detected in our dataset that catalyze the respective reactions are indicated with EC numbers. Enzyme labels are color-coded: black denotes no significant change between Δ/Δ IEC_D8 and fl/fl_D8 mice, cyan indicates down-regulation, and red indicates up-regulation in Δ/Δ IEC_D8 mice.

We believe the new analyses and results demonstrate the potential to explore the still-limited targetable landscape of microbial-specific protein networks in health and disease.

Synergistic and antagonistic interactions between approved drugs and the microbial changes observed in our study during IBD

It is well established that drug treatments influence the composition of the gut microbiota (doi.org/10.1038/nature25979; doi.org/10.1038/nature15766). However, evidence of such alterations using metaproteomics is scarce. A recent preprint by Prof. Daniel Figey's laboratory (doi.org/10.1101/2025.02.13.637346) shows the first efforts in this field - fecal cultures were incubated with 312 compounds to investigate their effects on microbial taxonomy and functions.

Now, we used their publicly available database (https://shiny2.imetalab.ca/shiny/rstudio/MPR_viz/) to cross-reference our proposed drug candidates (listed in Supplementary Data 11) and identified potential drug-microbiota interactions *in vivo*. In the Figure R3 below, the analysis of the directional change in the disease mouse model (Δ/Δ IEC_D8) compared to the taxonomic changes reported by Figey's laboratory enables the establishment of synergistic and antagonistic drug effects hypotheses.

In the previous version of the manuscript, we described an increased abundance of the *Bacteroides* genus in Δ/Δ IEC_D8 mice compared to controls (Supplementary Fig. 4c). As an example of a potential antagonistic drug effect, *Bacteroides* showed a decreased abundance when cultured with drugs nimodipine, indomethacin, and fluoxetine (Figure R3, below). At the same time, we hypothesize potential antagonistic synergistic effects by some of the same drugs on other genera (eg., Nimodipine and Fluoxetine on *Eubacterium*). These analyses warn against simple predictions on the final phenotypic effect of drug treatments on the gut microbiome composition and function when considered as a whole.

Figure R3: Summary of drug effects on various genera reported by Li et al. (doi.org/10.1101/2025.02.13.637346) and the abundance changes in Δ/Δ IEC_D8 mice of our data.

In the previous version of the manuscript, we defined (lines 498-499) the “druggable metaproteome” as: *The collection of host and microbiota proteins within a given environment that possess the structural and functional properties necessary to be targeted by pharmaceutical agents.*

We believe that the additional analyses shown above have expanded and contextualised this concept. As a result, in the new version, we redefine the concept like (lines 515-517): *The collection of host and microbiota proteins or functions within a given ecological environment that are prioritised for therapeutic targeting.*

The new data and text revisions have been integrated into the updated manuscript (lines 330-339 and 371-379, Fig. 5a-5b and Supplementary Fig. 5f).

We sincerely thank the reviewer for her/his constructive feedback, which has helped us improve the quality and clarity this concept, very significantly.

Several aspects require clarification:

Comment 2: How novel is this approach in the context of studying the druggable proteome?

Comment 3: What advantages does the authors' methodology offer over existing approaches for analyzing the host proteome?

We thank the reviewer for these important questions, which we address together.

While the concept of the druggable genome/proteome has been extensively studied in fields such as oncology and human genetics, its application to host–microbiome interactions in the gut—particularly under inflammatory conditions—remains, to our knowledge, unexplored. The novelty of our “druggable metaproteome” concept lies in its integrated view of host and microbial proteomic changes, based on the premise that these changes are functionally linked and should be considered jointly when identifying therapeutic targets.

Unlike conventional host-centric proteomic studies that analyze isolated epithelial cells or tissue biopsies (e.g., doi.org/10.1093/ecco-jcc/jjae169, doi.org/10.1016/j.isci.2024.110550), our workflow enables the simultaneous profiling of microbial and host proteins from the same complex samples. This provides direct insight into functional host–microbe interactions *in situ*. Thus, our approach moves beyond simply cataloguing host proteome changes, toward capturing the dynamic interplay that may drive disease processes.

Importantly, we acknowledge that establishing causality between microbiome changes and host disease phenotypes is a major challenge in microbiome research. Our collaboration with Prof. Haller's group (doi.org/10.1016/j.chom.2024.06.013) has provided strong evidence for such causality: for example, mono-colonization with *B. caecimuris* was sufficient to reproduce tissue injury and gene signatures associated with disease, while antibiotic-mediated eradication of *Bacteroides* species rescued survival. These findings support the underlying rationale for defining a “druggable metaproteome” shaped by host–microbiota interactions.

The host protein targets prioritized in Fig. 5 were not analyze in isolation but rather in the context of microbiome alterations identified by our metaproteomic workflow, uMetaP. As shown in Fig. 4, we observed parallel shifts in microbial taxa, microbial metabolic functions, and host inflammatory pathways, underscoring the functional interdependence between the two compartments. Additionally, our use of multi-omic cross-species validation adds robustness and translational relevance to the identified targets.

To complete the conceptual framework, the new analyses, prompted by the Reviewer's suggestion (see answer to Comment 1.2, above), further consider therapeutic strategies that directly target the microbiome—such as via dietary supplementation of vitamins and/or probiotics, or treatments with genetically modified strains—as well as the effects of approved drugs on microbiome composition and function *in vivo*.

In summary, the causal link between microbiome shifts and disease phenotypes, combined with the integrative nature of our analyses, offers novel perspectives that distinguish the “druggable metaproteome” from traditional approaches focused solely on the host proteome.

Regarding the advantages of our methodology, we assume the reviewer refers to the concept of the “druggable metaproteome” compared to the classical “druggable host proteome.” If this interpretation is incorrect, we apologize and would be happy to address a reformulated version of the question.

The advantages of our approach can be considered both conceptual and practical:

- **Conceptually**, it defines host and microbial targets in the context of their relationship. This approach promotes a systems-level understanding of disease mechanisms that reflects the complexity of gut pathophysiology.
- **Practically**, it opens new therapeutic avenues by allowing the targeting of either or both components (host and microbiota), with the potential for synergistic/antagonistic benefits or for avoiding unintended synergistic/antagonistic effects (as illustrated in Figure R3 above)

In this way, the concept of a “druggable metaproteome” acts as a translational bridge linking microbial activity with host-targeted therapeutic strategies, particularly relevant in complex diseases such as Crohn’s disease and diabetes, where host–microbiota interactions play a central role.

Comment 4: Many of the identified proteins have already been reported as being affected in IBD. What new insights does this study provide beyond existing knowledge? Addressing these points would strengthen the impact and relevance of this section.

We agree with the reviewer’s comment.

As discussed in the previous version of the manuscript, several of the host proteins identified in our study (e.g., RELA, NOS2, ITGAM) have been previously implicated in IBD pathophysiology. We believe that the discovery of these proteins reinforces the validity of our findings, particularly given the multi-species and multi-omic nature of our approach.

Building on this foundation, our study offers several novel insights that go beyond existing knowledge:

- **Integration of microbial context:** Unlike previous host-centric proteomic studies, we contextualize host protein regulation within concurrent microbial taxonomic and functional changes. In this revised version of the manuscript, we speculate on therapeutic interventions targeting microbiome-specific proteomic/functional changes and explore potential synergistic or antagonistic effects of approved drugs (lines 330-339 and 371-379). This ecosystem-level perspective offers a new dimension to understanding IBD-like pathology.
- **Network prioritization of targets:** Through protein-protein interaction analysis, we identify key regulatory hubs within the host proteome that show high centrality during tissue injury. Some of these hubs have not been reported as linked to IBD and may serve as novel or combinatorial therapeutic targets, particularly within the context of host–microbiome dysregulation.
- **Drug repurposing:** By mapping regulated host proteins to known drug-gene interactions, we identify 77 approved drugs and 57 additional compounds approved for other indications that may be repurposed for IBD therapy. This provides a data-driven foundation for future hypothesis generation and therapeutic exploration.
- **Early detection of functional changes:** Using uMetaP, we detect coordinated alterations in both host and microbial proteins at the earliest stages of tissue injury (day 0), prior to major histological changes. These early shifts—not captured by image-based or metagenomic methods (doi.org/10.1016/j.chom.2024.06.013)—may represent biomarkers for early-stage disease progression.

Following the reviewer’s suggestion, we have included additional text in the Discussion section (lines 509–514) of the revised manuscript to emphasize these novel contributions.

Reviewer #4 (Remarks to the Author):

The authors have made substantial revisions to their manuscript, including incorporating a *de novo* peptide identification strategy for database construction and providing a concrete example to illustrate the applicability of their workflow. Overall, they have addressed most of the reviewers' comments. However, I do have some comments regarding the newly added *de novo* sequencing section, along with a few additional comments for the authors' consideration.

We sincerely appreciate the reviewer's positive feedback on the revised manuscript, particularly the integration of the *de novo* peptide identification strategy and the added example demonstrating the workflow's applicability. Below, we provide a detailed, point-by-point response to address the reviewer's additional comments.

Comment 1.1: The reviewer appreciates the use of a *de novo* sequencing strategy, enhancing the comprehensiveness of the database used for metaproteomic identification. This authors stated that 1.75 million PSMs from PASEF data were used to train the Novor model. However, there is no detailed information on these 1.75 million PSMs, nor on how they are specific to metaproteomics.

Thank you for pointing this out.

We have revised the Methods section and added more relevant details under the subsection "Training of the Novor algorithm with PASEF datasets and performance evaluation."

The training datasets consisted of PASEF runs from Vero cells and human K562 cells digested with a variety of proteases (GluC, Pepsin, Elastase, Chymotrypsin, and Trypsin). As such, no metaproteomic-specific datasets were used for training or validation. However, our novel FDR validation strategy demonstrates that peptides identified via DIA-PASEF, whether incorporated into the reduced FASTA from novoMP or from a classical database search, are supported with similar confidence (Fig. 2c).

In addition, we have uploaded all raw training data to PRIDE under the dataset identifier PXD051792.

Comment 1.2: Previous Bruker public releases and application notes (e.g., for immunopeptidomics) have indicated that BPS-Novor was trained with 1.75 million PSMs. Were the same datasets used in this study? If so, were there any specific adaptations or optimizations for metaproteomics? If such optimization were made, more details are needed for this manuscript, particularly since the authors have renamed the approach as NovoMP.

Comment 1.3: If no specific optimizations were performed, I suggest keeping the name BPS-Novor to avoid potential misunderstandings.

We thank the reviewer for these important questions, which we address together.

We confirm that the 1.75 million PSMs used to train BPS-Novor are the same datasets referenced in the previously published Bruker application notes. As mentioned in our previous response, no additional training or optimization was performed using a specific metaproteomics dataset. This decision was based on the following considerations:

- **Fundamental similarity in peptide fragmentation:** The fragmentation patterns of peptides follow the same principles regardless of whether they originate from a single eukaryotic or prokaryotic

organism. The taxonomic origin does not fundamentally affect how the algorithm reconstructs sequences, as long as the spectra exhibit recognizable fragmentation patterns.

- **Diversity already captured in training data:** While the minimum amount of PSMs required to achieve training performance saturation is difficult to determine a priori, diversity in fragmentation patterns is particularly important for *de novo* approaches in metaproteomics, where endogenous proteolysis may deviate from classical tryptic digestion. The existing BPS-Novor training set includes 1.75 million PSMs, incorporating a wide range of peptide sequences, modifications, charge states, and fragmentation behaviors across multiple enzymes.
- **Empirical validation in metaproteomics using well-defined species/strains:** Complex metaproteomic samples cannot serve as true ground truth for algorithm training or validation. Despite the strong performance of novoMP in our study (as shown by our FDR validation strategy), we recognize the potential value of training with defined metaproteomic datasets. This will be explored in future work involving the culturing of hundreds of individual bacterial species or strains followed by deep DDA- or DIA-PASEF acquisitions, creating comprehensive training resources. However, due to the magnitude of such project, the current study focuses on developing new tools for evaluating *de novo* sequences in metaproteomics, including multi-tier filtering and FDR validation using DIA-PASEF.

The name "novoMP" was chosen to represent the complete *de novo* sequencing-based workflow we implemented. This workflow goes beyond the initial *de novo* sequencing step with BPS-Novor (Step 1), incorporating additional multi-tier filtering to enhance peptide confidence (Step 2), constructing a microbial protein database based on *de novo* peptides (Step 3), and an FDR validation framework using DIA-PASEF (Step 4).

This distinction is important: Steps 2 to Step 4 are independent of the specific *de novo* algorithm used, thereby offering a generalizable blueprint that others can apply to outputs from their preferred *de novo* tools.

We now clarify this naming and methodological distinction in a new Method section named **Components of novoMP workflow**. We apologize for any confusion and thank the reviewer for encouraging us to elaborate on this point.

Comment 2: Line 96: the authors indicated the identification of 551 additional species using NovoMP-DB. Could the authors include a figure panel to illustrate these species and discuss their previous evidence of presence in mouse/human gut? A Venn diagram comparing species/taxa identified by NovoMP and DB-search would also be informative.

We thank the reviewer for the helpful suggestion, which has helped to clarify key aspects of our results.

As shown Figure R4 below, the 551 unique species identified through novoMP-DB were predominantly mapped to the Bacteria (40.5%) and Eukaryota (59.5%) superkingdoms. Notably, within the Eukaryota group, 31.1% were annotated as fungi, consistent with our earlier observation (previous Fig. 1d) that *de novo* sequencing improves the detection of non-bacterial peptides. This highlights the added value of the novoMP approach in capturing microbial diversity beyond the bacterial domain.

Importantly, the reference database used for conventional DB search—derived from the MGnify mouse gut catalogue—is heavily biased toward bacteria, with only two archaeal species represented. This bias may partly explain the increased detection of non-bacterial taxa in the *de novo* workflow, and it makes it

difficult to estimate the value of the previous presence or the absence of a given species in mouse/human gut catalogues as a proxy of taxonomic confidence.

To ensure taxonomic relevance and reduce the likelihood of spurious assignments, we subjected all *de novo* peptides to BLAST-based homology searches restricted to Bacteria, Fungi, Archaea, and Viruses, as described in the Methods section (formerly lines 725–728). This allowed us to exclude peptides likely originating from dietary sources (e.g., plants, insects) or with high sequence similarity to host proteins. Furthermore, to enhance the robustness of novoMP annotations, we constructed a novoMP-DB and validated its performance using a DIA-PASEF dataset, as detailed in Fig. 2 and Supplementary Fig. 2.

To further evaluate the relevance of the 551 additional species identified by novoMP-DB, we have performed a new comparison using the EMGC (Expanded Mouse Gut Gene Catalog; doi.org/10.1128/msphere.01119-20). We found that 56% of these species overlapped with those reported in EMGC, supporting the biological plausibility of our identifications. It is worth noting that, as indicated above for the MGnify database, the EMGC is predominantly composed of bacterial species (91.2%), with smaller proportions assigned to Archaea (2.9%), Eukaryota (3.7%), and Viruses (2.2%). Of note, when focusing specifically on the bacterial subset of the 551 species identified by novoMP-DB, we observed a 92% overlap with bacterial species reported in EMGC. These findings further support the taxonomic validity of our *de novo* identifications and highlight the potential of the novoMP approach to expand species-level resolution, particularly for non-bacterial members of the gut microbiome.

Figure R4: Phylotree map (Superkingdom to Class) of 551 annotated species from novoMP workflow.

Due to the fundamental algorithmic differences between *de novo* sequencing (i.e., BPS-Novor) and conventional database search methods (i.e., FragPipe), the two approaches shared 87 annotated species in common (Figure R5A) when both were applied to the same DDA-PASEF dataset. This limited overlap, combined with the known taxonomic bias of the metagenome-derived reference database (see above), underscores the value of *de novo* sequencing in uncovering the "hidden" microbial diversity within DDA datasets. To further assess the confidence of species-specific peptide annotations, we compared *de novo* scores of peptides mapped to the 87 shared species versus those uniquely assigned to the 414 additional

species identified by novoMP. As shown in Figure R5B, both groups exhibited similar *de novo* score distributions, suggesting comparable confidence levels in peptide identification, regardless of whether the species were shared with the conventional database search or unique to novoMP.

Due to the importance of these topics, we have now incorporated these new analyses and explanations as Supplementary Fig. 6, and in the Discussion section (lines 430-440).

Figure R5: A) Venn diagram of annotated species (cut-off: minimum 3 species-specific peptides) using peptides identified from two workflows. B) Histogram shows the distribution of *de novo* score for *de novo* peptides annotated either to 87 common (blue) or 414 unique species (green).

Comment 3: An 80% sequence identity for blastp has been used in this study. Could the authors justify this choice? Given that peptide sequences identified using bottom-up proteomics are relatively short (7–30 amino acids or 21–90 nucleotides), whereas classical BLAST searches typically use sequences >100 nucleotides, was any evaluation performed to determine the suitability of the 80% identity threshold in this context?

We appreciate the reviewer's concern regarding the selection of an 80% sequence identity threshold for BLASTP, particularly given the short length of peptides identified via bottom-up proteomics.

While peptides from database searches are exact matches to reference sequences and would yield 100% identity in BLASTP (though this is often unnecessary), *de novo*-derived peptides are more prone to sequence ambiguities due to limitations in fragmentation spectrum interpretation. Minor misassignments such as isoleucine ↔ leucine or glutamine ↔ lysine are well-recognized hurdles (doi.org/10.1074/mcp.O111.014902).

Our choice of an 80% identity threshold was guided by several considerations rooted in current best practices for analyzing *de novo*-derived peptides via BLASTP:

- **Accounting for *de novo* sequencing inaccuracies:** State-of-the-art workflows that implement *de novo* sequencing tools (e.g., PEAKS+SPIDER, Novor+MS-BLAST) commonly apply relaxed identity thresholds to account for sequencing errors. Recommended criteria include:
 - E-value ≤ 1
 - Identity ≥ 70%
 - Coverage ≥ 80% of the peptide

In line with these standards, our threshold of $\geq 80\%$ identity ensures that minor inaccuracies in *de novo* peptide predictions do not disqualify true homologs from being identified.

- **Use of the PAM30 scoring matrix:** To enhance alignment performance for short peptides, we employed the PAM30 substitution matrix, which is specifically optimized for alignments with expected sequence identities of 75% or higher (doi.org/10.1002/0471250953.bi0305s43; <https://blast.ncbi.nlm.nih.gov/doc/blast-help/>). Short peptides are subject to fewer evolutionary substitutions; thus, lower identity thresholds could lead to increased false positives.
- **Precedent in related research:** Similar thresholds have been used in studies involving peptide homology searches where reference databases are limited. For instance:
 - In epitope prediction, sequences with $\geq 80\%$ identity and a minimum match length of 8 amino acids were classified as B cell epitopes (doi.org/10.1038/s41598-022-18021-1).
 - In antimicrobial peptide discovery, sequences with $\geq 80\%$ identity were considered related to known AMPs (doi.org/10.1186/s12864-018-5225-5).
- **Empirical evaluation in our dataset:** We further analyzed the distribution of sequence identity scores among our *de novo* peptide matches. The resulting histogram (see new Figure R6) revealed distinct peaks around 80%, 90%, and 100% identity. Notably, our 80% cutoff excluded approximately 12% of *de novo* peptides that had otherwise passed our multi-tier filtering steps. This finding highlights the balance we aimed to strike between stringency and inclusivity.

Figure R6: Identity distribution of *de novo* peptides with BLASTP analysis.

We have now included the above-mentioned references to justify this threshold in the revised Methods section.

Comment 4: In general, I would be very careful to make claims that metaproteomics achieves higher coverage than 16S or metagenomic sequencing. Specifically, in Figure R9/S4B, the large differences in the number of identified taxa and the low number for shotgun sequencing is surprising (even given the fact of shallow shotgun sequencing).

We thank the reviewer for highlighting the need for caution when comparing metaproteomic data to results from 16S rRNA and metagenomic sequencing.

The observed improvements in taxonomic resolution from our uMetaP workflow, relative to 16S rRNA, are in line with previous studies (doi.org/10.1128/spectrum.01466-22; doi.org/10.1016/j.watres.2023.120700; doi.org/10.1016/j.mcpro.2022.100197) that employed DDA-based metaproteomics, which generally provides lower taxonomic coverage than our DIA-PASEF-based workflow (doi.org/10.3389/fmicb.2023.1258703).

It is well-recognized that 16S rRNA sequencing, while efficient for genus-level profiling, has limitations in resolving species-level diversity. This is due to the conserved nature of the 16S rRNA gene and limited divergence within its variable regions (doi.org/10.1186/s12864-024-10213-5; doi.org/10.1093/ismeco/ycae034). Although universal primers enable broad bacterial detection, closely related species often cannot be distinguished.

As shown in Figure R7, a double-blinded comparison of 21 mouse fecal samples analyzed by 16S rRNA sequencing (performed by the Joint Microbiome facility at the University of Vienna) and uMetaP confirmed that metaproteomics provided broader taxonomic coverage across all taxonomic levels.

Figure R7: Comparison of detected taxa between the previous workflow (performed on a timsTOF Pro instrument), uMetap and the 16S analysis of the same samples (n=21). A minimum of 3 taxa-specific peptides was set as cut-off for the metaproteomic data. Note that: 250ng of peptides were analyzed in a 2-hour gradient on the timsTOF Pro (our previous workflow), while only 50ng peptides were analyzed in 1-hour gradient in uMetaP. This figure was shown in the previous response letter as Figure R8.

Nevertheless, we fully agree with the reviewer that a more meaningful benchmark would involve comparisons with deep metagenomic sequencing or latest long-read technologies, which lie beyond the scope of the present study.

Given that this comparison is not central to our manuscript, and to avoid potential misinterpretation, we have removed Supplementary Fig. 4B and the corresponding discussion in the revised version.

Comment 4.1: For metaproteomics, this study uses a threshold of three unique peptides for taxon identification, but was this criterion validated for this particular dataset? A statistical strategy is needed to justify the use of 3 unique peptides for confidently identifying a species, in particular when the authors make claims for taxonomic coverage in this study.

We thank the reviewer for raising this important point regarding the threshold for taxonomic identification in metaproteomics.

Given the incompleteness of current microbial reference databases and the inherent limitations of lowest common ancestor (LCA) algorithms used in tools such as Unipept and MetaLab, the specificity of species-level annotation requires careful justification. Our rationale for using a threshold of three taxon-specific peptides is supported by both previous literature and newly performed data analysis, as described below:

- **Established practice in the field:** A three-peptide threshold for species-level identification has been widely adopted in key metaproteomics studies (e.g., doi.org/10.1038/s41467-018-05357-4; doi.org/10.1080/19490976.2021.1994836). This standard reflects a balance between specificity and sensitivity in microbial taxonomic profiling.
- **Support from benchmarking studies:** A recent evaluation (doi.org/10.1016/j.mcpro.2024.100840) using artificial microbial communities found that a three-peptide cutoff achieved the highest F-score at the genus level (Figure 5C of the referenced study). At the species level, both three- and five-peptide cutoffs provided comparable results in a 32-species test community (Figure 5D of the referenced study), reinforcing the robustness of the three-peptide threshold.
- **New FDR analysis in our dataset:** To validate the three-peptide threshold in our own study, we conducted a decoy-based analysis. Our DIA-PASEF dataset yielded 210,051 peptides using the uMetaP workflow, enabling the annotation of 825 species supported by ≥ 3 taxon-specific peptides (Fig. 2).

We generated decoy peptides by randomizing the target sequences while ensuring no sequence overlap with real or other decoy peptides. Both the target and decoy peptide sets were submitted to MetaLab for taxonomic annotation. We then calculated the F-score and false discovery rate (FDR) for taxonomic identifications across thresholds from 1 to 10 peptides using the following definitions:

$$F = 2 \times \frac{\text{Precision} \times \text{Recall}}{\text{Precision} + \text{Recall}}, \text{ where: Precision} = \frac{\text{TP}}{\text{TP} + \text{FP}} \text{ and Recall} = \frac{\text{TP}}{\text{TP} + \text{FN}}$$

- True Positive (TP) = number of species uniquely identified from target peptides
- False Negative (FN) = number of species identified by both target and decoy peptides, but with more supporting peptides in the target set
- False Positive (FP) = number of species identified by both sets, but with equal or more peptides from decoys

We repeated this randomization five times to generate independent decoy datasets. The results are summarized in new Figure R8 below. The F-score increased sharply from 1 to 2 peptides and plateaued at 3 peptides (mean F-score = 0.987), while the FDR dropped to 0.012. Increasing the

cutoff to 4 or 5 peptides provided minimal gains in specificity but significantly reduced sensitivity, lowering the number of identified species from 825 (3-peptide cutoff) to 382 (5-peptide cutoff), as shown in the accompanying table.

Figure R8: F-score (blue) and FDR (green) of species annotated and filtered by different numbers of corresponding peptides as cut-offs. The gray-dotted line indicates FDR=0.01. The number of annotated species at different cut-offs is shown in the adjacent table.

These results support the use of a 3-peptide cutoff as a statistically sound and biologically meaningful criterion for species-level taxonomic assignments in our dataset.

Comment 5: Figure 1A: why there is no Pro-PSM?

We appreciate the reviewer’s attention to the difference between Fig. 1a and Supplementary Figure 1A in our initial submission, where the identified PSMs from timsTOF-Pro were shown.

Fig. 1 is designed to introduce the novoMP workflow. It serves two main purposes: first, to illustrate the substantial increase in the number of precursors fragmented using the timsTOF Ultra compared to the timsTOF Pro; and second, to highlight the large number of fragmented precursors that did not yield any PSMs using the timsTOF Ultra. This observation underscores the opportunity and need for *de novo* sequencing solutions in metaproteomics, especially when performed in the latest high-resolution mass spectrometers.

For this reason, we chose not to display the Pro-PSM data in Fig. 1A, focusing instead on the broader conceptual motivation for novoMP.

Comment 6: Figure 2E: could the authors explain what are the remaining 264 species (631- 253- 114 =264)? The figure is somewhat unclear, and additional explanation would help improve its interpretation.

We apologize for the lack of clarity in our explanation of Fig. 2e.

The 631 represents species annotation with peptides contributed from novoMP-DB. The 253 represents species annotation which only passed a minimum of 3 peptides cut-off due to the addition of peptides from novoMP-DB. The 114 represents species annotation uniquely using novoMP-DB peptides. The 264 species represent those that were co-annotated with peptides from both DB-search and novoMP-DB, but that could be identified through DB-search alone using the cut-off of 3 species-specific peptides. The peptide identifications from novoMP-DB added more peptides to those 264 species.

We have revised the figure legend (lines 186-192) to ensure clearer interpretation. Thank you for pointing this out.

Comment 7: Line 170: is this based on enrichment analysis? What are the statistical criteria used for this 'detailed functional analysis'?

The functional annotations of identified microbial proteins were not based on enrichment analysis. As described in the Methods section ("*Taxonomic and functional annotation and quantification*"), we assigned functional annotations to the entire microbial protein database using EggNOG-mapper. Identified proteins were subsequently mapped to these annotations.

To our knowledge, functional enrichment analysis typically requires a well-characterized reference database with a substantial proportion of genomes carrying known functional annotations. In microbiome research, this remains a major challenge, as many microbial proteins—particularly from uncultured or newly discovered species—lack functional characterization. This limitation reduces the feasibility of comprehensive enrichment analysis across diverse microbial communities.

We apologize for any confusion caused by the phrase "detailed functional analysis." As shown in Supplementary Fig. 2d, proteins identified from different workflow sources were mapped across all 24 Clusters of Orthologous Genes (COG) categories without major differences in global distribution. However, a closer inspection of individual categories (Fig. 2f) revealed that proteins uniquely identified through the novoMP-DB were disproportionately represented in specific COG classes, including RNA processing (COG-A), chromatin dynamics (COG-B), extracellular structures (COG-W), nuclear structure (COG-Y), and cytoskeleton (COG-Z).

We appreciate the reviewer's suggestion and have removed the term "detailed functional analysis" from the revised manuscript to avoid misinterpretation.

Reviewer #5 (Remarks to the Author):

Comment 1.1: Many of the issues highlighted by the previous reviewers remain unresolved.

We appreciate the reviewer's feedback. However, we would like to respectfully point out that the other two reviewers expressed their satisfaction regarding our efforts to address all the previous issues:

- **Reviewer #3:** *"Comment 1.1: The authors have adequately addressed my previous concerns."*
- **Reviewer #4:** *"Overall, they have addressed most of the reviewers' comments. However, I do have some comments regarding the newly added de novo sequencing section."* (Note: It is worth noting that Reviewer #4's remarks primarily pertain to newly added material, which we address in this reply, fully.)

Below, we provide detailed, point-by-point responses to all newly raised comments from Reviewer #5.

Here is a few examples:

Comment 1.2: Supplementary Table 1: The data shows that 774 species are identified with 3 or more peptides. However, this number drops significantly to 345 species with at least 5 peptides and only 131 species with 10 or more peptides.

Comment 1.3: Relying on 3 peptides to identify a bacterial species is not appropriate, as the false discovery rate (FDR) would likely be high, given the total number of peptides identified.

We thank the reviewer for these comments, which we address together.

The observed decrease in the number of identified species as the peptide count threshold increases reflects the dynamic range of species abundances in the gut microbiome. This trend is expected and consistent with previous metaproteomic studies (e.g., doi.org/10.1038/s41467-018-05357-4; doi.org/10.1186/s40168-017-0290-6), where highly abundant species yield more peptide identifications, while low-abundance taxa may be represented by only a few peptides. This behavior is well-documented and rooted in biological and technical limitations, including any mass spectrometry's technology incomplete coverage of species-specific peptide sequences (similar to genomics or other -omic technology).

To "Comment 1.3" and to address a related point also raised by Reviewer #4 (Comment 4.1), we now present an extended analysis showing three lines of supporting evidence—including a new experimental validation using our dataset to evaluate the impact of this cutoff on FDR—supporting the appropriateness of a three-peptide cutoff for species-level identification. These results are presented in Figures R8 (please, see above for the answer to Reviewer #4).

To avoid any misunderstanding related to "the total number of peptides identified", we would like to clarify that not all identified peptides can be confidently annotated at the species level. This is due to both the limitations of reference databases and sequence conservation across taxa. For example, in our DDA-PASEF dataset combining novoMP and database search results, a total of 300,239 peptides were identified, yet only 60,308 were assigned to any taxonomic rank (see Supplementary Data 1). Of these, just 20,293 peptides received species-level annotation prior to applying the three-peptide filter. Similarly, in the DIA-PASEF dataset used for Fig. 2, 210,051 peptides were identified, but only 15,520 could be

annotated to species level. These results illustrate the challenge of achieving high-resolution taxonomic assignments in metaproteomics.

Collectively, the new analysis demonstrate that this threshold provides a robust balance between sensitivity and specificity in complex microbiome samples.

We recognize that this is a critical and not well-addressed topic in the field of metaproteomics, and we hope that the additional analyses presented here help to clarify our rationale and provide further support for the chosen approach.

Comment 1.4: This raises concerns about the accuracy of species identification, particularly when comparing metaproteomics to 16S rRNA sequencing and metagenomics. The latter two methods are inherently more accurate for species identification.

To discuss this important point, it is worth defining the term accuracy in taxonomical terms.

Taxonomical accuracy tells how well the analytical method (e.g., 16S rRNA sequencing, metaproteomics, metagenomics) assigns genomic sequences or peptides to the correct taxonomic group.

Taxonomic accuracy can be assessed in different ways depending on the context:

Using simulated or mock communities

These are artificial samples where the species composition is known. Here, accuracy = how many identifications matched the known species/taxa. In this context, previous literature demonstrates similar or better accuracy of metaproteomics compared to 16S rRNA in pre-defined synthetic communities (doi.org/10.1038/s41467-017-01544-x).

Benchmarking against reference databases for complex samples

Accuracy refers to how many assignments match curated taxonomies (e.g., NCBI, GTDB). Misassignments or ambiguous mappings reduce accuracy. In Fig. 2 of the manuscript, we used a cutoff of at least three species-specific peptides per species, which rendered a total of 825 species confidently annotated. Among these, 84% overlapped with the EMGC catalogue, and 94% of the annotated bacterial species were also present in EMGC, providing strong support for the biological relevance and accuracy of our species-level assignments.

Moreover, we would like to comment on different aspects that influence taxonomic accuracy (as defined above):

- It is important to point out that in metaproteomics a taxonomic call-out relies on the genomic database used for producing the metaproteomic fasta used during the analysis of the MS/MS data. In other words, the taxonomic accuracy in metaproteomics is as good and as accurate as the genomic database is.
- The accuracy at a specific taxonomic level depends on the depth of the analytical method. In this sense, shotgun metagenomics is more accurate than 16S rRNA at the species level, as the latter is often limited due to high sequence similarity, particularly with short-read amplicons (e.g., V3–V4 regions), among closely related taxa.
- Metaproteomics tends to be quite conservative in low taxonomical rank (i.e. species-level) identification. To address the challenge of sequence redundancy, lowest common ancestor (LCA) algorithms have been developed to improve the specificity and reliability of taxonomic assignments in metaproteomic data.
- Increasing the number of taxa-specific peptides used for taxonomic classification will increase the taxonomic confidence and reduce FDR. For the latter, we showed that our current three species-specific peptide threshold is appropriate (see above).

- For complex communities (i.e. fecal samples, like the ones used in our study), metagenomic databases are considered the ground-truth. In our experiments, we have used one of the state-of-the-art metagenomic database (MGnify mouse gut catalog).

As an indication of the taxonomic accuracy reached in our study in a physiologically meaningful scenario, we mirrored and extended the taxonomic changes revealed by metagenomics (doi.org/10.1016/j.chom.2024.06.013) on an in-vivo mouse model of inflammatory disease.

In summary, the previous literature and the in-vivo taxonomic results strongly suggest that our taxonomic assignments are not more inaccurate than 16S rRNA or metagenomic methods.

Given that the metaproteomic vs. 16S rRNA comparison is not central to our manuscript, and to avoid potential misinterpretation, we have removed Supplementary Figure 4B and the corresponding discussion in the revised version.

Comment 2: Furthermore, Supplementary Figure 4B is problematic. The expected trend should show metagenomics detecting more species than 16S rRNA sequencing, which in turn should detect more species than metaproteomics. The analysis should be done using deep sequencing metagenomics to accurately determine the species present and validate the findings.

We thank the reviewer for this insightful comment and fully agree that a meaningful comparison of taxonomic coverage across methods should ideally be performed using deep metagenomic sequencing, which offers higher resolution and more comprehensive detection of species.

We acknowledge that Supplementary Figure 4B, as presented, may have caused confusion by not using the best technique of each approach. To avoid potential misinterpretation and as this supplementary figure is not central to our manuscript, we have removed Supplementary Figure 4B and the corresponding discussion in the revised version.

Comment 3.1: Table 2 also presents significant issues, particularly regarding the claim that the method can detect species down to 10 pg.

It seems there may be a misinterpretation of our text. In lines 147-148 (previous submission) we wrote “uMetaP detected an average of 200 microbial and 76 host protein groups at an ultra-low sample amount of 10 pg (Figure 2A and Supp. Table 2).”. Thus, in the submitted manuscript we did not state that species were detected at 10 pg. Rather, we described the identification of microbial peptides and proteins down to 10 pg using uMetaP powered by DIA-PASEF.

Comment 3.2: Of the peptides identified in the microbiome at the 10 pg level, 56 are not detected in all of the 100 ng experiments. Among these, 29 are entirely absent in the 100 ng experiments, and 11 are detected in only one experiment. These peptides are very likely to be false positives.

Thank you for the detailed analysis of the data and for pointing out a known limitation of the field of proteomics, therefore not specific to our study: the confidence of identifications, particularly with extremely low sample inputs (e.g., 10 pg). Our new analyses demonstrate that these 29 peptides are not false positives.

First, among the 211 peptides detected at 10 pg, all (including the 56 mentioned by the reviewer) were also detected in several of the samples with higher inputs, meaning no peptides were uniquely identified in the 10 pg samples. To further assess the identification confidence of the 29 peptides that were completely absent in the 100 ng samples, we extracted their spectra from the raw DIA-PASEF data at 10 pg using Skyline. As shown in Figure R9, a comparison with the 50 ng samples revealed matching retention times and fragment patterns, strongly supporting the confidence of these peptide identifications at 10 pg and indicating that they are not false positives.

Figure R9: Extracted ion chromatograms of example peptides identified at 10 pg, in comparing with 50 ng samples.

In addition, we took the opportunity to understand the potential reasons for the absence of those peptides in 100 ng samples, as we did see matched ion chromatograms of most peptides in 100 ng samples with Skyline (examples in Figure R10 below). After checking the outputs of DIA-NN process, we found that 21 of the 29 peptides appeared in the output after the first-pass (where individual raw file was searched against the predicted spectra library). This finding offered us hints of: 1) the missing 8 peptides after the first-pass search were likely false negatives, given that the Skyline spectra suggest strong ion features for these peptides; 2) As DIA-NN re-calculates FDR for the second-pass (match-between-run), these 21 missing peptides did not make to the quantification matrix (previous Supplementary Table 2) likely due to the run-specific FDR cut-off applied (1%), suggesting those missing identifications in 100 ng experiments are likely false negatives. After consulting Prof. Vadim Demichev (DIA-NN developer), several reasons can lead to the missingness in specific runs/samples such as: 1) Random RT-dependent fluke in MS/MS calibration that leads to multiple fragments of the precursor falling outside the mass tolerance window; 2) Presence of highly abundant precursors affecting everything else in the TIMS device, potentially interfering with width of peaks in ion mobility dimension. This can interfere with DIA-NN's peak picking and the mobility calibration within the specific frame; 3) Poor match between observed and predicted spectra, specifically for high peptide loads (e.g., 100ng) but not low loads (e.g., 10 pg). Reasons mentioned here have some probability of happening, but none of them indicate any issue with FDR in DIA-NN.

Figure R10: Extracted ion chromatograms of example peptides identified at 10 pg, in comparing with 100 ng samples.

Finally, we would like to point out that the goal of this experiment was to explore the limits of detection (at the peptide input level) in a metaproteomic sample. Together with the results from Fig. 3, these experiments set new detection thresholds in the field.

Comment 3.3: As well, the peptide LSAEFGSLR is reported at one-third the intensity in the 10 pg sample compared to the 100 ng sample, which is not possible.

We thank the reviewer for her/his observation that, as for the latter comment, touches on the current limitations in quantitative proteomics and, therefore, is not specific to our study.

Upon further inspection, we found that the Skyline-extracted signal for this peptide showed a difference of over 200-fold between the 100 ng and 10 pg samples (see Figure R11 below), which is relatively closer but still far from the 10,000-fold corresponding to the dilution factor.

In general, it is important to consider that the observed discrepancy is based on several known factors including the ionization efficiency, background noise, signal saturation, ratio compression, and stochastic sampling effects—especially in ultra-low-input samples such as 10 pg.

The observed deviation reflects the challenges of quantification at the extreme ends of a wide dynamic range. As noted in the DIA-NN documentation, the recommended quantification model (QuanUMS) was optimized for comparisons within a ~10-fold dynamic range. Our dataset spans a much broader range (up to 10,000-fold), which could influence quantification performance under such conditions. Upon consultation with Prof. Vadim Demichev (DIA-NN developer), he confirmed that no quantification algorithm in proteomic is tested with such a broader range (10,000-fold), as these tools assume that biological experiments aiming to identify biologically meaningful differences use consistent peptide input amounts across samples, wherever possible, to minimize potential artifacts introduced by extreme sample-to-sample variation. For this reason, our 100 ng vs 10 pg comparison (shown in Supp. Table 2) was not meant to test quantification accuracy.

To avoid confusion, we have removed the peptide intensity columns from Supplementary Data 2, as these values were not used in the analyses presented in Fig. 2.

We appreciate the reviewer's insight, which helped us improve the clarity of our data presentation and acknowledge the practical limitations of current tools when applied to extreme dynamic ranges.

Figure R11: Extracted ion chromatograms of LSAEFGSLR peptide identified at 10 pg (3 columns on the left) and 100 ng samples (3 columns on the right). The peak area was log₁₀-transformed.

Comment 4: Additionally, I performed a BLAST search on a few *de novo* sequenced peptides claimed to be of bacterial origin. These peptides show strong homology to mouse and human proteins. This raises the question of whether these peptides are truly of bacterial origin or if they are contaminants from human/mouse samples or misassigned to bacteria. This issue further undermines the reliability of the species identification.

We thank the reviewer for raising this important point. We fully understand the concern regarding the origin of *de novo* sequenced peptides and the potential for misassignment to bacterial taxa, particularly when some peptides may show homology to mouse or human proteins. We would greatly appreciate it if the reviewer could specify which peptides were tested in the BLAST search, as this would allow us to investigate these specific cases more thoroughly. Regardless, as explained below, we have applied several stringent filters along the pipeline to ensure that the bacterial taxonomic annotations done after DIA-PASEF analyses are reliable. As a test of the taxonomic reliability reached in our study, we mirrored and extended the taxonomical changes revealed by metagenomics on an *in-vivo* mouse model of inflammatory disease.

We believe this concern may relate to Supplementary Data 1, which summarizes the taxonomic assignments and peptide origins. Specifically, 10,491 peptides identified through the novoMP workflow were annotated to the Bacteria superkingdom across different taxonomic ranks. It is also worth noting that 483 *de novo* peptides were annotated as mouse-derived, which is consistent with the fact that the samples originated from the mouse gut microbiome.

To clarify, taxonomic annotation was performed using MetaLab, which applies a 100% sequence identity when matching peptide inputs to reference database. While this stringent approach helps reduce false positives—such as those arising from shared homologous sequences—it does not entirely eliminate the possibility that short, conserved peptides may be shared between bacterial and eukaryotic proteins. To further improve specificity, lowest common ancestor (LCA) algorithms were applied by MetaLab, ensuring that a peptide is only annotated to a particular taxonomic group (e.g., Bacteria) if no identical match exists in other domains, such as Eukaryota. However, we acknowledge that the annotation database itself has limitations, including incompleteness, which may impact taxonomic resolution.

Additionally, as described in the Methods section (previous lines 725–728), *de novo* sequenced peptides were subjected to a BLASTp search to retrieve representative protein sequences. This search was

restricted to microbial taxa, including Bacteria, Archaea, Fungi, and Viruses, to exclude peptides potentially originating from dietary sources (e.g., plants, insects) and those with high similarity to host proteins (e.g., mouse). This step helped reduce the likelihood of taxonomic misannotation due to conserved sequences shared with the host proteome. The resulting microbial protein database (from both database- and denovo-derived peptides) was then used to process the DIA-PASEF dataset presented in Fig. 2 of the manuscript. Using a cutoff of at least three species-specific peptides per species, a total of 825 species were confidently annotated. Among these, 84% overlapped with the EMGC catalog, and 94% of the annotated bacterial species were also present in EMGC, providing strong support for the biological relevance and accuracy of our species-level assignments.

REVIEWERS' COMMENTS

Reviewer #3 (Remarks to the Author)

Comment 1: The authors have addressed my previous concerns. I have no additional comments.

We thank the reviewer for their positive feedback and are pleased to know that all previous concerns have been satisfactorily addressed. We are grateful for the constructive suggestions and comments provided throughout the review process, which have helped to improve the quality of our manuscript.

Reviewer #4 (Remarks to the Author)

Comment 1.1: I would like to thank the authors for taking another significant effort to address the comments raised by myself and other reviewers.

We sincerely thank the reviewer for their kind words. We appreciate the valuable feedback and constructive suggestions provided throughout the review process, which have greatly contributed to strengthening our manuscript. For the newly raised comments, we provided a point-by-point reply below.

Comment 1.2: It was encouraging to see that the authors removed the comparison of taxonomic coverage between metaproteomics and metagenomics/16S data. This was one of the most concerning aspects of the previous version of the manuscript, and its removal addresses several of the concerns raised by both myself and Reviewer #5.

We thank the reviewer for acknowledging the revision. We agree that the removal of the comparison has helped to clarify the scope of the manuscript.

Comment 2.1: For the justification of using 3 unique peptides for species identification, I am not convinced by the authors. The way for the calculation of the FDR and F-values seems to be very questionable. Both MetaLab or Unipept assign taxonomic annotations based on exact match of identified peptide sequences to the sequences in their databases; it would be very surprising that the decoy peptide sequences generated by the authors could have a lot of matches in the databases. As a result, the 'false positives (FP)' would be very low, leading to very low FDR values across all levels. In my opinion, the unique peptide number threshold selection might be highly relevant to the depth of identification in metaproteomics. In particular in this ultra-deep dataset using DIA, the accuracy of species identification could be likely impacted/compounded by the potential lower confidence of peptide identification as well. As said this is a known challenge of the field, not limited to this work itself.

We thank the reviewer for raising these thoughtful points. We agree that defining FDR-controlled and robust thresholds for species-level assignments remains an open challenge in metaproteomics, particularly in the context of ultra-deep DIA datasets.

Our approach, including the calculation of FDR and F-scores based on taxonomic annotation, was inspired by the framework proposed by Sun et al. (doi.org/10.1016/j.mcpro.2024.100840). While their study used publicly available data obtained from a synthetic community (SynCom) with defined taxonomic compositions and did not employ randomized peptides to estimate false annotations, our adaptation incorporates such randomization to approximate FDR in complex, real-world samples without known

ground truth. While such decoy peptides should rarely match reference databases, our analysis showed that at relaxed thresholds (e.g., one peptide per species), false positive rates increased substantially.

We fully agree with the reviewer that the selection of a peptide threshold for confident taxonomic annotation is closely tied to profiling depth and the intrinsic abundance of specific taxa. Additionally, as noted, the overall confidence in peptide identification is a critical factor, particularly in deep datasets acquired via DIA. To mitigate over-annotation risks, we applied stringent multi-tier peptide filtering, FDR control at multiple levels, and non-spike controls in spike-in experiments. Nonetheless, we now explicitly state in the revised manuscript that species-level assignments should be interpreted cautiously (please, see answer to Comment 2.2, below), and that newer benchmarking efforts using more complex SynCom will be critical for future consensus.

We appreciate the reviewer's acknowledgement that this represents a broader challenge in the field, and not a limitation unique to our study.

Comment 2.2: However it would be necessary to discuss this limitation particularly when the authors make conclusions/statements on the significantly increased coverage of taxonomic identification.

We thank the reviewer for this important suggestion.

As mentioned above, we have revised the Results and Discussion section to explicitly acknowledge this limitation:

- Lines 84–87: Moreover, the combination of peptides from DB-search + novoMP (Combined strategy) enabled the annotation of 551 additional species, *suggesting a potential increase in taxonomic coverage of up to 247%* (Fig. 1e).
- Lines 134–135: *This suggests a potential 6-fold improvement in detected species compared to our previous DIA-PASEF workflow.*
- Lines 358–363: *However, it is important to note that the estimated improvements in taxonomic profiling presented in this study may represent an upper bound. These estimates should be interpreted with caution due to current limitations in metaproteomics, including incomplete reference databases, uncertainty in species-level assignment thresholds, and the absence of robust FDR estimation methods for taxonomic annotation in complex metaproteomic samples.*
- Lines 377–378: *...and an estimated 80% increase in taxonomy detection* (Supplementary Fig. 2c; Fig. 2e).

Comment 3: A similar point applies to the analysis based on 10 pg input. The authors are correct in noting that the proteomic analysis of extremely low input samples is a known challenge in the field. This limitation should be stated in the manuscript along with the discussion of improved sensitivity. The analysis exploring missing peptide identifications in high-input datasets is insightful and helps address some of these concerns. I would recommend that this part of the analysis be included in the supplementary information and discussed in the main text when addressing the limitations and interpreting the results.

We appreciate the reviewer's acknowledgement of the broader challenges associated with proteomic analysis of ultra-low-input samples. In line with the reviewer's suggestion, we have added a new Supplementary Fig. 7 (see below) and expanded the Discussion section (lines 378–388) to address this limitation and interpret the results in the context of the observed missing identifications in high-input datasets.

Supplementary Fig. 7: Confidence assessment of peptides identified at low input levels. **a** Venn diagram showing the overlap of peptide identifications between 10 pg and 100 ng input samples. Twenty-nine peptides were detected at 10 pg but not at 100 ng. **b** Venn diagram comparing peptides identified at 10 pg with all other higher-input samples. All 211 peptides identified at 10 pg were also observed in at least one other sample. **c** Extracted ion chromatograms for two representative peptides (from the 29 in Supplementary Fig. 7a) detected in 10 pg but not reported in 100 ng samples. Similar fragment ion patterns support confident identifications at 10 pg and suggest that absence in 100 ng is likely due to false negatives rather than false discovery in 10 pg.

Comment 4: Line 44: uMetaP enables novoMP? Shouldn't this be novoMP enables uMetaP? There should be a clear definition of uMetaP, novoMP as well as an explanation of their relationship in the manuscript.

We appreciate the reviewer's comment and understand the potential confusion. Our intention with the phrase "uMetaP enables novoMP" was to convey that novoMP benefits substantially from the high-quality spectral data generated by uMetaP (e.g., increased precursor coverage

and improved signal-to-noise). However, we agree that the relationship should be stated more clearly. uMetaP refers to the overall ultra-sensitive metaproteomics workflow combining advanced LC-MS acquisition and analysis strategies, while novoMP is a dedicated *de novo* sequencing module that can be integrated within uMetaP or applied independently to suitable datasets.

We have revised the manuscript (Methods section “Components of novoMP workflow) to explicitly define both components and clarify their interdependency and modular use.

Comment 5: Supplementary Figure 1b-e: please make sure these data has not been previously published; otherwise these need to be removed or clearly indicate their prior publication or public release with links/references.

We thank the reviewer for raising this important point.

After consultation with the BPS-Novor developers (co-authors), we confirm that the data used in Supplementary Fig. 1b–e have not been previously published in any peer-reviewed journal. Only parts of Supplementary Fig. 1c and 1e were presented in poster format at the ASMS 2023 conference by co-authors of this manuscript (Tharan Srikumar, Qixin Liu, Bin Ma, and Jonathan Krieger), but they have not been formally published or publicly released elsewhere. Furthermore, the raw data used for Novor training have been deposited in PRIDE by our study for the first time.